# Cohort-Based Active Modality Acquisition

## Abstract

Real-world machine learning applications often involve data from multiple modalities that must be integrated effectively to make robust predictions. However, in many practical settings, not all modalities are available for every sample, and acquiring additional modalities can be costly. This raises the question: which samples should be prioritized for additional modality acquisition when resources are limited? While prior work has explored individual-level acquisition strategies and training-time active learning paradigms, test-time and cohort-based acquisition remain underexplored. We introduce Cohort-based Active Modality Acquisition (CAMA), a novel test-time setting to formalize the challenge of selecting which samples should receive an additional modality. We derive acquisition strategies that leverage a combination of generative imputation and discriminative modeling to estimate the expected benefit of acquiring a missing modality based on common evaluation metrics. We also introduce upper-bound heuristics that provide performance ceilings to benchmark acquisition strategies. Experiments on multimodal datasets with up to 15 modalities demonstrate that our proposed imputation-based strategies can more effectively guide the acquisition of an additional modality for selected samples compared with methods relying solely on pre-acquisition information, entropy-based guidance, or random selection. We showcase the real-world relevance and scalability of our method by demonstrating its ability to effectively guide the costly acquisition of proteomics data for disease prediction in a large prospective cohort, the UK Biobank (UKBB). Our work provides an effective approach for optimizing modality acquisition at the cohort level, enabling more effective use of resources in constrained settings.[1]

## 1 Introduction

Consider a clinical healthcare setting where all patients in a cohort undergo a standard, inexpensive set of initial examinations, such as basic blood tests and anamnesis. However, a more advanced, expensive, or invasive procedure, like genomic sequencing or specialized imaging, could offer crucial diagnostic or prognostic information for a subset of these patients (Huang et al., 2021). Given a limited budget or capacity for the more advanced procedure, the central question becomes: which patients should receive this additional resource to maximize the overall diagnostic yield or improve treatment outcomes across the entire cohort? For healthcare, budgets are often resource-specific rather than flexible. For example, a hospital may have a fixed capacity for one MRI scanner, or a cohort may have a specific grant for one modality. The critical decision is prioritizing access to that single resource across the cohort, and not necessarily dynamically acquiring for different modalities per patient. Consider a healthcare system that can afford 1,000 expensive tests for a 100,000-person cohort. The goal is to improve health outcomes across the whole population, and this typically happens through resource allocation: who receives preventive interventions, who gets enrolled in clinical trials, who is flagged for closer monitoring. These decisions depend on accurate risk stratification. A global ranking of all 100,000 individuals by predicted risk (measured, for example, by Area Under the Receiver Operating Characteristic (AUROC)) becomes the tool through which such allocation decisions are made. The question is: which 1,000 patients should we test so that our final ranking of all 100,000 is as accurate as possible? Balancing potential gains from data modalities against the costs and complexities of acquisition is not unique to healthcare. In remote sensing, for instance, decisions must be made regarding which geographical areas warrant costly high-resolution satellite imagery

---

[1]Code will be published on GitHub.

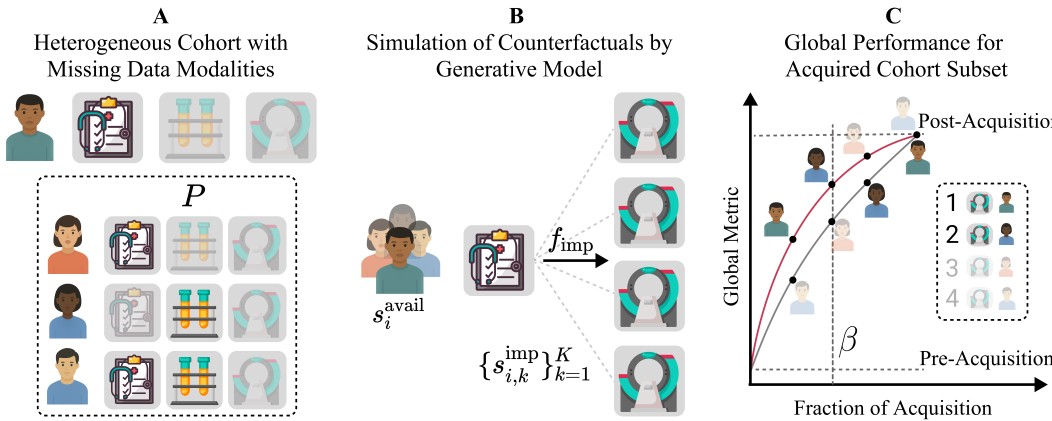

**A** Heterogeneous Cohort with Missing Data Modalities

**B** Simulation of Counterfactuals by Generative Model

**C** Global Performance for Acquired Cohort Subset

Figure 1: Motivational example for CAMA determining the added value of obtaining the magnetic resonance image (MRI) modality. **(A)** A heterogeneous cohort for which each sample has $P$ distinct modalities. **(B)** Instead of using the initial subset logit scores $s_i^{\text{avail}}$, a generative model $f_{\text{imp}}$ imputes the target missing modality for every patient in the cohort. This yields imputed, augmented-modality logit scores $\{s_{i,k}^{\text{imp}}\}_{k=1}^{K}$ that approximate the logits as if that modality were available. These scores approximate $s_i^{\text{acquired}}$, *i.e.,* the counterfactual with only the imputed modality added. **(C)** An acquisition function (AF) utilizes these scores to rank samples by acquisition priority. The graph demonstrates how the global performance metric improves from the initial baseline towards the performance of a model with access to post-acquisition data, as an increasing fraction of the cohort receives the additional modality. This acquisition process is guided by the proposed strategies operating under the acquisition budget constraint $\beta$.

to supplement widely available, lower-resolution data, aiming to optimize regional environmental monitoring under budget constraints. Likewise, in industrial quality assurance, manufacturers could decide which components from a production batch should undergo detailed, time-consuming testing in addition to rapid, standard visual inspections to effectively identify defects at a batch level. The topic of efficient data acquisition has led to several established paradigms in machine learning, such as Active Learning (AL) (Holzmüller et al., 2023), Active Feature Acquisition (AFA) (Shim et al., 2018), Active Modality Acquisition (AMA) (Kossen et al., 2023), and multimodal learning with missing data (Wu et al., 2024). However, previous research predominantly centers on optimizing acquisition for individual samples and often does not directly address test-time budget constraints for an entire cohort. Consequently, the strategic, test-time acquisition of an additional modality from a cohort perspective remains a significant, largely unaddressed gap. This setting involves deciding, for a given batch of new samples where different subsets of modalities are available, which specific samples should receive an additional, costly modality to best achieve a global objective, *e.g.,* maximizing overall predictive performance or diagnostic accuracy for the cohort, subject to budget constraints. We hypothesize that imputation-based acquisition functions (AFs) can effectively guide resource allocation under cohort-level constraints. The main contributions of this work are as follows:

- **The CAMA setting** We introduce and formalize CAMA, a previously unexplored setting that addresses the challenge of prioritizing which samples within a test-time cohort should undergo additional modality acquisition based on an available subset of modalities.

- **Development of AFs for CAMA** We propose a theoretical framework, derived from established evaluation metrics, *e.g.,* AUROC and Area Under the Precision-Recall Curve (AUPRC), that provides a foundation for developing AFs within the CAMA setting.

- **Architectures for CAMA** We develop novel architectures for approaching CAMA, including a) derivations of AFs by combining generative and discriminative deep learning and b) the definition of corresponding upper bounds to serve as performance benchmarks.

- **Comprehensive evaluation** We present a comprehensive empirical evaluation of our proposed methods across several multimodal datasets, which vary in their number of modalities and application domains, with up to 100,000 samples and 15 modalities. This includes an

analysis of key assumptions, upper bounds and oracle strategies, performance challenges, and robustness.

## 2    RELATED WORK

In the following, we contextualize our work on CAMA by reviewing the key concepts and contributions from several relevant research domains summarized briefly in Table 1.

Table 1: Comparison of active data acquisition paradigms. Our proposed CAMA setting is unique in its focus on cohort-level, test-time modality acquisition.

| Paradigm | Acquisition | Decision Level | Time | Primary Objective |
|---|---|---|---|---|
| AL | Labels | Individual | Training | Maximize model performance |
| AFA | Features | Individual | Test | Optimize sample-level prediction |
| AMA | Modalities | Individual | Test | Optimize sample-level prediction |
| CAMA (Ours) | Modalities | Cohort | Test | Maximize global cohort metric |

**Active Learning (AL)**    AL seeks to enhance model training by selecting unlabeled data points for annotation by an oracle (Settles, 2012; Ren et al., 2022; Li et al., 2025). Our methodology draws significantly from AL principles, particularly in the development of an AF to guide the selection process. Consequently, established AL strategies and concepts, such as those rooted in measuring uncertainty (Settles, 2012; Han & Kang, 2021; Hoarau et al., 2025; Raj & Bach, 2022; Ma et al., 2019) or using generative models (Tran et al., 2019; Zhu & Bento, 2017; Zhang et al., 2024; Ma et al., 2019; Peis et al., 2022), are central to our work. Existing work on multimodal acquisition (Rudovic et al., 2019; Das et al., 2022), batch-level selection (Ash et al., 2020; Kirsch et al., 2019; Holzmüller et al., 2023), and balanced AL (Aggarwal et al., 2020; Shen et al., 2023; Zhang et al., 2023; Hoarau et al., 2025) is especially relevant. Our approach, however, diverges from the conventional goals of directly optimizing model training or seeking labels for specific data points: We aim to identify those samples for which the acquisition of an additional data modality would be most beneficial.

**Active Feature Acquisition (AFA)**    AFA builds upon AL by focusing on selecting the most informative individual features for a given sample, often considering their acquisition costs (Rahbar et al., 2025). Similar to AL approaches, methods for AFA encompass a diverse range of techniques, including strategies based on measuring uncertainty (Hoarau et al., 2025; Astorga et al., 2024), the use of generative models (Li & Oliva, 2021; 2024; Gong et al., 2019; Zannone et al., 2019), and Reinforcement Learning (RL) (Valancius et al., 2024; Janisch et al., 2020; Kleist et al., 2025; Shim et al., 2018; Baja et al., 2025). Other common methodologies involve batch-level perspectives (Asgaonkar et al., 2024), leveraging information bottlenecks (Norcliffe et al., 2025), or employing the Kullback-Leibler Divergence (KL-Divergence) (Natarajan et al., 2018). Some AFA techniques rely on gradient calculations (Ghosh & Lan, 2023), while distinct approaches are formulated as individual, sequential recommender systems (Freyberg et al., 2024; Vivar et al., 2020). At an application level, even Large Language Models (LLMs), such as Med-PaLM 2 (Singhal et al., 2025), could be employed for AFA, although such deployments remain unexplored in this context. While our setting shares the core idea of AFA, it differs significantly: We are not concerned with the selection of individual features, but rather with identifying which entire data modalities to acquire. Furthermore, this decision-making process is applied at the cohort level, rather than optimizing for individual samples.

**Active Modality Acquisition (AMA)**    AMA can be conceptualized as an extension of AFA, distinguished by its focus on selecting entire data modalities rather than individual features or labels. Prominent related research includes approaches employing RL for multimodal data (Kossen et al., 2023; Jain et al., 2025; Li & Oliva, 2025) and methods utilizing submodular optimization in conjunction with Shapley values (Shapley, 1953; He et al., 2024). The approach by Kossen et al. (2023) differs from ours through its reliance on RL, whereas He et al. (2024) primarily investigate how modalities affect optimal learning performance. Further studies have explored the use of Gaussian mixtures within Bayesian optimal experimental design to enhance data acquisition efficiency for model training (Long, 2022). This objective differs from ours, as our focus is not on improving the

model training process itself, but rather on optimizing performance for a downstream task at test time. The relative sparsity of existing work for AMA underscores the significance of the research gap that our proposed setting, *i.e.,* CAMA aims to address.

**Multimodal Learning with Missing Data Modalities**   Research in multimodal learning with missing data modalities offers techniques for robustly handling incomplete datasets. These methods are broadly classified into strategy design aspects, *i.e.,* architecture-focused designs and model combinations, and data processing aspects, *i.e.,* representation learning and modality imputation (Wu et al., 2024). Acknowledging the utility of these approaches, our work emphasizes imputation-based strategies, and thus this paragraph highlights those methods. Imputation of missing features is commonly performed using Auto Encoders (AEs) (Hinton & Zemel, 1993), Variational Auto Encoders (VAEs) (Kingma & Welling, 2014), Generative Adversarial Networks (GANs) (Goodfellow et al., 2014), or Denoising Diffusion Probabilistic Models (DDPMs) (Ho et al., 2020; Rombach et al., 2022). These methods naturally extend to multiple modalities, for example, with VAE-based (Wesego & Rooshenas, 2024; Sutter et al., 2021; Lewis et al., 2021) and DDPM-based (Wang et al., 2023) approaches. Notably, the latter, *i.e., IMDer* (Wang et al., 2023), a multimodal deep learning architecture that imputes missing values with DDPMs in latent spaces, is adapted in our work (Section 5). However, this research area focuses on handling absent modalities rather than deciding which ones to acquire.

## 3   PROBLEM FORMULATION

Let $\mathcal{D} = \{(\boldsymbol{x}_i, \mathrm{y}_i)\}_{i=1}^N$ be a dataset of $N$ samples. For each sample $i$, the full feature set $\boldsymbol{x}_i$ is composed of $P$ distinct data modalities, $\boldsymbol{x}_i = \{\boldsymbol{x}_i^{(1)}, \ldots, \boldsymbol{x}_i^{(P)}\}$, and $\mathrm{y}_i \in \{0, 1\}$ is the corresponding binary label. In practice, only a subset of these modalities may be available. We denote the set of indices of available modalities for sample $i$ as $\mathcal{P}_i^{\mathrm{avail}} \subseteq \{1, \ldots, P\}$. Our goal is to decide for which samples to acquire costly missing data to maximize a cohort-level performance metric. This decision is guided by predictive scores (logits), and we consider three key predictive scores for each sample $i$:

- $\mathrm{s}_i^{\mathrm{avail}}$: The available score, computed using the subset of data modalities that are already observed for the sample.

- $\mathrm{s}_i^{\mathrm{acquired}}$: The acquired score, computed using the sample's available modalities plus the newly acquired modality.

- $\{\mathrm{s}_{i,k}^{\mathrm{imp}}\}_{k=1}^K$: A set of $K$ imputed scores that estimate the unknown $\mathrm{s}_i^{\mathrm{acquired}}$ using only the available data modalities.

For instance, given the example from Figure 1, in a simple clinical setting with a cheap, universally available base modality, *e.g.,* cardiac biomarkers such as troponin or B-type natriuretic peptide (BNP), and an expensive additional modality, *e.g.,* cardiac MRI, $\mathrm{s}_i^{\mathrm{avail}}$ would be the score from the blood tests alone, while $\mathrm{s}_i^{\mathrm{acquired}}$ would be the score using both tests and MRI. To compute these scores, we assume a single model $f$ parameterized by $\boldsymbol{\theta}$ that can process any subset of modalities. The available and acquired scores are thus:

$$\mathrm{s}_i^{\mathrm{avail}} = f(\boldsymbol{x}_i^{\mathrm{avail}}, \boldsymbol{\theta}) \tag{1}$$

$$\mathrm{s}_i^{\mathrm{acquired}} = f(\boldsymbol{x}_i^{\mathrm{acquired}}, \boldsymbol{\theta}) \tag{2}$$

where $\boldsymbol{x}_i^{\mathrm{avail}}$ and $\boldsymbol{x}_i^{\mathrm{acquired}}$ represent the feature sets for the available and acquired modalities, respectively. To estimate the acquired score without costly acquisition, we use a generative imputation model $f_{\mathrm{imp}}$. This model generates a set of $K$ plausible embeddings that enable the classifier $f_C$ to predict the scores $\{\mathrm{s}_{i,k}^{\mathrm{imp}}\}_{k=1}^K$. These imputation-based scores form the basis of our acquisition functions.

The goal of the optimization is to select a subset of samples $\mathcal{S}$ from the cohort of $N$ total samples for which an additional modality should be acquired. This subset $\mathcal{S} \subseteq \{1, \ldots, N\}$ has a predetermined size $|\mathcal{S}| = \beta$, where $\beta$ is the acquisition budget, *i.e.,* the number of samples for which additional modalities will be acquired. The final score $\mathrm{s}_i(\mathcal{S})$ used for the evaluation of a sample $i$ is then

determined by the selection:

$$\mathrm{s}_i(\mathcal{S}) = \begin{cases} \mathrm{s}_i^{\mathrm{acquired}} & \text{if } i \in \mathcal{S} \\ \mathrm{s}_i^{\mathrm{avail}} & \text{if } i \notin \mathcal{S} \end{cases} . \tag{3}$$

The optimization problem is to find the set $\mathcal{S}^*$ that maximizes the chosen performance metric:

$$\mathcal{S}^* = \underset{\mathcal{S} \subseteq \{1,...,N\}:|\mathcal{S}|=\beta}{\arg\max} \mathrm{Metric}(\mathbf{y}, \mathbf{s}(\mathcal{S})) \tag{4}$$

where $\mathbf{y} = \{\mathrm{y}_i\}_{i=1}^{N}$ is the vector of true labels, and $\mathbf{s}(\mathcal{S}) = \{\mathrm{s}_i(\mathcal{S})\}_{i=1}^{N}$ is the vector of resulting scores for all samples in the cohort. Consequently, the task is to identify an optimal, constrained subset for which to acquire additional modalities, while maximizing a performance metric across the entire cohort.

# 4 ACQUISITION FUNCTION STRATEGIES

Directly solving the cohort-level optimization problem to identify the optimal sample set $\mathcal{S}^*$ is computationally intractable due to its combinatorial nature. Therefore, we employ several heuristic acquisition functions (AFs) that approximate the optimal selection by ranking samples for modality acquisition. These strategies, detailed further in Section C, are derived from standard discriminative metrics (Section C.1) and can be categorized as follows (Table 2):

- **Oracle Strategies:** As upper-bound benchmarks, they assume perfect knowledge of outcomes and true labels to greedily select samples yielding the largest immediate gain in the target metric.

- **Upper-Bound Heuristic Strategies:** These heuristics assume knowledge of scores under modality completion but are label-agnostic, relying on metrics like the true uncertainty reduction, rank change, or KL-Divergence.

- **Imputation-Based Strategies:** Grounded in counterfactual reasoning, these strategies use a generative model to predict how a sample's score might change if a missing modality were acquired.

- **Baseline Information Strategies:** These strategies make decisions using only information from the initially available modalities, *i.e.,* without any imputation and pre-acquisition, such as its predicted uncertainty or probability.

- **Random Strategy:** This serves as a fundamental baseline by selecting samples randomly, without regard to any model scores.

Table 2: Summary of AF Strategies.

| Category | Strategies | Input Variables | Ranking Criteria |
|---|---|---|---|
| Oracle | AUROC, AUPRC | True labels & acquired scores | Greedy selection for maximum gain. |
| Upper-Bound | KL-Divergence, Rank, Uncertainty | True acquired scores | True change in prediction, cohort rank, or uncertainty. |
| Imputation-Based | KL-Divergence, Rank, Uncertainty & Probability | Imputed acquired scores | Expected change in prediction, rank, or uncertainty. |
| Baselines | Uncertainty, Probability | Pre-acquisition scores | Uncertainty or probability using the available modality. |
| Random | Random | None | Random selection. |

Intuitively, these acquisition functions approximate the expected information gain (EIG) from acquiring an additional modality. In our setting, EIG quantifies the expected improvement in a chosen

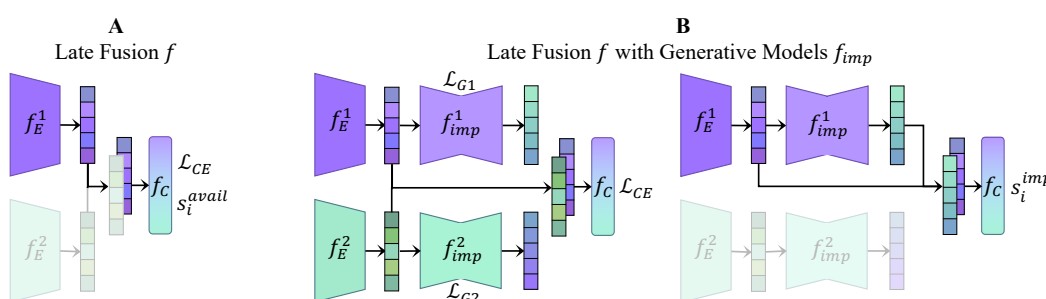

Figure 2: End-to-end architectures to determine the scores for different AFs in our proposed CAMA setting. **(A)** Vanilla late fusion (LF) architecture of a model $f$ that can handle missing data modalities by masking. The model creates scores $s_i^{\text{avail}}$ given the available modalities. **(B)** Architecture for training (left) and inference (right) with a late fusion (LF) model $f$ and a generative model $f_{imp}$ to create scores $s_i^{\text{imp}}$ for the imputation-based AFs.

performance metric given the additional information that would become available through a new modality.

For evaluation, we introduce a metric that describes the cumulative performance of an AF, normalized by the total possible gain achievable by transitioning all samples to post-acquisition performance (see Figure 1 C, for an illustrative curve). Let $M_{\text{AF}}(b)$ denote the performance curve of an AF strategy for a primary metric $M$ as a function of the budget fraction $b$ of the acquisition budget $\beta$, $M_{\text{pre}}$ the performance of the pre-acquisition baseline, and $M_{\text{post}}$ the performance of the post-acquisition model. The normalized area of gain for an acquisition function AF, which measures the portion of achievable performance gain captured across different budgets, is defined as the area under the performance gain curve, normalized by the maximum possible gain (Equation (5)). Intuitively, a value of 0 indicates no improvement over the pre-acquisition baseline across budgets. A value of 1 indicates matching the post-acquisition performance on average across budgets. Values greater than 1 occur when the cohort's performance at intermediate budgets temporarily exceeds the post-acquisition cohort as detailed later and shown in Figure 3.

$$G_{\text{full}}^{M}(\text{AF}) = \frac{\int_0^1 (M_{\text{AF}}(b) - M_{\text{pre}})\, db}{M_{\text{post}} - M_{\text{pre}}} \tag{5}$$

## 5 EVALUATION

To evaluate these strategies in practice, we require architectures that produce the necessary scores. The oracle, upper-bound, baseline, and random AFs can be evaluated using a vanilla discriminative late fusion (LF) model (Figure 2 A), as they operate on true labels $y_i$ and true scores $s_i^{\text{acquired}}$ and $s_i^{\text{avail}}$ (Section 3). In contrast, our proposed imputation-based AFs are grounded in counterfactual reasoning: They require the model to predict how its output would change if a missing modality were present. This necessitates a more sophisticated architecture that combines the discriminative classifier with a generative component capable of imputing the missing modality (Figure 2 B).

**Model Architecture** First, we implement a multimodal architecture consisting of modality-specific encoders $f_E^{(m)} : \mathcal{X}^{(m)} \to \mathbb{R}^d$ and a fusion classifier $f_C$ (Figure 2 A). The encoders map raw inputs for a sample $i$ and modality $m$ to latent embeddings $z_i^{(m)} = f_E^{(m)}(x_i^{(m)})$, *e.g.*, with a Vision Transformer (ViT) (Dosovitskiy et al., 2021) for images and BERT (Devlin et al., 2019) for text. The discriminative classifier $f_C$ aggregates these embeddings to produce logits $s_i^{\text{avail}}$ depending on the availability of the raw inputs, *i.e.*, with a Transformer encoder (Vaswani et al., 2017). Second, for our generative AFs, we incorporate generative modules additionally to the discriminative late fusion (Figure 2 B) (Wang et al., 2023). The generative modules $f_{\text{imp}}$ are parameterized as Diffusion Transformers (DiTs) (Peebles & Xie, 2023) or Beta-Conditional-VAEs (BC-VAEs) (Higgins et al., 2017) trading off

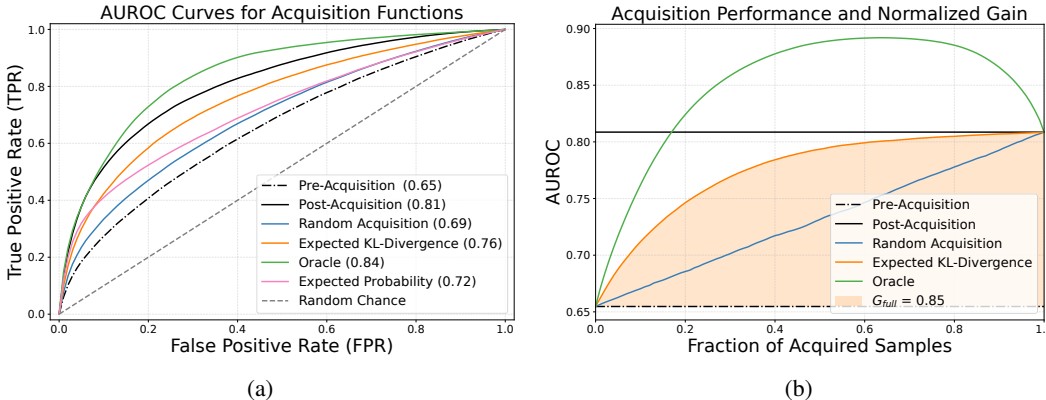

(a)                    (b)

Figure 3: **(a)** AUROC curves for several AFs on the MOSEI dataset (Zadeh et al., 2018) at an acquisition budget of $25\%$ of the dataset size. **(b)** Acquisition performance of the best-performing AF from (a), visualizing the gain achieved during the progressive acquisition of modalities as the cohort transitions from pre-acquisition scores towards post-acquisition. Notably, the oracle AF can exceed the post-acquisition cohort's AUROC at certain fractions of acquired modalities before subsequently declining towards it again.

performance vs. efficiency. The generative modules $f_{\text{imp}}$ are trained to approximate the conditional distribution $p_\theta(z^{(k)}|\{z^{(m)}\}_{m\in\mathcal{P}_i^{\text{avail}}})$ for each target modality $k$. The generative loss for sample $i$ is:

$$\mathcal{L}_{G_i} = \sum_{k\in\mathcal{P}_i^{\text{avail}}} \mathcal{L}_{\text{gen}}\left(z_i^{(k)}; \{z_i^{(m)}\}_{m\in\mathcal{P}_i^{\text{avail}}}\right) \tag{6}$$

where $\mathcal{L}_{\text{gen}}$ is a variational bound on the negative conditional log-likelihood: for DDPMs, this corresponds to the denoising objective (Ho et al., 2020); for BC-VAEs, this is the negative conditional evidence lower bound (ELBO) (Higgins et al., 2017). For the discriminative task, the loss for sample $i$ is defined as binary cross entropy loss with the label $y$ and the predicted probability $p$:

$$\mathcal{L}_{\text{CE}_i} = -\left[y_i \log(p_i) + (1-y_i)\log(1-p_i)\right]. \tag{7}$$

The final loss function for the whole architecture is defined as the combination of both loss terms:

$$\mathcal{L} = \lambda_1 \mathcal{L}_{\text{CE}} + \lambda_2 \mathcal{L}_{\text{G}} \tag{8}$$

with loss weightings $\lambda$ for which we find $\lambda_1 = \lambda_2 = 1$ is important for downstream performance. During inference, the classifier $f_C$ also uses samples of $p(z^{(k)}|\{z^{(m)}\}_{m\in\mathcal{P}_i^{\text{avail}}})$ for missing modalities to create the scores $s_i^{\text{imp}}$ needed for our generative AFs (Figure 2 B, right). During training, samples from $p(z^{(k)}|\{z^{(m)}\}_{m\in\mathcal{P}_i^{\text{avail}}})$ are not passed to $f_C$, even when modalities are missing (Figure 2 B, left). Instead, the discriminative components ($f_E^{(m)}$ and $f_C$) are trained only on available modalities via attention masks. This means that the generative and discriminative parts are trained jointly, but the generative outputs do not directly influence the classifier during training beyond the shared encoders being updated by the classification loss. For model training, we use the ScheduleFree optimizer (Defazio et al., 2024) with hyperparameters determined through sweeps. We find the following architectural decisions essential, which are ablated in Table 7: (a) applying Layer Normalization (Ba et al., 2016) at the end of each modality's encoder to stabilize the DDPMs operating between latent spaces, (b) calibrating the model with label smoothing (Szegedy et al., 2016) to produce less overconfident and better-calibrated probability distributions, (c) decoupling the generative modules from the classifier during training and (d) class balancing the training dataset as detailed in the next paragraph. Regarding missing modalities, we do not pre-train on all available data modalities, in contrast to Wang et al. (2023). We use a predefined, seed-dependent missing-modality mask to control data modality leakage during training unlike batch-dependent masks, which eventually reveal all modalities for every sample across numerous epochs. Further details in Section D.

**Datasets** We evaluate the setting of CAMA on four real-world multimodal datasets: UKBB (Sudlow et al., 2015), MIMIC Symile (Saporta et al., 2024), MIMIC HAIM (Soenksen et al., 2022a;b), and

Table 3: Acquisition performance on Symile, with $G_{\text{full}}$ shown for AUROC/AUPRC as an example for the class with the best and worst performance and the mean value of all ten classes. Strategies are grouped by category. Best strategy among proposed ones and baselines in bold for each column.

| | Acquisitions by AUROC, $G_{\text{full}} \uparrow \pm$ SEM | | | Acquisitions by AUPRC, $G_{\text{full}} \uparrow \pm$ SEM | | |
|---|---|---|---|---|---|---|
| Strategy | Cardiomegaly | Pneumothorax | Mean | Lung Lesion | Pneumothorax | Mean |
| *Upper Bounds (for reference)* | | | | | | |
| Oracle | $2.787 \pm 0.139$ | $9.461 \pm 1.049$ | 4.580 | $2.520 \pm 0.250$ | $10.623 \pm 0.708$ | 4.231 |
| True KL-Div. | $0.885 \pm 0.011$ | $0.910 \pm 0.054$ | 0.883 | $0.828 \pm 0.073$ | $0.827 \pm 0.043$ | 0.871 |
| True Rank | $0.878 \pm 0.019$ | $0.605 \pm 0.053$ | 0.811 | $0.676 \pm 0.088$ | $0.483 \pm 0.075$ | 0.776 |
| True Uncert. | $0.524 \pm 0.025$ | $-0.136 \pm 0.065$ | 0.481 | $0.181 \pm 0.067$ | $0.293 \pm 0.052$ | 0.450 |
| *Imputation-based (proposed)* | | | | | | |
| KL-Divergence | $\mathbf{0.747 \pm 0.039}$ | $0.773 \pm 0.134$ | **0.833** | $\mathbf{0.896 \pm 0.146}$ | $0.581 \pm 0.084$ | **0.777** |
| Probability | $0.350 \pm 0.053$ | $\mathbf{0.898 \pm 0.061}$ | 0.426 | $0.320 \pm 0.104$ | $\mathbf{0.965 \pm 0.027}$ | 0.449 |
| Rank | $0.378 \pm 0.016$ | $0.115 \pm 0.082$ | 0.378 | $0.564 \pm 0.086$ | $0.396 \pm 0.054$ | 0.407 |
| Uncertainty | $0.450 \pm 0.041$ | $0.055 \pm 0.060$ | 0.440 | $0.130 \pm 0.053$ | $0.513 \pm 0.066$ | 0.444 |
| *Baselines (no imputation)* | | | | | | |
| Uncertainty | $0.480 \pm 0.013$ | $0.536 \pm 0.040$ | 0.480 | $0.215 \pm 0.033$ | $0.811 \pm 0.041$ | 0.443 |
| Probability | $0.431 \pm 0.015$ | $0.536 \pm 0.040$ | 0.458 | $0.756 \pm 0.136$ | $0.811 \pm 0.041$ | 0.550 |
| Random | $0.385 \pm 0.015$ | $0.327 \pm 0.061$ | 0.376 | $0.503 \pm 0.103$ | $0.527 \pm 0.053$ | 0.388 |

MOSEI (Zadeh et al., 2018), which cover diverse domains such as healthcare and emotion recognition. For the publicly available datasets, missing modalities are synthetically created, whereas for UKBB they are an inherent characteristic. We design the datasets for binary classification, resulting in ten binary targets for the MIMIC datasets and one binary target for MOSEI and UKBB. While MOSEI is already class-balanced (Zadeh et al., 2018), HAIM and Symile exhibit significant class imbalance (Soenksen et al., 2022a; Zadeh et al., 2018). To address this, we employ random oversampling during training, which we find essential for the effective operation of AFs (Table 7). Importantly, during testing we retain the original imbalanced distributions, and no class-balancing steps are applied to UKBB. We highlight UKBB as the most challenging dataset to demonstrate that CAMA scales to a broad multimodal range and large-scale cohorts with approximately 100,000 samples and 15 modalities. In this setting, we focus on acquiring the exceptionally costly proteomics data for predicting the onset of systemic lupus erythematosus (SLE), which has been shown to benefit from proteomics combined with other clinical data (Yang et al., 2025). Additional details are provided in Section E.

**Model and AF Evaluation** For datasets with at least three modalities, we apply five-fold cross-validation. Due to initially noisy results for MIMIC HAIM, we increase the number of folds to ten. For each sample in the test set, the initial score $s_i^{\text{avail}}$ is established by randomly assigning a subset of available modalities $\mathcal{P}_i^{\text{avail}}$. This procedure is repeated over several runs for robustness. In each run, every sample is stochastically assigned a new subset $\mathcal{P}_i^{\text{avail}}$. Performance metrics are averaged across these independent runs to ensure our evaluation is robust to any single random assignment of patient data. Acquisition is simulated by incrementally increasing the budget $\beta$. We focus on tasks where the post-acquisition model demonstrates a performance improvement over the pre-acquisition baseline. For certain prediction tasks, a simpler pre-acquisition model can outperform a more complex post-acquisition one, potentially due to the introduction of noisy or conflicting signals. In such cases, the final post-acquisition performance falls below the pre-acquisition baseline, resulting in a negative normalized area of gain, indicating that acquisition was detrimental. To ensure a meaningful evaluation, we exclude any tasks exhibiting this negative gain from the analysis at the split level. For each budget, the top-ranked samples in $\mathcal{S}$ are considered acquired, and their logits are updated from $s_i^{\text{avail}}$ to $s_i^{\text{acquired}}$. Final reported results are aggregated across all cross-validation splits, combinations of missing and available modalities, and random runs to ensure robustness of the evaluation.

## 6 RESULTS

Our empirical evaluation confirms the effectiveness of CAMA. We benchmark our imputation-based strategies against oracles, upper-bound heuristics, and baselines across multiple datasets. Full results are aggregated in Tables 3 to 5 by averaging over permutations of missing input modali-

Table 4: Acquisition performance on UKBB, showing $G_{\text{full}}$ for AUROC/AUPRC. Strategies are grouped by category. Best strategy among proposed ones and baselines in bold.

| | AUROC | AUPRC |
|---|---|---|
| **Strategy** | $G_{\text{full}} \pm$ SEM $\uparrow$ | $G_{\text{full}} \pm$ SEM $\uparrow$ |
| *Upper Bounds (for reference)* | | |
| Oracle | $1.141 \pm 0.051$ | $1.721 \pm 0.315$ |
| True KL-Div. | $0.978 \pm 0.007$ | $0.986 \pm 0.005$ |
| True Rank | $0.887 \pm 0.022$ | $0.466 \pm 0.110$ |
| True Uncert. | $0.436 \pm 0.088$ | $0.507 \pm 0.074$ |
| *Imputation-based (proposed)* | | |
| KL-Divergence | $\mathbf{0.641 \pm 0.029}$ | $0.658 \pm 0.045$ |
| Probability | $0.535 \pm 0.026$ | $\mathbf{0.713 \pm 0.029}$ |
| Rank | $0.437 \pm 0.028$ | $0.340 \pm 0.114$ |
| Uncertainty | $0.373 \pm 0.053$ | $0.332 \pm 0.058$ |
| *Baselines (no imputation)* | | |
| Uncertainty | $0.365 \pm 0.042$ | $0.556 \pm 0.073$ |
| Probability | $0.365 \pm 0.042$ | $0.556 \pm 0.073$ |
| Random | $0.528 \pm 0.018$ | $0.485 \pm 0.052$ |

Table 5: Acquisition performance on MOSEI, showing $G_{\text{full}}$ for AUROC/AUPRC. Strategies are grouped by category. Best strategy among proposed ones and baselines in bold.

| | AUROC | AUPRC |
|---|---|---|
| **Strategy** | $G_{\text{full}} \pm$ SEM $\uparrow$ | $G_{\text{full}} \pm$ SEM $\uparrow$ |
| *Upper Bounds (for reference)* | | |
| Oracle | $1.478 \pm 0.091$ | $1.666 \pm 0.161$ |
| True KL-Div. | $0.882 \pm 0.006$ | $0.838 \pm 0.006$ |
| True Rank | $0.849 \pm 0.008$ | $0.806 \pm 0.010$ |
| True Uncert. | $0.663 \pm 0.006$ | $0.708 \pm 0.005$ |
| *Imputation-based (proposed)* | | |
| KL-Divergence | $\mathbf{0.855 \pm 0.034}$ | $\mathbf{0.889 \pm 0.052}$ |
| Probability | $0.707 \pm 0.037$ | $0.846 \pm 0.070$ |
| Rank | $0.432 \pm 0.014$ | $0.457 \pm 0.019$ |
| Uncertainty | $0.630 \pm 0.015$ | $0.706 \pm 0.037$ |
| *Baselines (no imputation)* | | |
| Uncertainty | $0.525 \pm 0.005$ | $0.540 \pm 0.006$ |
| Probability | $0.433 \pm 0.007$ | $0.543 \pm 0.009$ |
| Random | $0.490 \pm 0.004$ | $0.525 \pm 0.003$ |

Table 6: Efficiency analysis for different architectures.

| Architecture | Train (sec) $\downarrow$ | Validation (sec) $\downarrow$ | Parameters (M) $\downarrow$ |
|---|---|---|---|
| Late fusion | 0.02 | 0.015 | 86.5 |
| Late fusion w/ DDPMs | 0.17 | 0.16 | 125 |
| Late fusion w/ BC-VAEs | 0.08 | 0.08 | 313 |

ties and multiple random instantiations for each missingness configuration. As expected, oracle strategies serve as an upper bound and consistently achieve the highest performance. Surprisingly, oracle gains can exceed the value of one, as a strategic mix of pre-acquisition and post-acquisition samples can outperform a purely post-acquisition cohort. To benchmark the acquisition logic itself, we use label-agnostic upper-bound heuristics that access acquired scores $s_i^{\text{acquired}}$. Among these, strategies based on KL-Divergence and rank change perform well, indicating that prioritizing large predictive shifts or cohort reordering is an effective heuristic in this setting. Our main approach for handling the CAMA setting comprises imputation-based strategies that leverage a generative model $f_{\text{imp}}$ to predict counterfactual outcomes. The imputation-based KL-Divergence strategy consistently and significantly outperforms all other non-oracle methods. This AF effectively identifies samples predicted to have the largest shift in their class probability distribution (Figure 3).

In contrast, imputation-based strategies relying on rank change, final uncertainty, or final probability are considerably weaker, suggesting that quantifying the change in prediction is more effective than estimating the final state. While our primary results with respect to imputation-based AFs use DDPMs, a BC-VAE variant offers significantly faster inference for a minor trade-off in performance (Table 6 and section F). The relative performance ranking of these strategies is largely consistent across all datasets, including the large-scale UKBB cohort with approximately 100,000 samples and 15 modalities. This confirms the robustness and scalability of our framework in a challenging setting. In summary,

Table 7: Cross-validated ablation of the proposed model adjustments on the Symile dataset, exemplary for the mean across all endpoints with the expected KL-Divergence and acquisitions by AUROC.

| Ablation | $\mathbf{G_{\text{full}}} \uparrow$ |
|---|---|
| KL-Divergence (w.r.t. Table 3) | 0.833 |
| *w/o* Layer Norm | 0.772 |
| *w/o* label smoothing | 0.746 |
| *w/o* decoupled data flow | 0.599 |
| *w/o* balanced train set | 0.568 |

our results affirm the superiority of the imputation-based KL-Divergence strategy, which achieved substantial and reliable gains over all baselines and heuristic methods. Additional results in Sections G to I.

## 7 DISCUSSION

We introduce CAMA to address the challenge of strategic data acquisition under budget constraints. Our experiments consistently demonstrate that imputation-based AFs provide a robust and effective solution. In the following, we discuss the key implications. The ability of oracles to yield gains exceeding that of a model using post-acquisition data for all samples (Figure 3 (b)), suggests that an underlying predictive model can achieve better global performance with a strategic curation of samples, rather than applying all modalities across the cohort. This likely occurs because additional modalities may introduce variance, redundancy, or conflicting information that imperfect models cannot optimally reconcile. The oracles circumvent this by selecting only additional modalities beneficial to the global metric. To our surprise, the imputation-based KL-Divergence AF can slightly outperform the corresponding upper-bound heuristic (Table 5). Conversely, the substantial performance gap between the rank-change heuristic and its imputation-based counterpart suggests that global, rank-based metrics may be particularly vulnerable to imputation noise. While the KL-Divergence AF demonstrated strong performance, not all imputation-based AFs consistently outperformed simpler strategies across all datasets or endpoints (Sections H and I). This indicates that optimal CAMA AFs can be context-dependent and that effectiveness hinges on how imputations are leveraged rather than on imputation quality alone. Regarding the impact of imputation quality, it is important to note that we impute latent embeddings optimized for the discriminative task rather than raw data. Consequently, standard generative metrics (like Fréchet inception distance (FID)) are not applicable for comparing imputation quality across different generative models since every generative model influences the encoders latent spaces indirectly. While we observe that utilizing stronger generative models, *e.g.,* DDPMs, results in higher acquisition performance compared to weaker models, *e.g.,* VAEs, our findings indicate that the generative imputation quality is not the only factor. As detailed in Table 7, the coherence of the overall architecture design, *i.e.,* ensuring the classifier is robust to the distribution of imputed latents, is equally critical for effective acquisition. We show CAMAs robustness to imputation errors since $f_{\text{imp}}$ models a distribution of plausible outcomes rather than aiming for a single reconstruction. By averaging the expected impact across this distribution, the acquisition decision becomes less sensitive to uncertainty. Additionally, the primary KL-Divergence AF is resilient to noise, as it prioritizes samples expected to cause a large predictive shift, effectively ignoring minor imputation errors. Taken together, CAMA is not only practical for constrained settings, but also reveals insights into post-acquisition behavior. The successful KL-Divergence strategy and the surprising oracle performance underscore that the value of an additional modality is not absolute but highly contextual. The most effective AFs are not those that simply predict an outcome, but estimate the magnitude of the predictive shift.

## 8 CONCLUSION AND FUTURE WORK

We introduce CAMA, a novel setting addressing the real-world challenge of optimizing global discriminative performance through strategic test-time acquisition of an additional modality under resource constraints. Our evaluation across multiple multimodal datasets shows that imputation-based AFs can effectively guide resource allocation under cohort-level constraints. The generally consistent relative ordering of AFs across diverse datasets and the low variance in overall results lend confidence to the robustness of our core findings. In settings such as healthcare, strategic allocation of costly or invasive diagnostic procedures is essential, and our approach offers a promising direction for these applications. Future work includes extending CAMA to multi-class problems or regression tasks, exploring additional imputation techniques, directly optimizing cohort-level metrics, and dynamically selecting which modality to acquire instead of pre-selecting one.

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

## A  BROADER IMPACT AND ETHICS

The CAMA setting introduced in this paper offers potential for positive broader impacts, primarily by enabling more efficient use of resources in multimodal machine learning. In resource-constrained fields like healthcare, this could facilitate access to more robust and comprehensive model performance by strategically guiding the acquisition of costly or limited additional data modalities. This could translate to improved diagnostic accuracy where such data is critical but not uniformly available for all samples in a cohort. However, the deployment of CAMA, particularly its core function of ranking and prioritizing samples for modality acquisition, necessitates careful ethical consideration. This raises concerns about equity and fairness, especially if the downstream application impacts critical decisions. A significant risk is the potential to introduce biases, including racial, socioeconomic, or other demographic biases. Therefore, the development and application of CAMA must be approached with a strong commitment to ethical principles.

## B  REPRODUCIBILITY

To ensure the reproducibility of our results, we provide the following details:

**Code**  The complete source code used for all experiments will be made publicly available on GitHub upon publication. The repository will include scripts for model training and evaluation.

**Hyperparameters**  All hyperparameters, including learning rates, batch sizes, and model-specific parameters, are explicitly listed in Section D. Additionally, we provide the complete sweep configurations used for hyperparameter tuning to allow for full replication of our optimization process.

**Datasets**  Three of the four datasets used in our evaluation are publicly available. For more details see Section 5 and Section E.

**Implementation Details**  We provide a full section in Section D and a dedicated paragraph in Section 5 describing implementation details that we found to be crucial.

## C  DETAILS ABOUT ACQUISITION FUNCTION STRATEGIES

### C.1  AUROC AND AUPRC

To derive the proposed acquisition strategies, we briefly explain the metrics used in the following paragraphs.

**AUROC**  The Area Under the Receiver Operating Characteristic (AUROC) measures the model's ability to discriminate between positive and negative classes and is defined as

$$\text{AUROC}(\mathbf{y}, \mathbf{s}) = \frac{1}{N_+ N_-} \sum_{i: y_i = 1} \sum_{j: y_j = 0} \left( \mathbb{I}(s_i > s_j) + \frac{1}{2} \mathbb{I}(s_i = s_j) \right) \tag{9}$$

where $N_+ = |\{i \mid y_i = 1\}|$ and $N_- = |\{j \mid y_j = 0\}|$.

**AUPRC**  The Area Under the Precision-Recall Curve (AUPRC) summarizes the trade-off between precision ($P_t$) and recall ($R_t$) across different decision thresholds $t$ and is defined as

$$\text{AUPRC}(\mathbf{y}, \mathbf{p}) = \sum_{k=1}^{N'} (R_k - R_{k-1}) P_k \tag{10}$$

where points $(R_k, P_k)$ are ordered by threshold from the PR curve, $N'$ is the number of unique thresholds, and $\mathbf{p} = \sigma(\mathbf{s})$.

## C.2 ORACLE ACQUISITION STRATEGIES: EXACT GAIN CALCULATION

Oracle acquisition strategies serve as theoretical upper limits for the performance of greedy acquisition approaches. They operate under the ideal assumption that the true labels $y_i$ and the outcome scores $s_i^{\text{acquired}}$ are known for all samples $i \in \{1, \ldots, N\}$. While not implementable in practice, these oracle strategies provide benchmarks by selecting samples based on their exact marginal contribution to the global evaluation metric. The general principle is to iteratively select $\beta$ samples. At each step, among the samples for which the additional modality has not yet been acquired, the oracle picks the one that provides the largest true immediate gain to the chosen global metric.

**AUROC Oracle**  The AUROC oracle strategy aims to maximize the cohort's AUROC by identifying, at each step, the sample $i$ that yields the largest immediate increase in this metric if its additional modality were acquired (changing its score from $s_i^{\text{avail}}$ to $s_i^{\text{acquired}}$), *i.e.*, a greedy selection. This prospective increase is quantified by the marginal gain $g_i^{\text{AUROC}}$. The components of this gain, $g_i^{\text{AUROC}}(y_i = 1)$ (for positive samples) and $g_i^{\text{AUROC}}(y_i = 0)$ (for negative samples), reflect the net change in favorable pairwise score comparisons relative to samples of the other class. Recall the definition of AUROC from Equation (9):

$$\text{AUROC}(\mathbf{y}, \mathbf{s}) = \frac{1}{N_+ N_-} \sum_{i:y_i=1} \sum_{j:y_j=0} \left( \mathbb{I}(s_i > s_j) + \frac{1}{2}\mathbb{I}(s_i = s_j) \right).$$

The total marginal gain for sample $i$, representing the exact change in the cohort's AUROC value, is then, by considering positive and negative samples and neglecting the normalization factor:

$$
\begin{aligned}
g_i^{\text{AUROC}}(y_i = 1) = \sum_{j:y_j=0} \bigg( & \mathbb{I}(s_i^{\text{acquired}} > s_j^{\text{avail}}) - \mathbb{I}(s_i^{\text{avail}} > s_j^{\text{avail}}) \\
& + \frac{1}{2}\Big[ \mathbb{I}(s_i^{\text{acquired}} = s_j^{\text{avail}}) - \mathbb{I}(s_i^{\text{avail}} = s_j^{\text{avail}}) \Big] \bigg)
\end{aligned}
\tag{11}
$$

$$
\begin{aligned}
g_i^{\text{AUROC}}(y_i = 0) = \sum_{j:y_j=1} \bigg( & \mathbb{I}(s_j^{\text{avail}} > s_i^{\text{acquired}}) - \mathbb{I}(s_j^{\text{avail}} > s_i^{\text{avail}}) \\
& + \frac{1}{2}\Big[ \mathbb{I}(s_j^{\text{avail}} = s_i^{\text{acquired}}) - \mathbb{I}(s_j^{\text{avail}} = s_i^{\text{avail}}) \Big] \bigg)
\end{aligned}
\tag{12}
$$

$$g_i^{\text{AUROC}} = \frac{1}{N_+ N_-} \left( g_i^{\text{AUROC}}(y_i = 1) \cdot \mathbb{I}(y_i = 1) + g_i^{\text{AUROC}}(y_i = 0) \cdot \mathbb{I}(y_i = 0) \right) \tag{13}$$

**AUPRC Oracle**  The AUPRC oracle strategy seeks to maximize the cohort's AUPRC. It operates by identifying, at each step, the sample $i$ which, if its additional modality were acquired (changing its score from $s_i^{\text{avail}}$ to $s_i^{\text{acquired}}$), would yield the largest immediate increase in the global AUPRC value, *i.e.*, a greedy selection. This marginal gain, $g_i^{\text{AUPRC}}$, represents the exact change in the cohort's AUPRC. To calculate the marginal gain for a sample $i$, we compute the change in the cohort's AUPRC. Let $\mathbf{s}^{\text{current}}$ be the vector of scores for the whole cohort. We define a new vector, $\mathbf{s}^{\text{updated}}$, which is identical to $\mathbf{s}^{\text{current}}$ except that for sample i, the score is changed from $s_i^{\text{avail}}$ to $s_i^{\text{acquired}}$. The marginal gain is then:

$$g_i^{\text{AUPRC}} = \text{AUPRC}(\mathbf{y}, \mathbf{p}^{\text{updated}}) - \text{AUPRC}(\mathbf{y}, \mathbf{p}^{\text{current}}) \tag{14}$$

where $\mathbf{p}^{\text{current}} = \sigma(\mathbf{s}^{\text{current}})$ and $\mathbf{p}^{\text{updated}} = \sigma(\mathbf{s}^{\text{updated}})$.

## C.3 Upper-Bound Heuristic Strategies

The preceding oracle strategies make the assumption of perfect foresight into both the true labels $y_i$ and the exact outcome scores $s_i^{\text{acquired}}$. We now introduce a distinct class of upper-bound heuristic strategies. These strategies still presume access to the true future scores $s_i^{\text{acquired}}$ for any sample $i$ if its additional modality were acquired. However, the following upper-bound heuristics are label-agnostic, *i.e.,* the true label $y_i$ of a candidate sample is not used when determining its priority for acquisition. Consequently, the selection principle for these strategies must rely on how the known change from an initial score $s_i^{\text{avail}}$ to the future score $s_i^{\text{acquired}}$ is expected to influence the global evaluation metric, without direct reference to the sample's ground-truth label.

**Maximum True Uncertainty Reduction** The uncertainty reduction strategy prioritizes acquiring the additional modality for samples where doing so is expected to yield the largest decrease in predictive uncertainty. For each sample $i$, uncertainty is quantified using the binary entropy $\mathcal{H}(p_i)$ of its predicted probability $p_i$ for the positive class, defined as:

$$\mathcal{H}(p_i) = -p_i \log_2 p_i - (1 - p_i) \log_2(1 - p_i), \tag{15}$$

The acquisition strategy operates with knowledge of the initial probability $p_i^{\text{avail}} = \sigma(s_i^{\text{avail}})$ derived from the available modalities, and crucially, the true future probability $p_i^{\text{acquired}} = \sigma(s_i^{\text{acquired}})$ that would be obtained if the additional modality were acquired (where $s_i^{\text{acquired}}$ is the oracle score). The acquisition score $g_i^{\text{UR}}$ for sample $i$ is then the exact reduction in entropy:

$$g_i^{\text{UR}} = \mathcal{H}(p_i^{\text{avail}}) - \mathcal{H}(p_i^{\text{acquired}}). \tag{16}$$

Samples with higher $g_i^{\text{UR}}$ values, indicating a greater expected reduction in uncertainty, are prioritized for modality acquisition.

**Maximum True Rank Change** This rank change strategy prioritizes samples whose relative standing within the cohort, based on predicted probability of belonging to the positive class, would change most significantly if the additional modality were acquired. For each sample $i$, we consider its rank $R(p_i)$ when all $N$ samples in the cohort are ordered by their respective probabilities $p_i$. The acquisition score $g_i^{\text{RC}}$ for sample $i$ is defined as the absolute magnitude of this change in rank:

$$g_i^{\text{RC}} = |R(p_i^{\text{acquired}}) - R(p_i^{\text{avail}})|. \tag{17}$$

Samples exhibiting a higher $g_i^{\text{RC}}$ are prioritized for modality acquisition, since they are expected to cause the largest shift in the sample's rank-ordered position relative to its peers.

**KL-Divergence** The KL-Divergence acquisition strategy aims to identify samples for which acquiring the additional modality would lead to the largest change in the predicted probability distribution. Specifically, it quantifies the divergence from the predicted probability distribution based on the true future score, $P_i^{\text{acquired}} \sim \text{Bernoulli}(p_i^{\text{acquired}})$, back to the initial distribution based on baseline data, $P_i^{\text{avail}} \sim \text{Bernoulli}(p_i^{\text{avail}})$. This is measured by the KL-Divergence $D_{\text{KL}}(P_i^{\text{avail}} \| P_i^{\text{acquired}})$ and can be defined as follows for an acquisition function:

$$g_i^{\text{KLD}} = D_{\text{KL}}\left(P_i^{\text{avail}} \middle\| P_i^{\text{acquired}}\right) \tag{18}$$

$$= p_i^{\text{avail}} \log_2 \frac{p_i^{\text{avail}}}{p_i^{\text{acquired}}} + (1 - p_i^{\text{avail}}) \log_2 \frac{1 - p_i^{\text{avail}}}{1 - p_i^{\text{acquired}}} \tag{19}$$

Samples with a higher $g_i^{\text{KLD}}$ are prioritized, as this indicates a greater discrepancy between the prediction based on available data and the prediction that would be made with the additional modality.

## C.4 BASELINE INFORMATION STRATEGIES

Shifting from approaches that leverage oracle knowledge of future scores ($s_i^{\text{acquired}}$), the present section details methods serving as practical, label-agnostic baselines. They make acquisition decisions based exclusively on information derived from the initially available modality ($s_i^{\text{avail}}$). A random acquisition strategy serves as a fundamental baseline.

**Maximum Baseline Uncertainty** The Maximum Baseline Uncertainty strategy is a baseline that prioritizes samples for which the prediction based on the initially available modality is most uncertain. The acquisition score for sample $i$ is directly the binary entropy $\mathcal{H}(p_i^{\text{avail}})$, as defined in Equation (15):

$$g_i^{\text{UU}} = \mathcal{H}(p_i^{\text{avail}}). \tag{20}$$

Samples with a higher $g_i^{\text{UU}}$, *i.e.,* $p_i^{\text{avail}}$ closer to 0.5, since the entropy $H(p_i^{\text{avail}})$ is symmetric around $p_i^{\text{avail}} = 0.5$, are selected first.

**Maximum Baseline Probability** This approach prioritizes acquiring the additional modality for samples that the baseline model already predicts as belonging to the positive class with high confidence. The acquisition score $g_i^{\text{UP}}$ for sample $i$ is simply its initial probability $p_i^{\text{avail}}$ based on the available modality:

$$g_i^{\text{UP}} = p_i^{\text{avail}}, \tag{21}$$

Samples with a higher $g_i^{\text{UP}}$ are prioritized for acquisition.

## C.5 IMPUTATION-BASED STRATEGIES

Having explored strategies that assume perfect knowledge of the true labels $y_i$ and/or future scores $s_i^{\text{acquired}}$, and simpler baselines relying only on current information $s_i^{\text{avail}}$, we now introduce methods aiming to bridge the gap by offering a practical and label-agnostic pathway to modality acquisition. They operate by utilizing an imputation model, $f_{\text{imp}}$, to generate a set of $K$ plausible future scores, denoted $\{s_{i,k}^{\text{imp}}\}_{k=1}^K$, conditioned on the initially available data $s_i^{\text{avail}}$. The core principle of these strategies is to then derive acquisition scores from statistics of this imputed score distribution, with the goal of emulating the decision-making process, but without requiring true future knowledge at test time.

**Maximum Expected Probability** The Maximum Expected Probability strategy prioritizes samples which have the highest average probability of belonging to the positive class after modality acquisition. It relies on the set of $K$ imputed future probabilities $\{p_{i,k}^{\text{imp}}\}_{k=1}^K$, where each $p_{i,k}^{\text{imp}} = \sigma(s_{i,k}^{\text{imp}})$ is derived from an imputed future score $s_{i,k}^{\text{imp}}$. The acquisition score $g_i^{\text{eP}}$ for sample $i$ is the mean of these imputed probabilities:

$$g_i^{\text{eP}} = \frac{1}{K} \sum_{k=1}^K p_{i,k}^{\text{imp}}. \tag{22}$$

Samples with a higher $g_i^{\text{eP}}$ are selected, representing instances where the imputation model, on average, predicts a high likelihood of being positive if the additional modality were acquired.

**Maximum Expected Uncertainty Reduction** The Maximum Expected Uncertainty Reduction strategy aims to select samples for which the acquisition of the additional modality is anticipated to yield the largest average decrease in predictive uncertainty (Equation (15)). This strategy considers the initial entropy $\mathcal{H}(p_i^{\text{avail}})$, and the distribution of entropies $\{\mathcal{H}(p_{i,k}^{\text{imp}})\}_{k=1}^K$. The acquisition score $g_i^{\text{eUR}}$ is the difference between the initial entropy and the mean of the imputed future entropies:

$$g_i^{\text{eUR}} = \mathcal{H}(p_i^{\text{avail}}) - \frac{1}{K} \sum_{k=1}^{K} \mathcal{H}(\text{p}_{i,k}^{\text{imp}}). \tag{23}$$

Samples with higher $g_i^{\text{eUR}}$ are prioritized, indicating a greater expected clarification of the prediction upon acquiring the new modality.

**Expected Rank Change**   The Maximum Expected Rank Change strategy prioritizes samples for which the acquisition of the additional modality is anticipated to cause the largest change in their rank, relative to the initial ranking based on $p_i^{\text{avail}}$. It aims to mirror the "Maximum True Rank Change" strategy by using imputed future probabilities. Let $R(p_i^{\text{avail}})$ denote the rank of sample $i$ when all $N$ samples in the cohort are ordered by their initial probabilities $p_j^{\text{avail}}$ (for $j = 1, \ldots, N$). For each of the $K$ imputed future probabilities $\text{p}_{i,k}^{\text{imp}}$ for sample $i$, let $R(\text{p}_{i,k}^{\text{imp}})$ denote the rank of sample $i$ if its probability were $\text{p}_{i,k}^{\text{imp}}$ while all other samples $j \neq i$ retain their initial probabilities $p_j^{\text{avail}}$. The acquisition score $g_i^{\text{eRC}}$ is then the mean of the absolute differences between these imputed future ranks and the initial rank:

$$g_i^{\text{eRC}} = \frac{1}{K} \sum_{k=1}^{K} |R(\text{p}_{i,k}^{\text{imp}}) - R(p_i^{\text{avail}})|. \tag{24}$$

Samples with a higher $g_i^{\text{eRC}}$ are selected, as they are expected to experience the largest shift in their rank-ordered position relative to other samples in the cohort upon modality acquisition.

**Expected KL-Divergence**   The Expected KL-Divergence strategy selects samples where the initial probability distribution is expected to diverge most significantly from the future probability distributions derived from the $K$ imputed scores. The acquisition score $g_i^{\text{eKLD}}$ is the average KL-Divergence $D_{\text{KL}}(P_i^{\text{avail}} \| P_i^{(\text{imp},k)})$ over the $K$ imputations:

$$g_i^{\text{eKLD}} = \frac{1}{K} \sum_{k=1}^{K} D_{\text{KL}} \left( P_i^{\text{avail}} \middle\| P_i^{(\text{imp},k)} \right). \tag{25}$$

A higher $g_i^{\text{eKLD}}$ indicates that, on average, the imputed future predictions substantially differ from the initial baseline prediction, suggesting a significant informational update from acquiring the additional modality.

## D   HYPERPARAMETERS, MODEL DETAILS AND COMPUTE ENVIRONMENT

We employ domain-specific encoders to process the respective modalities: for language inputs, we use a pre-trained BERT model (Devlin et al., 2019), for vision, a Vision Transformer (ViT) (Dosovitskiy et al., 2021). Other data types, *e.g.,* temporal sequences, tabular data, or pre-extracted embeddings, are handled by Transformer encoders (Vaswani et al., 2017). We use well-established hyperparameters from the literature for the modality-specific encoders and only optimize the remaining parameters. Notably, our experiments compared three approaches for normalizing the encoder output: No Normalization, Batch Normalization, and Layer Normalization. We found Layer Normalization to be particularly advantageous, as it both stabilized training convergence and significantly enhanced the performance of the DDPMs. We also evaluated the impact of using only the `CLS` token representation from the encoder versus leveraging the full output sequence. This comparison revealed no substantial effect on performance, suggesting the sufficiency of the `CLS` token representation for our task. Layers in the network are initialized using He initialization (He et al., 2015) if they were not pre-initialized by the specific encoder architecture. We find this particularly important for stabilizing the DDPMs during the early epochs of end-to-end model training.

We perform hyperparameter sweeps for the remaining parts of the designed model in the following ranges:

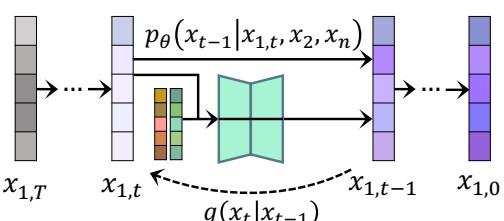

Figure 4: The latent DDPM with its (de)noising functions. Coloring represents less noise in the latent space, starting with pure noise in $X_{i,T} = X_{1,T}$ with $T$ steps. The DDPM is conditioned with two non-missing latent spaces, each from one remaining modality respectively.

- Transformer Head
    - Embedding dimension: `[32, 64, 128, 256, 512, 1024]`
    - Feed-Forward network: `[128, 256, 512, 1024, 2048]`
    - Dropout: `[0, 0.1, 0.2]`
    - Number of heads: `[4, 8, 16]`
    - Number of layers: `[2, 4, 6, 8]`
- DDPMs
    - Embedding dimension: analogous to Transformer head
    - Hidden dimension: `[32, 64, 128, 256, 512, 1024]`
    - Dropout: `[0, 0.1, 0.2]`
    - Number of heads: `[4, 8, 16]`
    - Number of layers: `[2, 4, 6, 8]`
    - Number of steps: `[10, 25, 50, 100, 250, 500]`
- ScheduleFree Optimizer
    - Learning rate: `[1e-1, 1e-2, 1e-3, 3e-4, 1e-4, 1e-5]`
    - Warmup steps: `[0, 100, 200]`
    - Weight decay: `[0, 0.01, 0.001]`

The models are trained with early stopping but without any maximum number of epochs. For the imputation-based acquisition functions, 100 DDPM samples are used during inference of the model.

Our experiments are conducted on a High-Performance Cluster (HPC) with the following environment:

1. 21 Dell PowerEdge R7525 compute nodes, each with:
    - 64 AMD Epyc cores (Rome)
    - 512GB RAM
    - 1 NVIDIA A100 40G GPU
2. 2 Dell PowerEdge XE8545 compute nodes, each with:
    - 128 AMD Epyc cores (Milan)
    - 512GB RAM
    - 4 NVIDIA A100 40G GPUs (NVLink-connected)

# E   DATASET DETAILS

We evaluate CAMA on four diverse, real-world multimodal datasets, spanning domains from health-care to emotion recognition.

**MIMIC Symile**    This clinical dataset is derived from the MIMIC database and is designed for predicting the diagnosis of ten classes (Fracture, Enlarged Cardiomediastinum, Consolidation, Atelectasis, Edema, Cardiomegaly, Lung Lesion, Lung Opacity, Pneumonia, Pneumothorax). It contains 10,345 samples from patients in intensive care units. For our experiments, we utilize three distinct modalities: laboratory values, chest X-ray images, and electrocardiograms (ECGs).

**MIMIC HAIM**    This healthcare benchmark also focuses on the diagnostic prediction of ten classes (Fracture, Enlarged Cardiomediastinum, Consolidation, Atelectasis, Edema, Cardiomegaly, Lung Lesion, Lung Opacity, Pneumonia, Pneumothorax). The bimodal dataset consists of 45,050 samples. The two modalities used in our study are laboratory values and chest X-ray images.

**CMU-MOSEI**    This large-scale benchmark targets multimodal sentiment analysis and emotion recognition with seven classes covering different emotions. It contains 22,856 video samples of speakers expressing opinions. The dataset comprises three modalities: vision, acoustics, and language. Notably, unlike the other datasets, we utilize the pre-computed embeddings provided by the authors rather than the raw data.

**UK Biobank (UKBB)**    The UK Biobank is a large-scale, prospective biomedical database from half a million UK participants. In our experiments, the costly modality targeted for acquisition is proteomics, which is available for only a fraction of the full cohort. We constructed a subset of 100,000 samples in which approximately half include proteomics data, accurately simulating a resource-constrained acquisition scenario. The 15 modalities utilized include electronic health records (EHRs), NMR metabolomics, proteomics, physical activity measurements, diet and alcohol consumption questionnaires, baseline characteristics, smoking status, physiological measurements, anthropometry, hand grip strength, cognitive function tests, ECGs, polygenic risk scores (PRS), and arterial stiffness measurements.

# F  RESULTS FOR SYMILE WITH BC-VAEs

Table 8: Acquisition performance on Symile (AUROC) with Beta-Conditional Variational Auto Encoders. Strategies are grouped by category. Best strategy among proposed ones and baselines in bold for each column.

| Strategy | Fracture | Enl. Card. | Consolidation | Atelectasis | Edema | Mean |
|---|---|---|---|---|---|---|
| *Upper Bounds (for reference)* | | | | | | |
| Oracle | $2.807 \pm 0.326$ | $4.224 \pm 0.501$ | $2.716 \pm 0.147$ | $5.901 \pm 0.861$ | $2.096 \pm 0.078$ | 4.423 |
| True KL-Div. | $1.009 \pm 0.115$ | $0.752 \pm 0.095$ | $0.855 \pm 0.020$ | $0.689 \pm 0.092$ | $0.900 \pm 0.007$ | 0.800 |
| True Rank | $0.714 \pm 0.094$ | $0.728 \pm 0.070$ | $0.853 \pm 0.017$ | $0.777 \pm 0.104$ | $0.890 \pm 0.007$ | 0.719 |
| True Uncert. | $0.939 \pm 0.088$ | $0.735 \pm 0.113$ | $0.571 \pm 0.024$ | $0.106 \pm 0.079$ | $0.719 \pm 0.009$ | 0.555 |
| *Imputation-based (proposed)* | | | | | | |
| KL-Div. | $0.834 \pm 0.060$ | $0.420 \pm 0.109$ | $\mathbf{0.684 \pm 0.017}$ | $\mathbf{0.527 \pm 0.077}$ | $\mathbf{0.744 \pm 0.013}$ | $\mathbf{0.584}$ |
| Prob. | $0.643 \pm 0.073$ | $\mathbf{0.559 \pm 0.037}$ | $0.489 \pm 0.022$ | $-0.304 \pm 0.170$ | $0.603 \pm 0.009$ | 0.395 |
| Rank | $0.252 \pm 0.109$ | $0.281 \pm 0.063$ | $0.526 \pm 0.019$ | $0.327 \pm 0.087$ | $0.444 \pm 0.009$ | 0.366 |
| Uncert. | $\mathbf{0.911 \pm 0.073}$ | $0.557 \pm 0.042$ | $0.593 \pm 0.023$ | $0.162 \pm 0.075$ | $0.637 \pm 0.012$ | 0.519 |
| *Baselines (no imputation)* | | | | | | |
| Uncert. | $0.862 \pm 0.092$ | $0.397 \pm 0.052$ | $0.510 \pm 0.017$ | $0.423 \pm 0.046$ | $0.592 \pm 0.008$ | 0.477 |
| Prob. | $0.127 \pm 0.066$ | $0.508 \pm 0.026$ | $0.526 \pm 0.016$ | $0.054 \pm 0.133$ | $0.462 \pm 0.006$ | 0.388 |
| Random | $0.429 \pm 0.085$ | $0.290 \pm 0.060$ | $0.497 \pm 0.019$ | $0.102 \pm 0.097$ | $0.480 \pm 0.006$ | 0.350 |
| **Strategy** | **Cardiomegaly** | **Lung Lesion** | **Lung Opacity** | **Pneumonia** | **Pneumothorax** | **Mean** |
| *Upper Bounds (for reference)* | | | | | | |
| Oracle | $2.715 \pm 0.124$ | $4.677 \pm 0.799$ | $6.125 \pm 0.911$ | $4.317 \pm 0.181$ | $8.654 \pm 0.796$ | 4.423 |
| True KL-Div. | $0.871 \pm 0.011$ | $0.701 \pm 0.189$ | $0.582 \pm 0.014$ | $0.892 \pm 0.014$ | $0.745 \pm 0.096$ | 0.800 |
| True Rank | $0.843 \pm 0.015$ | $0.457 \pm 0.225$ | $0.501 \pm 0.140$ | $0.825 \pm 0.022$ | $0.603 \pm 0.102$ | 0.719 |
| True Uncert. | $0.701 \pm 0.016$ | $0.626 \pm 0.130$ | $0.147 \pm 0.057$ | $0.664 \pm 0.023$ | $0.343 \pm 0.036$ | 0.555 |
| *Imputation-based (proposed)* | | | | | | |
| KL-Div. | $\mathbf{0.718 \pm 0.019}$ | $0.456 \pm 0.097$ | $\mathbf{0.324 \pm 0.232}$ | $\mathbf{0.757 \pm 0.025}$ | $0.380 \pm 0.129$ | $\mathbf{0.584}$ |
| Prob. | $0.553 \pm 0.014$ | $\mathbf{0.710 \pm 0.213}$ | $0.023 \pm 0.076$ | $0.096 \pm 0.030$ | $\mathbf{0.580 \pm 0.052}$ | 0.395 |
| Rank | $0.416 \pm 0.020$ | $0.357 \pm 0.309$ | $0.311 \pm 0.087$ | $0.493 \pm 0.019$ | $0.251 \pm 0.065$ | 0.366 |
| Uncert. | $0.629 \pm 0.018$ | $0.596 \pm 0.131$ | $0.095 \pm 0.161$ | $0.567 \pm 0.019$ | $0.448 \pm 0.041$ | 0.519 |
| *Baselines (no imputation)* | | | | | | |
| Uncert. | $0.531 \pm 0.015$ | $0.534 \pm 0.087$ | $-0.022 \pm 0.318$ | $0.441 \pm 0.017$ | $0.500 \pm 0.023$ | 0.477 |
| Prob. | $0.424 \pm 0.013$ | $0.468 \pm 0.137$ | $0.291 \pm 0.066$ | $0.518 \pm 0.028$ | $0.499 \pm 0.023$ | 0.388 |
| Random | $0.418 \pm 0.014$ | $0.441 \pm 0.220$ | $0.152 \pm 0.115$ | $0.399 \pm 0.018$ | $0.291 \pm 0.087$ | 0.350 |

Table 9: Acquisition performance on Symile (AUPRC) with Beta-Conditional Variational Auto Encoders. Strategies are grouped by category. Best strategy among proposed ones and baselines in bold for each column.

| Strategy | Fracture | Enl. Card. | Consolidation | Atelectasis | Edema | Mean |
|---|---|---|---|---|---|---|
| *Upper Bounds (for reference)* | | | | | | |
| Oracle | $1.785 \pm 0.110$ | $3.468 \pm 0.317$ | $2.780 \pm 0.146$ | $2.632 \pm 0.146$ | $2.460 \pm 0.099$ | 4.116 |
| True KL-Div. | $0.798 \pm 0.050$ | $0.700 \pm 0.101$ | $0.828 \pm 0.020$ | $0.686 \pm 0.026$ | $0.888 \pm 0.007$ | 0.766 |
| True Rank | $0.736 \pm 0.061$ | $0.625 \pm 0.096$ | $0.822 \pm 0.022$ | $0.733 \pm 0.046$ | $0.843 \pm 0.009$ | 0.689 |
| True Uncert. | $0.778 \pm 0.050$ | $0.623 \pm 0.044$ | $0.606 \pm 0.034$ | $0.206 \pm 0.033$ | $0.731 \pm 0.011$ | 0.513 |
| *Imputation-based (proposed)* | | | | | | |
| KL-Div. | $0.725 \pm 0.062$ | $0.574 \pm 0.095$ | $\mathbf{0.636 \pm 0.023}$ | $\mathbf{0.599 \pm 0.034}$ | $\mathbf{0.733 \pm 0.013}$ | $\mathbf{0.624}$ |
| Prob. | $0.642 \pm 0.033$ | $\mathbf{0.610 \pm 0.037}$ | $0.539 \pm 0.039$ | $0.072 \pm 0.059$ | $0.696 \pm 0.009$ | 0.433 |
| Rank | $0.355 \pm 0.050$ | $0.334 \pm 0.101$ | $0.409 \pm 0.029$ | $0.405 \pm 0.032$ | $0.427 \pm 0.010$ | 0.409 |
| Uncert. | $\mathbf{0.757 \pm 0.056}$ | $0.584 \pm 0.048$ | $0.592 \pm 0.032$ | $0.228 \pm 0.030$ | $0.637 \pm 0.013$ | 0.480 |
| *Baselines (no imputation)* | | | | | | |
| Uncert. | $0.713 \pm 0.054$ | $0.239 \pm 0.127$ | $0.429 \pm 0.024$ | $0.335 \pm 0.022$ | $0.579 \pm 0.008$ | 0.438 |
| Prob. | $0.265 \pm 0.039$ | $0.594 \pm 0.023$ | $0.567 \pm 0.025$ | $0.405 \pm 0.035$ | $0.565 \pm 0.009$ | 0.523 |
| Random | $0.455 \pm 0.056$ | $0.248 \pm 0.076$ | $0.444 \pm 0.027$ | $0.336 \pm 0.044$ | $0.485 \pm 0.010$ | 0.380 |
| **Strategy** | **Cardiomegaly** | **Lung Lesion** | **Lung Opacity** | **Pneumonia** | **Pneumothorax** | **Mean** |
| *Upper Bounds (for reference)* | | | | | | |
| Oracle | $2.902 \pm 0.184$ | $2.559 \pm 0.272$ | $3.156 \pm 0.152$ | $5.255 \pm 0.253$ | $14.162 \pm 1.561$ | 4.116 |
| True KL-Div. | $0.871 \pm 0.014$ | $0.488 \pm 0.088$ | $0.700 \pm 0.026$ | $0.879 \pm 0.021$ | $0.818 \pm 0.044$ | 0.766 |
| True Rank | $0.828 \pm 0.022$ | $0.533 \pm 0.068$ | $0.667 \pm 0.036$ | $0.824 \pm 0.028$ | $0.278 \pm 0.139$ | 0.689 |
| True Uncert. | $0.745 \pm 0.016$ | $0.183 \pm 0.045$ | $0.151 \pm 0.032$ | $0.525 \pm 0.023$ | $0.579 \pm 0.076$ | 0.513 |
| *Imputation-based (proposed)* | | | | | | |
| KL-Div. | $\mathbf{0.749 \pm 0.023}$ | $0.390 \pm 0.117$ | $\mathbf{0.584 \pm 0.023}$ | $\mathbf{0.736 \pm 0.021}$ | $0.514 \pm 0.099$ | $\mathbf{0.624}$ |
| Prob. | $0.609 \pm 0.016$ | $0.221 \pm 0.103$ | $0.013 \pm 0.048$ | $0.064 \pm 0.037$ | $0.868 \pm 0.034$ | 0.433 |
| Rank | $0.472 \pm 0.015$ | $0.495 \pm 0.095$ | $0.336 \pm 0.030$ | $0.412 \pm 0.026$ | $0.448 \pm 0.085$ | 0.409 |
| Uncert. | $0.663 \pm 0.019$ | $-0.030 \pm 0.109$ | $0.202 \pm 0.029$ | $0.454 \pm 0.021$ | $0.719 \pm 0.105$ | 0.480 |
| *Baselines (no imputation)* | | | | | | |
| Uncert. | $0.534 \pm 0.019$ | $0.021 \pm 0.127$ | $0.301 \pm 0.025$ | $0.357 \pm 0.018$ | $\mathbf{0.871 \pm 0.076}$ | 0.438 |
| Prob. | $0.478 \pm 0.016$ | $\mathbf{0.549 \pm 0.063}$ | $0.407 \pm 0.037$ | $0.531 \pm 0.022$ | $\mathbf{0.871 \pm 0.076}$ | 0.523 |
| Random | $0.465 \pm 0.020$ | $0.320 \pm 0.099$ | $0.202 \pm 0.030$ | $0.364 \pm 0.023$ | $0.483 \pm 0.075$ | 0.380 |

# G   DETAILED RESULTS FOR MOSEI

Table 10: Acquisition performance on MOSEI (Image imputed by Text), showing $G_{\text{full}}$ for AU-ROC/AUPRC. Strategies are grouped by category. Best strategy among proposed ones and baselines in bold.

| | AUROC | AUPRC |
|---|---|---|
| **Strategy** | $G_{\text{full}} \pm$ SEM ↑ | $G_{\text{full}} \pm$ SEM ↑ |
| *Upper Bounds (for reference)* | | |
| Oracle | $0.995 \pm 0.006$ | $0.995 \pm 0.012$ |
| True KL-Div. | $0.777 \pm 0.005$ | $0.790 \pm 0.015$ |
| True Rank | $0.763 \pm 0.012$ | $0.781 \pm 0.014$ |
| True Uncert. | $0.599 \pm 0.015$ | $0.673 \pm 0.019$ |
| *Imputation-based (proposed)* | | |
| KL-Div | $\mathbf{0.551 \pm 0.012}$ | $0.590 \pm 0.009$ |
| Probability | $0.473 \pm 0.010$ | $\mathbf{0.598 \pm 0.006}$ |
| Rank | $0.524 \pm 0.012$ | $0.560 \pm 0.012$ |
| Uncertainty | $0.500 \pm 0.021$ | $0.567 \pm 0.009$ |
| *Baselines (no imputation)* | | |
| Uncertainty | $0.507 \pm 0.015$ | $0.565 \pm 0.018$ |
| Probability | $0.451 \pm 0.013$ | $0.578 \pm 0.009$ |
| Random | $0.521 \pm 0.006$ | $0.575 \pm 0.008$ |

Table 11: Acquisition performance on MOSEI (Image imputed by Audio), showing $G_{\text{full}}$ for AU-ROC/AUPRC. Strategies are grouped by category. Best strategy among proposed ones and baselines in bold.

| | AUROC | AUPRC |
|---|---|---|
| **Strategy** | $G_{\text{full}} \pm$ SEM ↑ | $G_{\text{full}} \pm$ SEM ↑ |
| *Upper Bounds (for reference)* | | |
| Oracle | $1.052 \pm 0.007$ | $1.011 \pm 0.010$ |
| True KL-Div. | $0.785 \pm 0.009$ | $0.803 \pm 0.005$ |
| True Rank | $0.783 \pm 0.010$ | $0.802 \pm 0.012$ |
| True Uncert. | $0.672 \pm 0.007$ | $0.742 \pm 0.006$ |
| *Imputation-based (proposed)* | | |
| KL-Div | $0.566 \pm 0.011$ | $0.601 \pm 0.009$ |
| Probability | $0.547 \pm 0.002$ | $\mathbf{0.629 \pm 0.004}$ |
| Rank | $\mathbf{0.576 \pm 0.013}$ | $0.614 \pm 0.007$ |
| Uncertainty | $0.545 \pm 0.009$ | $0.586 \pm 0.007$ |
| *Baselines (no imputation)* | | |
| Uncertainty | $0.545 \pm 0.009$ | $0.601 \pm 0.014$ |
| Probability | $0.526 \pm 0.014$ | $0.616 \pm 0.008$ |
| Random | $0.553 \pm 0.008$ | $0.603 \pm 0.005$ |

Table 12: Acquisition performance on MOSEI (Image imputed by Text and Audio), showing $G_{\text{full}}$ for AUROC/AUPRC. Strategies are grouped by category. Best strategy among proposed ones and baselines in bold.

|  | AUROC | AUPRC |
|---|---|---|
| **Strategy** | $G_{\text{full}} \pm \text{SEM} \uparrow$ | $G_{\text{full}} \pm \text{SEM} \uparrow$ |
| *Upper Bounds (for reference)* |  |  |
| Oracle | $1.321 \pm 0.009$ | $1.315 \pm 0.012$ |
| True KL-Div. | $0.979 \pm 0.006$ | $0.862 \pm 0.006$ |
| True Rank | $0.960 \pm 0.005$ | $0.811 \pm 0.006$ |
| True Uncert. | $0.716 \pm 0.004$ | $0.750 \pm 0.004$ |
| *Imputation-based (proposed)* |  |  |
| KL-Div | $\mathbf{0.603 \pm 0.007}$ | $0.627 \pm 0.009$ |
| Probability | $0.513 \pm 0.002$ | $0.610 \pm 0.003$ |
| Rank | $0.463 \pm 0.005$ | $0.491 \pm 0.005$ |
| Uncertainty | $0.499 \pm 0.004$ | $0.452 \pm 0.004$ |
| *Baselines (no imputation)* |  |  |
| Uncertainty | $0.513 \pm 0.004$ | $0.480 \pm 0.005$ |
| Probability | $0.534 \pm 0.008$ | $\mathbf{0.676 \pm 0.007}$ |
| Random | $0.489 \pm 0.002$ | $0.527 \pm 0.002$ |

Table 13: Acquisition performance on MOSEI (Text imputed by Image), showing $G_{\text{full}}$ for AUROC/AUPRC. Strategies are grouped by category. Best strategy among proposed ones and baselines in bold.

|  | AUROC | AUPRC |
|---|---|---|
| **Strategy** | $G_{\text{full}} \pm \text{SEM} \uparrow$ | $G_{\text{full}} \pm \text{SEM} \uparrow$ |
| *Upper Bounds (for reference)* |  |  |
| Oracle | $1.613 \pm 0.269$ | $1.493 \pm 0.166$ |
| True KL-Div. | $0.845 \pm 0.018$ | $0.843 \pm 0.010$ |
| True Rank | $0.772 \pm 0.024$ | $0.784 \pm 0.025$ |
| True Uncert. | $0.633 \pm 0.060$ | $0.697 \pm 0.030$ |
| *Imputation-based (proposed)* |  |  |
| KL-Div | $\mathbf{0.900 \pm 0.023}$ | $\mathbf{0.896 \pm 0.022}$ |
| Probability | $0.806 \pm 0.022$ | $0.861 \pm 0.023$ |
| Rank | $0.309 \pm 0.089$ | $0.420 \pm 0.065$ |
| Uncertainty | $0.651 \pm 0.049$ | $0.747 \pm 0.030$ |
| *Baselines (no imputation)* |  |  |
| Uncertainty | $0.466 \pm 0.045$ | $0.537 \pm 0.014$ |
| Probability | $0.418 \pm 0.055$ | $0.532 \pm 0.045$ |
| Random | $0.417 \pm 0.061$ | $0.489 \pm 0.051$ |

Table 14: Acquisition performance on MOSEI (Text imputed by Audio), showing $G_{\text{full}}$ for AUROC/AUPRC. Strategies are grouped by category. Best strategy among proposed ones and baselines in bold.

|  | AUROC | AUPRC |
|---|---|---|
| **Strategy** | $G_{\text{full}} \pm \text{SEM} \uparrow$ | $G_{\text{full}} \pm \text{SEM} \uparrow$ |
| *Upper Bounds (for reference)* |  |  |
| Oracle | $8.867 \pm 1.712$ | $16.066 \pm 2.592$ |
| True KL-Div. | $0.645 \pm 0.051$ | $0.387 \pm 0.136$ |
| True Rank | $0.494 \pm 0.147$ | $0.015 \pm 0.262$ |
| True Uncert. | $0.913 \pm 0.121$ | $0.893 \pm 0.141$ |
| *Imputation-based (proposed)* |  |  |
| KL-Div | $2.962 \pm 0.925$ | $4.582 \pm 1.938$ |
| Probability | $\mathbf{3.157 \pm 1.229}$ | $\mathbf{6.861 \pm 1.806}$ |
| Rank | $-0.413 \pm 0.421$ | $-1.134 \pm 0.463$ |
| Uncertainty | $1.592 \pm 0.350$ | $3.844 \pm 0.806$ |
| *Baselines (no imputation)* |  |  |
| Uncertainty | $0.591 \pm 0.147$ | $0.473 \pm 0.204$ |
| Probability | $0.662 \pm 0.071$ | $0.971 \pm 0.014$ |
| Random | $0.316 \pm 0.059$ | $0.376 \pm 0.086$ |

Table 15: Acquisition performance on MOSEI (Text imputed by Image and Audio), showing $G_{\text{full}}$ for AUROC/AUPRC. Strategies are grouped by category. Best strategy among proposed ones and baselines in bold.

| | AUROC | AUPRC |
|---|---|---|
| **Strategy** | $G_{\text{full}} \pm$ SEM $\uparrow$ | $G_{\text{full}} \pm$ SEM $\uparrow$ |
| *Upper Bounds (for reference)* | | |
| Oracle | $1.207 \pm 0.012$ | $1.280 \pm 0.013$ |
| True KL-Div. | $0.851 \pm 0.002$ | $0.842 \pm 0.004$ |
| True Rank | $0.836 \pm 0.004$ | $0.840 \pm 0.005$ |
| True Uncert. | $0.649 \pm 0.009$ | $0.691 \pm 0.009$ |
| *Imputation-based (proposed)* | | |
| KL-Div | $\mathbf{0.892 \pm 0.002}$ | $\mathbf{0.894 \pm 0.004}$ |
| Probability | $0.665 \pm 0.004$ | $0.727 \pm 0.003$ |
| Rank | $0.489 \pm 0.003$ | $0.520 \pm 0.003$ |
| Uncertainty | $0.662 \pm 0.008$ | $0.720 \pm 0.007$ |
| *Baselines (no imputation)* | | |
| Uncertainty | $0.538 \pm 0.009$ | $0.567 \pm 0.009$ |
| Probability | $0.379 \pm 0.003$ | $0.462 \pm 0.004$ |
| Random | $0.512 \pm 0.002$ | $0.531 \pm 0.003$ |

Table 16: Acquisition performance on MOSEI (Audio imputed by Image), showing $G_{\text{full}}$ for AUROC/AUPRC. Strategies are grouped by category. Best strategy among proposed ones and baselines in bold.

| | AUROC | AUPRC |
|---|---|---|
| **Strategy** | $G_{\text{full}} \pm$ SEM $\uparrow$ | $G_{\text{full}} \pm$ SEM $\uparrow$ |
| *Upper Bounds (for reference)* | | |
| Oracle | $1.238 \pm 0.124$ | $1.207 \pm 0.093$ |
| True KL-Div. | $0.826 \pm 0.014$ | $0.821 \pm 0.019$ |
| True Rank | $0.752 \pm 0.030$ | $0.780 \pm 0.025$ |
| True Uncert. | $0.544 \pm 0.034$ | $0.627 \pm 0.023$ |
| *Imputation-based (proposed)* | | |
| KL-Div | $\mathbf{0.800 \pm 0.012}$ | $\mathbf{0.803 \pm 0.017}$ |
| Probability | $0.684 \pm 0.020$ | $0.737 \pm 0.024$ |
| Rank | $0.326 \pm 0.072$ | $0.445 \pm 0.052$ |
| Uncertainty | $0.552 \pm 0.035$ | $0.625 \pm 0.023$ |
| *Baselines (no imputation)* | | |
| Uncertainty | $0.436 \pm 0.043$ | $0.502 \pm 0.016$ |
| Probability | $0.374 \pm 0.034$ | $0.510 \pm 0.034$ |
| Random | $0.397 \pm 0.052$ | $0.478 \pm 0.039$ |

Table 17: Acquisition performance on MOSEI (Audio imputed by Text), showing $G_{\text{full}}$ for AUROC/AUPRC. Strategies are grouped by category. Best strategy among proposed ones and baselines in bold.

| | AUROC | AUPRC |
|---|---|---|
| **Strategy** | $G_{\text{full}} \pm$ SEM $\uparrow$ | $G_{\text{full}} \pm$ SEM $\uparrow$ |
| *Upper Bounds (for reference)* | | |
| Oracle | $4.955 \pm 0.417$ | $6.275 \pm 0.948$ |
| True KL-Div. | $0.815 \pm 0.043$ | $0.817 \pm 0.057$ |
| True Rank | $0.563 \pm 0.073$ | $0.645 \pm 0.098$ |
| True Uncert. | $0.520 \pm 0.026$ | $0.604 \pm 0.059$ |
| *Imputation-based (proposed)* | | |
| KL-Div | $2.305 \pm 0.360$ | $2.335 \pm 0.499$ |
| Probability | $\mathbf{2.306 \pm 0.259}$ | $\mathbf{2.571 \pm 0.483}$ |
| Rank | $-0.210 \pm 0.123$ | $-0.093 \pm 0.135$ |
| Uncertainty | $1.154 \pm 0.141$ | $1.626 \pm 0.303$ |
| *Baselines (no imputation)* | | |
| Uncertainty | $0.519 \pm 0.023$ | $0.604 \pm 0.044$ |
| Probability | $0.416 \pm 0.007$ | $0.511 \pm 0.018$ |
| Random | $0.326 \pm 0.024$ | $0.439 \pm 0.023$ |

Table 18: Acquisition performance on MOSEI (Audio imputed by Image and Text), showing $G_{\text{full}}$ for AUROC/AUPRC. Strategies are grouped by category. Best strategy among proposed ones and baselines in bold.

| | AUROC | AUPRC |
|---|---|---|
| **Strategy** | $G_{\text{full}} \pm$ SEM $\uparrow$ | $G_{\text{full}} \pm$ SEM $\uparrow$ |
| *Upper Bounds (for reference)* | | |
| Oracle | $1.215 \pm 0.010$ | $1.275 \pm 0.011$ |
| True KL-Div. | $0.865 \pm 0.002$ | $0.850 \pm 0.004$ |
| True Rank | $0.833 \pm 0.004$ | $0.837 \pm 0.005$ |
| True Uncert. | $0.645 \pm 0.007$ | $0.693 \pm 0.006$ |
| *Imputation-based (proposed)* | | |
| KL-Div | $\mathbf{0.857 \pm 0.002}$ | $\mathbf{0.846 \pm 0.004}$ |
| Probability | $0.667 \pm 0.003$ | $0.725 \pm 0.003$ |
| Rank | $0.459 \pm 0.004$ | $0.489 \pm 0.004$ |
| Uncertainty | $0.646 \pm 0.007$ | $0.694 \pm 0.007$ |
| *Baselines (no imputation)* | | |
| Uncertainty | $0.536 \pm 0.008$ | $0.566 \pm 0.008$ |
| Probability | $0.368 \pm 0.005$ | $0.462 \pm 0.005$ |
| Random | $0.503 \pm 0.003$ | $0.530 \pm 0.003$ |

# H   DETAILED RESULTS FOR MIMIC SYMILE

Table 19: Acquisition performance on MIMIC Symile for AUROC, showing $G_{\text{full}}$. Strategies are grouped by category. Best strategy among proposed and baseline methods in bold for each column.

| Strategy | Fracture | Enl. Card. | Consolidation | Atelectasis | Edema | Mean |
|---|---|---|---|---|---|---|
| *Upper Bounds (for reference)* | | | | | | |
| Oracle | $2.876 \pm 0.368$ | $3.722 \pm 0.370$ | $2.856 \pm 0.147$ | $4.862 \pm 0.356$ | $2.793 \pm 0.324$ | 4.580 |
| True KL-Div. | $1.029 \pm 0.165$ | $1.019 \pm 0.051$ | $0.885 \pm 0.025$ | $0.841 \pm 0.045$ | $0.920 \pm 0.011$ | 0.883 |
| True Rank | $0.963 \pm 0.153$ | $0.924 \pm 0.058$ | $0.946 \pm 0.025$ | $0.802 \pm 0.068$ | $0.887 \pm 0.022$ | 0.811 |
| True Uncert. | $0.915 \pm 0.138$ | $0.763 \pm 0.052$ | $0.749 \pm 0.031$ | $0.152 \pm 0.052$ | $0.648 \pm 0.012$ | 0.481 |
| *Imputation-based (proposed)* | | | | | | |
| KL-Div | $0.838 \pm 0.164$ | $\mathbf{0.882 \pm 0.064}$ | $\mathbf{0.706 \pm 0.021}$ | $\mathbf{0.779 \pm 0.114}$ | $\mathbf{0.893 \pm 0.080}$ | $\mathbf{0.833}$ |
| Probability | $\mathbf{0.861 \pm 0.118}$ | $0.638 \pm 0.046$ | $0.610 \pm 0.021$ | $0.099 \pm 0.107$ | $0.514 \pm 0.099$ | 0.426 |
| Rank | $0.123 \pm 0.155$ | $0.331 \pm 0.043$ | $0.434 \pm 0.026$ | $0.371 \pm 0.083$ | $0.352 \pm 0.030$ | 0.378 |
| Uncertainty | $0.851 \pm 0.138$ | $0.686 \pm 0.051$ | $0.701 \pm 0.029$ | $0.188 \pm 0.094$ | $0.588 \pm 0.018$ | 0.440 |
| *Baselines (no imputation)* | | | | | | |
| Uncertainty | $0.616 \pm 0.100$ | $0.482 \pm 0.033$ | $0.543 \pm 0.024$ | $0.380 \pm 0.031$ | $0.539 \pm 0.010$ | 0.480 |
| Probability | $0.153 \pm 0.086$ | $0.519 \pm 0.035$ | $0.464 \pm 0.020$ | $0.446 \pm 0.068$ | $0.421 \pm 0.010$ | 0.458 |
| Random | $0.241 \pm 0.121$ | $0.399 \pm 0.044$ | $0.479 \pm 0.020$ | $0.250 \pm 0.080$ | $0.425 \pm 0.017$ | 0.376 |

| Strategy | Cardiomegaly | Lung Lesion | Lung Opacity | Pneumonia | Pneumothorax | Mean |
|---|---|---|---|---|---|---|
| *Upper Bounds (for reference)* | | | | | | |
| Oracle | $2.787 \pm 0.139$ | $4.974 \pm 0.581$ | $6.657 \pm 0.649$ | $4.817 \pm 0.430$ | $9.461 \pm 1.049$ | 4.580 |
| True KL-Div. | $0.885 \pm 0.011$ | $0.753 \pm 0.179$ | $0.750 \pm 0.087$ | $0.837 \pm 0.043$ | $0.910 \pm 0.054$ | 0.883 |
| True Rank | $0.878 \pm 0.019$ | $0.411 \pm 0.216$ | $0.793 \pm 0.056$ | $0.902 \pm 0.029$ | $0.605 \pm 0.053$ | 0.811 |
| True Uncert. | $0.524 \pm 0.025$ | $0.251 \pm 0.177$ | $0.212 \pm 0.054$ | $0.728 \pm 0.023$ | $-0.136 \pm 0.065$ | 0.481 |
| *Imputation-based (proposed)* | | | | | | |
| KL-Div | $\mathbf{0.747 \pm 0.039}$ | $\mathbf{1.266 \pm 0.258}$ | $\mathbf{0.683 \pm 0.106}$ | $\mathbf{0.761 \pm 0.060}$ | $0.773 \pm 0.134$ | $\mathbf{0.833}$ |
| Probability | $0.350 \pm 0.053$ | $0.190 \pm 0.223$ | $-0.075 \pm 0.077$ | $0.172 \pm 0.142$ | $\mathbf{0.898 \pm 0.061}$ | 0.426 |
| Rank | $0.378 \pm 0.016$ | $0.607 \pm 0.150$ | $0.635 \pm 0.080$ | $0.437 \pm 0.054$ | $0.115 \pm 0.082$ | 0.378 |
| Uncertainty | $0.450 \pm 0.041$ | $0.022 \pm 0.173$ | $0.199 \pm 0.057$ | $0.658 \pm 0.045$ | $0.055 \pm 0.060$ | 0.440 |
| *Baselines (no imputation)* | | | | | | |
| Uncertainty | $0.480 \pm 0.013$ | $0.201 \pm 0.179$ | $0.406 \pm 0.041$ | $0.615 \pm 0.019$ | $0.536 \pm 0.040$ | 0.480 |
| Probability | $0.431 \pm 0.015$ | $0.778 \pm 0.212$ | $0.417 \pm 0.065$ | $0.416 \pm 0.055$ | $0.536 \pm 0.040$ | 0.458 |
| Random | $0.385 \pm 0.015$ | $0.365 \pm 0.225$ | $0.381 \pm 0.057$ | $0.505 \pm 0.038$ | $0.327 \pm 0.061$ | 0.376 |

Table 20: Acquisition performance on MIMIC Symile for AUPRC, showing $G_{\text{full}}$. Strategies are grouped by category. Best strategy among proposed and baseline methods in bold for each column.

| Strategy | Fracture | Enl. Card. | Consolidation | Atelectasis | Edema | Mean |
|---|---|---|---|---|---|---|
| *Upper Bounds (for reference)* | | | | | | |
| Oracle | $2.579 \pm 0.343$ | $2.964 \pm 0.201$ | $4.784 \pm 1.130$ | $3.093 \pm 0.201$ | $3.015 \pm 0.141$ | 4.231 |
| True KL-Div. | $0.883 \pm 0.094$ | $0.970 \pm 0.038$ | $0.933 \pm 0.057$ | $0.781 \pm 0.038$ | $0.939 \pm 0.013$ | 0.871 |
| True Rank | $0.659 \pm 0.176$ | $0.858 \pm 0.033$ | $0.903 \pm 0.090$ | $0.720 \pm 0.064$ | $0.897 \pm 0.022$ | 0.776 |
| True Uncert. | $0.604 \pm 0.073$ | $0.812 \pm 0.044$ | $0.709 \pm 0.036$ | $0.050 \pm 0.039$ | $0.645 \pm 0.015$ | 0.450 |
| *Imputation-based (proposed)* | | | | | | |
| KL-Div | $\mathbf{0.770 \pm 0.210}$ | $\mathbf{0.899 \pm 0.044}$ | $\mathbf{0.895 \pm 0.159}$ | $\mathbf{0.684 \pm 0.063}$ | $\mathbf{0.832 \pm 0.039}$ | **0.777** |
| Probability | $0.631 \pm 0.073$ | $0.673 \pm 0.036$ | $0.503 \pm 0.106$ | $0.028 \pm 0.069$ | $0.624 \pm 0.049$ | 0.449 |
| Rank | $0.461 \pm 0.188$ | $0.388 \pm 0.034$ | $0.421 \pm 0.064$ | $0.318 \pm 0.053$ | $0.351 \pm 0.022$ | 0.407 |
| Uncertainty | $0.663 \pm 0.108$ | $0.744 \pm 0.046$ | $0.611 \pm 0.073$ | $0.118 \pm 0.034$ | $0.569 \pm 0.017$ | 0.444 |
| *Baselines (no imputation)* | | | | | | |
| Uncertainty | $0.428 \pm 0.155$ | $0.524 \pm 0.031$ | $0.509 \pm 0.036$ | $0.246 \pm 0.018$ | $0.519 \pm 0.011$ | 0.443 |
| Probability | $0.242 \pm 0.068$ | $0.523 \pm 0.026$ | $0.546 \pm 0.030$ | $0.512 \pm 0.064$ | $0.533 \pm 0.013$ | 0.550 |
| Random | $0.149 \pm 0.103$ | $0.423 \pm 0.032$ | $0.429 \pm 0.097$ | $0.222 \pm 0.094$ | $0.448 \pm 0.012$ | 0.388 |

| Strategy | Cardiomegaly | Lung Lesion | Lung Opacity | Pneumonia | Pneumothorax | Mean |
|---|---|---|---|---|---|---|
| *Upper Bounds (for reference)* | | | | | | |
| Oracle | $2.746 \pm 0.110$ | $2.520 \pm 0.250$ | $4.092 \pm 0.259$ | $5.895 \pm 0.406$ | $10.623 \pm 0.708$ | 4.231 |
| True KL-Div. | $0.853 \pm 0.010$ | $0.828 \pm 0.073$ | $0.792 \pm 0.037$ | $0.906 \pm 0.032$ | $0.827 \pm 0.043$ | 0.871 |
| True Rank | $0.882 \pm 0.018$ | $0.676 \pm 0.088$ | $0.771 \pm 0.045$ | $0.911 \pm 0.030$ | $0.483 \pm 0.075$ | 0.776 |
| True Uncert. | $0.473 \pm 0.029$ | $0.181 \pm 0.067$ | $0.140 \pm 0.029$ | $0.595 \pm 0.024$ | $0.293 \pm 0.052$ | 0.450 |
| *Imputation-based (proposed)* | | | | | | |
| KL-Div | $\mathbf{0.729 \pm 0.034}$ | $\mathbf{0.896 \pm 0.146}$ | $\mathbf{0.722 \pm 0.043}$ | $\mathbf{0.763 \pm 0.059}$ | $0.581 \pm 0.084$ | **0.777** |
| Probability | $0.366 \pm 0.047$ | $0.320 \pm 0.104$ | $0.041 \pm 0.065$ | $0.339 \pm 0.079$ | $\mathbf{0.965 \pm 0.027}$ | 0.449 |
| Rank | $0.384 \pm 0.014$ | $0.564 \pm 0.086$ | $0.402 \pm 0.050$ | $0.389 \pm 0.034$ | $0.396 \pm 0.054$ | 0.407 |
| Uncertainty | $0.424 \pm 0.038$ | $0.130 \pm 0.053$ | $0.149 \pm 0.034$ | $0.519 \pm 0.033$ | $0.513 \pm 0.066$ | 0.444 |
| *Baselines (no imputation)* | | | | | | |
| Uncertainty | $0.434 \pm 0.018$ | $0.215 \pm 0.033$ | $0.260 \pm 0.023$ | $0.482 \pm 0.019$ | $0.811 \pm 0.041$ | 0.443 |
| Probability | $0.479 \pm 0.017$ | $0.756 \pm 0.136$ | $0.555 \pm 0.037$ | $0.541 \pm 0.029$ | $0.811 \pm 0.041$ | 0.550 |
| Random | $0.386 \pm 0.013$ | $0.503 \pm 0.103$ | $0.354 \pm 0.053$ | $0.435 \pm 0.033$ | $0.527 \pm 0.053$ | 0.388 |

Table 21: Acquisition performance on MIMIC Symile for AUROC (Image imputed by Lab), showing $G_{\text{full}}$. Strategies are grouped by category. Best strategy among proposed and baseline methods in bold for each column.

| Strategy | Fracture | Enl. Card. | Consolidation | Atelectasis | Edema | Mean |
|---|---|---|---|---|---|---|
| *Upper Bounds (for reference)* | | | | | | |
| Oracle | $4.144 \pm 2.172$ | $2.120 \pm 0.465$ | $2.803 \pm 0.306$ | $2.381 \pm 0.215$ | $1.684 \pm 0.070$ | 4.104 |
| True KL-Div. | $0.728 \pm 0.137$ | $0.816 \pm 0.011$ | $0.611 \pm 0.084$ | $0.837 \pm 0.028$ | $0.813 \pm 0.009$ | 0.825 |
| True Rank | $1.062 \pm 0.643$ | $0.614 \pm 0.111$ | $0.591 \pm 0.157$ | $0.816 \pm 0.065$ | $0.765 \pm 0.024$ | 0.707 |
| True Uncert. | $0.686 \pm 0.374$ | $0.606 \pm 0.068$ | $0.637 \pm 0.140$ | $0.496 \pm 0.148$ | $0.619 \pm 0.024$ | 0.601 |
| *Imputation-based (proposed)* | | | | | | |
| KL-Div | $-1.410 \pm 0.037$ | $0.477 \pm 0.108$ | $0.282 \pm 0.146$ | $\mathbf{0.626 \pm 0.137}$ | $0.503 \pm 0.022$ | 0.247 |
| Probability | $\mathbf{0.231 \pm 0.288}$ | $0.498 \pm 0.036$ | $0.514 \pm 0.060$ | $0.503 \pm 0.056$ | $0.454 \pm 0.020$ | 0.379 |
| Rank | $-0.725 \pm 0.858$ | $0.190 \pm 0.201$ | $0.520 \pm 0.018$ | $0.403 \pm 0.085$ | $\mathbf{0.527 \pm 0.032}$ | 0.225 |
| Uncertainty | $0.168 \pm 0.317$ | $0.400 \pm 0.025$ | $\mathbf{0.620 \pm 0.179}$ | $0.422 \pm 0.130$ | $0.493 \pm 0.019$ | **0.448** |
| *Baselines (no imputation)* | | | | | | |
| Uncertainty | $0.129 \pm 0.339$ | $0.332 \pm 0.069$ | $0.423 \pm 0.201$ | $0.456 \pm 0.106$ | $0.503 \pm 0.020$ | 0.433 |
| Probability | $0.129 \pm 0.339$ | $\mathbf{0.515 \pm 0.057}$ | $0.530 \pm 0.049$ | $0.601 \pm 0.101$ | $0.477 \pm 0.018$ | 0.444 |
| Random | $-0.509 \pm 0.041$ | $0.356 \pm 0.070$ | $0.503 \pm 0.163$ | $0.609 \pm 0.091$ | $0.492 \pm 0.023$ | 0.307 |

| Strategy | Cardiomegaly | Lung Lesion | Lung Opacity | Pneumonia | Pneumothorax | Mean |
|---|---|---|---|---|---|---|
| *Upper Bounds (for reference)* | | | | | | |
| Oracle | $1.793 \pm 0.077$ | $5.417 \pm 4.001$ | $7.073 \pm 2.202$ | $4.743 \pm 1.268$ | $8.881 \pm 1.894$ | 4.104 |
| True KL-Div. | $0.822 \pm 0.010$ | $1.543 \pm 0.689$ | $0.663 \pm 0.095$ | $0.884 \pm 0.063$ | $0.530 \pm 0.109$ | 0.825 |
| True Rank | $0.663 \pm 0.042$ | $1.291 \pm 0.557$ | $0.530 \pm 0.154$ | $0.705 \pm 0.170$ | $0.035 \pm 0.068$ | 0.707 |
| True Uncert. | $0.272 \pm 0.091$ | $1.128 \pm 0.870$ | $0.622 \pm 0.187$ | $0.687 \pm 0.038$ | $0.254 \pm 0.213$ | 0.601 |
| *Imputation-based (proposed)* | | | | | | |
| KL-Div | $\mathbf{0.492 \pm 0.030}$ | $\mathbf{0.847 \pm 0.033}$ | $0.459 \pm 0.146$ | $\mathbf{0.481 \pm 0.158}$ | $-0.285 \pm 0.221$ | 0.247 |
| Probability | $0.301 \pm 0.081$ | $0.418 \pm 0.193$ | $0.034 \pm 0.303$ | $0.372 \pm 0.256$ | $0.461 \pm 0.128$ | 0.379 |
| Rank | $0.434 \pm 0.045$ | $-0.219 \pm 0.837$ | $\mathbf{0.956 \pm 0.383}$ | $0.363 \pm 0.080$ | $-0.201 \pm 0.261$ | 0.225 |
| Uncertainty | $0.395 \pm 0.020$ | $0.574 \pm 0.327$ | $0.400 \pm 0.126$ | $0.395 \pm 0.040$ | $\mathbf{0.608 \pm 0.201}$ | **0.448** |
| *Baselines (no imputation)* | | | | | | |
| Uncertainty | $0.433 \pm 0.015$ | $0.490 \pm 0.096$ | $0.558 \pm 0.358$ | $0.424 \pm 0.053$ | $0.587 \pm 0.180$ | 0.433 |
| Probability | $0.419 \pm 0.065$ | $0.505 \pm 0.108$ | $0.263 \pm 0.254$ | $0.417 \pm 0.228$ | $0.587 \pm 0.180$ | 0.444 |
| Random | $0.420 \pm 0.063$ | $0.384 \pm 0.068$ | $0.200 \pm 0.455$ | $0.444 \pm 0.085$ | $0.169 \pm 0.359$ | 0.307 |

Table 22: Acquisition performance on MIMIC Symile for AUPRC (Image imputed by Lab), showing $G_{\text{full}}$. Strategies are grouped by category. Best strategy among proposed and baseline methods in bold for each column.

| Strategy | Fracture | Enl. Card. | Consolidation | Atelectasis | Edema | Mean |
|---|---|---|---|---|---|---|
| *Upper Bounds (for reference)* | | | | | | |
| Oracle | $3.844 \pm 2.595$ | $1.792 \pm 0.288$ | $2.826 \pm 0.677$ | $1.930 \pm 0.221$ | $1.778 \pm 0.101$ | 4.862 |
| True KL-Div. | $0.869 \pm 0.023$ | $0.827 \pm 0.021$ | $0.620 \pm 0.156$ | $0.812 \pm 0.024$ | $0.775 \pm 0.013$ | 0.770 |
| True Rank | $1.398 \pm 0.874$ | $0.566 \pm 0.140$ | $0.492 \pm 0.345$ | $0.827 \pm 0.034$ | $0.759 \pm 0.024$ | 0.575 |
| True Uncert. | $0.712 \pm 0.210$ | $0.650 \pm 0.111$ | $0.655 \pm 0.118$ | $0.538 \pm 0.165$ | $0.579 \pm 0.041$ | 0.498 |
| *Imputation-based (proposed)* | | | | | | |
| KL-Div | $-1.450 \pm 1.121$ | $0.464 \pm 0.132$ | $0.149 \pm 0.370$ | $\mathbf{0.652 \pm 0.149}$ | $0.610 \pm 0.010$ | 0.131 |
| Probability | $\mathbf{0.210 \pm 0.451}$ | $0.575 \pm 0.018$ | $0.735 \pm 0.084$ | $0.591 \pm 0.035$ | $0.598 \pm 0.016$ | 0.568 |
| Rank | $-0.777 \pm 1.263$ | $0.261 \pm 0.185$ | $0.548 \pm 0.051$ | $0.438 \pm 0.134$ | $0.480 \pm 0.042$ | 0.290 |
| Uncertainty | $0.157 \pm 0.477$ | $0.456 \pm 0.075$ | $0.516 \pm 0.245$ | $0.477 \pm 0.170$ | $0.358 \pm 0.021$ | 0.441 |
| *Baselines (no imputation)* | | | | | | |
| Uncertainty | $0.132 \pm 0.484$ | $0.369 \pm 0.094$ | $0.199 \pm 0.387$ | $0.466 \pm 0.152$ | $0.368 \pm 0.022$ | 0.401 |
| Probability | $0.132 \pm 0.484$ | $\mathbf{0.592 \pm 0.029}$ | $\mathbf{0.754 \pm 0.084}$ | $0.632 \pm 0.071$ | $\mathbf{0.612 \pm 0.012}$ | **0.621** |
| Random | $-0.110 \pm 0.296$ | $0.408 \pm 0.060$ | $0.420 \pm 0.208$ | $0.599 \pm 0.156$ | $0.489 \pm 0.037$ | 0.440 |

| Strategy | Cardiomegaly | Lung Lesion | Lung Opacity | Pneumonia | Pneumothorax | Mean |
|---|---|---|---|---|---|---|
| *Upper Bounds (for reference)* | | | | | | |
| Oracle | $1.749 \pm 0.092$ | $2.456 \pm 1.140$ | $7.505 \pm 3.891$ | $4.470 \pm 1.249$ | $20.270 \pm 6.707$ | 4.862 |
| True KL-Div. | $0.774 \pm 0.017$ | $1.035 \pm 0.214$ | $0.413 \pm 0.176$ | $0.793 \pm 0.046$ | $0.781 \pm 0.044$ | 0.770 |
| True Rank | $0.667 \pm 0.037$ | $0.888 \pm 0.274$ | $0.326 \pm 0.248$ | $0.669 \pm 0.058$ | $-0.837 \pm 0.865$ | 0.575 |
| True Uncert. | $0.269 \pm 0.134$ | $0.598 \pm 0.430$ | $0.291 \pm 0.099$ | $0.562 \pm 0.027$ | $0.129 \pm 0.407$ | 0.498 |
| *Imputation-based (proposed)* | | | | | | |
| KL-Div | $\mathbf{0.507 \pm 0.059}$ | $\mathbf{0.742 \pm 0.101}$ | $0.382 \pm 0.251$ | $0.639 \pm 0.066$ | $-1.388 \pm 1.198$ | 0.131 |
| Probability | $0.327 \pm 0.126$ | $0.398 \pm 0.274$ | $0.407 \pm 0.140$ | $0.659 \pm 0.129$ | $1.176 \pm 0.314$ | 0.568 |
| Rank | $0.390 \pm 0.068$ | $0.069 \pm 0.552$ | $\mathbf{1.071 \pm 0.611}$ | $0.469 \pm 0.059$ | $-0.047 \pm 0.555$ | 0.290 |
| Uncertainty | $0.321 \pm 0.042$ | $0.317 \pm 0.186$ | $0.176 \pm 0.196$ | $0.242 \pm 0.030$ | $\mathbf{1.390 \pm 0.419}$ | 0.441 |
| *Baselines (no imputation)* | | | | | | |
| Uncertainty | $0.343 \pm 0.033$ | $0.372 \pm 0.120$ | $0.142 \pm 0.320$ | $0.265 \pm 0.044$ | $1.352 \pm 0.410$ | 0.401 |
| Probability | $0.452 \pm 0.103$ | $0.528 \pm 0.184$ | $0.497 \pm 0.287$ | $\mathbf{0.664 \pm 0.107}$ | $1.352 \pm 0.410$ | **0.621** |
| Random | $0.369 \pm 0.101$ | $0.478 \pm 0.117$ | $0.810 \pm 0.961$ | $0.449 \pm 0.051$ | $0.484 \pm 0.166$ | 0.440 |

Table 23: Acquisition performance on MIMIC Symile for AUROC (Image imputed by ECG), showing $G_{\text{full}}$. Strategies are grouped by category. Best strategy among proposed and baseline methods in bold for each column.

| Strategy | Fracture | Enl. Card. | Consolidation | Atelectasis | Edema | Mean |
|---|---|---|---|---|---|---|
| *Upper Bounds (for reference)* | | | | | | |
| Oracle | $4.028 \pm 2.600$ | $1.959 \pm 0.358$ | $1.960 \pm 0.164$ | $3.526 \pm 0.947$ | $1.639 \pm 0.050$ | 3.892 |
| True KL-Div. | $0.963 \pm 0.224$ | $0.852 \pm 0.051$ | $0.725 \pm 0.030$ | $0.614 \pm 0.152$ | $0.777 \pm 0.003$ | 0.874 |
| True Rank | $0.192 \pm 0.075$ | $0.559 \pm 0.101$ | $0.753 \pm 0.013$ | $-0.784 \pm 1.091$ | $0.719 \pm 0.043$ | 0.327 |
| True Uncert. | $0.934 \pm 0.189$ | $0.700 \pm 0.131$ | $0.717 \pm 0.039$ | $-0.853 \pm 0.584$ | $0.658 \pm 0.050$ | 0.024 |
| *Imputation-based (proposed)* | | | | | | |
| KL-Div | $0.425 \pm 0.323$ | $\mathbf{0.561 \pm 0.079}$ | $\mathbf{0.769 \pm 0.096}$ | $-0.353 \pm 0.827$ | $\mathbf{0.621 \pm 0.030}$ | $\mathbf{0.669}$ |
| Probability | $0.650 \pm 0.024$ | $0.538 \pm 0.107$ | $0.709 \pm 0.095$ | $0.162 \pm 0.130$ | $0.591 \pm 0.024$ | 0.443 |
| Rank | $0.411 \pm 0.193$ | $0.337 \pm 0.122$ | $0.538 \pm 0.050$ | $-0.165 \pm 0.412$ | $0.445 \pm 0.038$ | 0.380 |
| Uncertainty | $\mathbf{0.750 \pm 0.011}$ | $0.527 \pm 0.112$ | $0.720 \pm 0.083$ | $0.466 \pm 0.229$ | $0.431 \pm 0.026$ | 0.167 |
| *Baselines (no imputation)* | | | | | | |
| Uncertainty | $-0.962 \pm 1.403$ | $0.380 \pm 0.050$ | $0.567 \pm 0.064$ | $\mathbf{0.591 \pm 0.180}$ | $0.427 \pm 0.022$ | $-0.055$ |
| Probability | $0.133 \pm 0.291$ | $0.442 \pm 0.050$ | $0.500 \pm 0.028$ | $-0.684 \pm 0.967$ | $0.427 \pm 0.022$ | 0.650 |
| Random | $-0.803 \pm 0.529$ | $0.482 \pm 0.038$ | $0.647 \pm 0.019$ | $0.227 \pm 0.307$ | $0.513 \pm 0.018$ | 0.263 |

| Strategy | Cardiomegaly | Lung Lesion | Lung Opacity | Pneumonia | Pneumothorax | Mean |
|---|---|---|---|---|---|---|
| *Upper Bounds (for reference)* | | | | | | |
| Oracle | $2.263 \pm 0.151$ | $9.495 \pm 7.441$ | $4.112 \pm 0.961$ | $2.196 \pm 0.132$ | $7.745 \pm 2.386$ | 3.892 |
| True KL-Div. | $0.777 \pm 0.016$ | $1.758 \pm 1.206$ | $0.881 \pm 0.085$ | $0.725 \pm 0.019$ | $0.664 \pm 0.088$ | 0.874 |
| True Rank | $0.560 \pm 0.050$ | $-0.731 \pm 0.850$ | $0.808 \pm 0.103$ | $0.777 \pm 0.022$ | $0.416 \pm 0.209$ | 0.327 |
| True Uncert. | $0.741 \pm 0.021$ | $-3.450 \pm 3.932$ | $-0.150 \pm 0.160$ | $0.727 \pm 0.010$ | $0.213 \pm 0.412$ | 0.024 |
| *Imputation-based (proposed)* | | | | | | |
| KL-Div | $0.271 \pm 0.065$ | $2.123 \pm 1.791$ | $\mathbf{0.776 \pm 0.125}$ | $0.571 \pm 0.018$ | $0.924 \pm 0.134$ | $\mathbf{0.669}$ |
| Probability | $0.374 \pm 0.034$ | $-0.235 \pm 0.921$ | $0.122 \pm 0.178$ | $0.568 \pm 0.030$ | $\mathbf{0.952 \pm 0.095}$ | 0.443 |
| Rank | $0.201 \pm 0.072$ | $0.764 \pm 0.047$ | $0.272 \pm 0.153$ | $0.639 \pm 0.029$ | $0.363 \pm 0.307$ | 0.380 |
| Uncertainty | $\mathbf{0.382 \pm 0.028}$ | $-2.196 \pm 2.882$ | $0.086 \pm 0.165$ | $0.630 \pm 0.021$ | $-0.129 \pm 0.190$ | 0.167 |
| *Baselines (no imputation)* | | | | | | |
| Uncertainty | $0.307 \pm 0.038$ | $-3.361 \pm 3.790$ | $0.231 \pm 0.070$ | $\mathbf{0.649 \pm 0.017}$ | $0.625 \pm 0.086$ | $-0.055$ |
| Probability | $0.336 \pm 0.018$ | $\mathbf{3.511 \pm 2.928}$ | $0.632 \pm 0.027$ | $0.575 \pm 0.048$ | $0.625 \pm 0.086$ | 0.650 |
| Random | $0.222 \pm 0.044$ | $0.102 \pm 0.610$ | $0.485 \pm 0.131$ | $0.515 \pm 0.036$ | $0.240 \pm 0.153$ | 0.263 |

Table 24: Acquisition performance on MIMIC Symile for AUPRC (Image imputed by ECG), showing $G_{\text{full}}$. Strategies are grouped by category. Best strategy among proposed and baseline methods in bold for each column.

| Strategy | Fracture | Enl. Card. | Consolidation | Atelectasis | Edema | Mean |
|---|---|---|---|---|---|---|
| *Upper Bounds (for reference)* | | | | | | |
| Oracle | $6.343 \pm 5.165$ | $1.593 \pm 0.113$ | $1.950 \pm 0.162$ | $2.785 \pm 0.530$ | $1.832 \pm 0.041$ | 4.085 |
| True KL-Div. | $1.358 \pm 0.553$ | $0.818 \pm 0.060$ | $0.776 \pm 0.039$ | $0.526 \pm 0.206$ | $0.855 \pm 0.004$ | 0.814 |
| True Rank | $-0.157 \pm 0.576$ | $0.643 \pm 0.064$ | $0.808 \pm 0.008$ | $-0.477 \pm 0.824$ | $0.741 \pm 0.031$ | 0.420 |
| True Uncert. | $1.331 \pm 0.520$ | $0.699 \pm 0.124$ | $0.790 \pm 0.026$ | $-0.640 \pm 0.325$ | $0.745 \pm 0.040$ | 0.489 |
| *Imputation-based (proposed)* | | | | | | |
| KL-Div | $-0.784 \pm 1.571$ | $\mathbf{0.613 \pm 0.082}$ | $0.870 \pm 0.127$ | $-0.203 \pm 0.570$ | $\mathbf{0.737 \pm 0.035}$ | 0.434 |
| Probability | $0.498 \pm 0.210$ | $0.552 \pm 0.123$ | $0.868 \pm 0.103$ | $0.136 \pm 0.118$ | $0.722 \pm 0.027$ | **0.590** |
| Rank | $0.116 \pm 0.576$ | $0.468 \pm 0.033$ | $0.652 \pm 0.045$ | $-0.068 \pm 0.285$ | $0.524 \pm 0.027$ | 0.419 |
| Uncertainty | $\mathbf{0.642 \pm 0.147}$ | $0.553 \pm 0.129$ | $\mathbf{0.874 \pm 0.099}$ | $0.297 \pm 0.183$ | $0.553 \pm 0.015$ | 0.462 |
| *Baselines (no imputation)* | | | | | | |
| Uncertainty | $-4.523 \pm 5.002$ | $0.451 \pm 0.068$ | $0.668 \pm 0.070$ | $\mathbf{0.360 \pm 0.105}$ | $0.534 \pm 0.013$ | $-0.049$ |
| Probability | $-0.190 \pm 0.747$ | $0.496 \pm 0.049$ | $0.578 \pm 0.027$ | $-0.626 \pm 0.854$ | $0.534 \pm 0.013$ | 0.382 |
| Random | $-3.009 \pm 2.890$ | $0.530 \pm 0.060$ | $0.715 \pm 0.037$ | $-0.038 \pm 0.461$ | $0.591 \pm 0.012$ | 0.098 |

| Strategy | Cardiomegaly | Lung Lesion | Lung Opacity | Pneumonia | Pneumothorax | Mean |
|---|---|---|---|---|---|---|
| *Upper Bounds (for reference)* | | | | | | |
| Oracle | $2.458 \pm 0.268$ | $2.158 \pm 0.482$ | $2.735 \pm 0.464$ | $3.162 \pm 0.383$ | $15.830 \pm 5.601$ | 4.085 |
| True KL-Div. | $0.827 \pm 0.009$ | $0.679 \pm 0.250$ | $0.711 \pm 0.063$ | $0.727 \pm 0.026$ | $0.861 \pm 0.024$ | 0.814 |
| True Rank | $0.555 \pm 0.041$ | $-0.005 \pm 0.389$ | $0.715 \pm 0.109$ | $0.809 \pm 0.034$ | $0.573 \pm 0.150$ | 0.420 |
| True Uncert. | $0.803 \pm 0.012$ | $0.184 \pm 0.270$ | $-0.069 \pm 0.111$ | $0.723 \pm 0.046$ | $0.326 \pm 0.207$ | 0.489 |
| *Imputation-based (proposed)* | | | | | | |
| KL-Div | $0.288 \pm 0.055$ | $0.468 \pm 0.419$ | $0.466 \pm 0.061$ | $0.595 \pm 0.049$ | $\mathbf{1.285 \pm 0.172}$ | 0.434 |
| Probability | $0.471 \pm 0.038$ | $0.519 \pm 0.046$ | $0.173 \pm 0.148$ | $\mathbf{0.712 \pm 0.048}$ | $1.249 \pm 0.094$ | **0.590** |
| Rank | $0.234 \pm 0.057$ | $0.554 \pm 0.118$ | $0.214 \pm 0.099$ | $0.688 \pm 0.053$ | $0.810 \pm 0.388$ | 0.419 |
| Uncertainty | $\mathbf{0.477 \pm 0.035}$ | $0.224 \pm 0.340$ | $0.181 \pm 0.070$ | $0.625 \pm 0.030$ | $0.192 \pm 0.082$ | 0.462 |
| *Baselines (no imputation)* | | | | | | |
| Uncertainty | $0.304 \pm 0.125$ | $0.033 \pm 0.364$ | $0.225 \pm 0.012$ | $0.659 \pm 0.046$ | $0.801 \pm 0.041$ | $-0.049$ |
| Probability | $0.427 \pm 0.037$ | $\mathbf{0.704 \pm 0.458}$ | $\mathbf{0.524 \pm 0.057}$ | $0.574 \pm 0.072$ | $0.801 \pm 0.041$ | 0.382 |
| Random | $0.227 \pm 0.042$ | $0.518 \pm 0.037$ | $0.362 \pm 0.020$ | $0.505 \pm 0.048$ | $0.584 \pm 0.155$ | 0.098 |

Table 25: Acquisition performance on MIMIC Symile for AUROC (Image imputed by Lab and ECG), showing $G_{\text{full}}$. Strategies are grouped by category. Best strategy among proposed and baseline methods in bold for each column.

| Strategy | Fracture | Enl. Card. | Consolidation | Atelectasis | Edema | Mean |
|---|---|---|---|---|---|---|
| *Upper Bounds (for reference)* | | | | | | |
| Oracle | $2.679 \pm 0.496$ | $3.081 \pm 0.257$ | $2.539 \pm 0.150$ | $5.098 \pm 0.624$ | $2.119 \pm 0.086$ | 3.911 |
| True KL-Div. | $1.006 \pm 0.111$ | $1.105 \pm 0.082$ | $0.880 \pm 0.033$ | $0.703 \pm 0.095$ | $1.017 \pm 0.023$ | 0.922 |
| True Rank | $0.760 \pm 0.142$ | $1.076 \pm 0.073$ | $0.993 \pm 0.052$ | $0.930 \pm 0.136$ | $1.036 \pm 0.023$ | 0.884 |
| True Uncert. | $0.975 \pm 0.140$ | $0.877 \pm 0.083$ | $0.886 \pm 0.065$ | $0.300 \pm 0.043$ | $0.634 \pm 0.018$ | 0.494 |
| *Imputation-based (proposed)* | | | | | | |
| KL-Div | $0.770 \pm 0.122$ | $\mathbf{0.736 \pm 0.077}$ | $0.711 \pm 0.031$ | $0.049 \pm 0.191$ | $0.535 \pm 0.015$ | 0.512 |
| Probability | $\mathbf{0.883 \pm 0.111}$ | $0.563 \pm 0.062$ | $0.620 \pm 0.034$ | $\mathbf{0.706 \pm 0.096}$ | $\mathbf{0.593 \pm 0.010}$ | **0.628** |
| Rank | $0.594 \pm 0.082$ | $0.335 \pm 0.079$ | $0.423 \pm 0.070$ | $0.130 \pm 0.226$ | $0.466 \pm 0.012$ | 0.392 |
| Uncertainty | $0.785 \pm 0.085$ | $0.535 \pm 0.064$ | $\mathbf{0.728 \pm 0.050}$ | $0.545 \pm 0.088$ | $0.556 \pm 0.015$ | 0.505 |
| *Baselines (no imputation)* | | | | | | |
| Uncertainty | $0.514 \pm 0.151$ | $0.495 \pm 0.038$ | $0.529 \pm 0.045$ | $0.466 \pm 0.056$ | $0.545 \pm 0.014$ | 0.474 |
| Probability | $0.265 \pm 0.046$ | $0.463 \pm 0.035$ | $0.461 \pm 0.037$ | $0.268 \pm 0.136$ | $0.401 \pm 0.009$ | 0.421 |
| Random | $0.005 \pm 0.204$ | $0.386 \pm 0.047$ | $0.467 \pm 0.031$ | $0.373 \pm 0.145$ | $0.472 \pm 0.014$ | 0.425 |

| Strategy | Cardiomegaly | Lung Lesion | Lung Opacity | Pneumonia | Pneumothorax | Mean |
|---|---|---|---|---|---|---|
| *Upper Bounds (for reference)* | | | | | | |
| Oracle | $2.552 \pm 0.116$ | $3.249 \pm 0.605$ | $4.964 \pm 0.431$ | $3.736 \pm 0.369$ | $9.096 \pm 1.223$ | 3.911 |
| True KL-Div. | $0.910 \pm 0.030$ | $0.517 \pm 0.243$ | $0.811 \pm 0.052$ | $0.967 \pm 0.061$ | $1.298 \pm 0.121$ | 0.922 |
| True Rank | $1.047 \pm 0.032$ | $0.545 \pm 0.093$ | $0.839 \pm 0.055$ | $1.010 \pm 0.056$ | $0.607 \pm 0.109$ | 0.884 |
| True Uncert. | $0.514 \pm 0.027$ | $0.200 \pm 0.151$ | $0.293 \pm 0.058$ | $0.729 \pm 0.044$ | $-0.468 \pm 0.155$ | 0.494 |
| *Imputation-based (proposed)* | | | | | | |
| KL-Div | $0.374 \pm 0.016$ | $0.403 \pm 0.210$ | $0.457 \pm 0.079$ | $0.518 \pm 0.036$ | $0.565 \pm 0.103$ | 0.512 |
| Probability | $0.453 \pm 0.016$ | $0.680 \pm 0.352$ | $0.391 \pm 0.064$ | $0.443 \pm 0.042$ | $\mathbf{0.950 \pm 0.096}$ | **0.628** |
| Rank | $0.408 \pm 0.017$ | $0.581 \pm 0.211$ | $0.493 \pm 0.063$ | $0.487 \pm 0.054$ | $-0.000 \pm 0.092$ | 0.392 |
| Uncertainty | $\mathbf{0.476 \pm 0.011}$ | $0.289 \pm 0.162$ | $0.366 \pm 0.052$ | $\mathbf{0.599 \pm 0.031}$ | $0.171 \pm 0.078$ | 0.505 |
| *Baselines (no imputation)* | | | | | | |
| Uncertainty | $0.457 \pm 0.013$ | $0.241 \pm 0.261$ | $0.381 \pm 0.040$ | $0.588 \pm 0.023$ | $0.520 \pm 0.055$ | 0.474 |
| Probability | $0.437 \pm 0.011$ | $0.448 \pm 0.230$ | $0.496 \pm 0.057$ | $0.449 \pm 0.027$ | $0.519 \pm 0.055$ | 0.421 |
| Random | $0.418 \pm 0.015$ | $\mathbf{0.739 \pm 0.256}$ | $\mathbf{0.530 \pm 0.100}$ | $0.493 \pm 0.051$ | $0.362 \pm 0.101$ | 0.425 |

Table 26: Acquisition performance on MIMIC Symile for AUPRC (Image imputed by Lab and ECG), showing $G_{\text{full}}$. Strategies are grouped by category. Best strategy among proposed and baseline methods in bold for each column.

| Strategy | Fracture | Enl. Card. | Consolidation | Atelectasis | Edema | Mean |
|---|---|---|---|---|---|---|
| *Upper Bounds (for reference)* | | | | | | |
| Oracle | $3.244 \pm 0.899$ | $2.660 \pm 0.365$ | $3.290 \pm 0.320$ | $3.756 \pm 0.538$ | $2.484 \pm 0.159$ | 4.063 |
| True KL-Div. | $1.111 \pm 0.223$ | $0.983 \pm 1.014$ | $0.998 \pm 0.079$ | $0.674 \pm 0.088$ | $1.014 \pm 0.034$ | 0.939 |
| True Rank | $0.503 \pm 0.538$ | $0.727 \pm 0.084$ | $1.063 \pm 0.108$ | $0.762 \pm 0.088$ | $1.014 \pm 0.036$ | 0.822 |
| True Uncert. | $0.826 \pm 0.166$ | $0.950 \pm 0.103$ | $0.923 \pm 0.060$ | $0.143 \pm 0.041$ | $0.636 \pm 0.029$ | 0.493 |
| *Imputation-based (proposed)* | | | | | | |
| KL-Div | $\mathbf{1.238 \pm 0.634}$ | $\mathbf{0.636 \pm 0.075}$ | $0.754 \pm 0.075$ | $0.330 \pm 0.146$ | $0.467 \pm 0.020$ | 0.585 |
| Probability | $0.821 \pm 0.184$ | $0.581 \pm 0.048$ | $0.714 \pm 0.055$ | $\mathbf{0.478 \pm 0.083}$ | $\mathbf{0.648 \pm 0.012}$ | **0.610** |
| Rank | $1.133 \pm 0.543$ | $0.375 \pm 0.081$ | $0.582 \pm 0.103$ | $0.139 \pm 0.174$ | $0.469 \pm 0.018$ | 0.491 |
| Uncertainty | $1.018 \pm 0.316$ | $0.516 \pm 0.058$ | $\mathbf{0.783 \pm 0.081}$ | $0.285 \pm 0.039$ | $0.505 \pm 0.015$ | 0.500 |
| *Baselines (no imputation)* | | | | | | |
| Uncertainty | $0.843 \pm 0.289$ | $0.410 \pm 0.050$ | $0.562 \pm 0.061$ | $0.275 \pm 0.038$ | $0.521 \pm 0.019$ | 0.467 |
| Probability | $0.196 \pm 0.143$ | $0.567 \pm 0.044$ | $0.520 \pm 0.042$ | $0.442 \pm 0.111$ | $0.572 \pm 0.014$ | 0.515 |
| Random | $-0.073 \pm 0.199$ | $0.397 \pm 0.084$ | $0.556 \pm 0.053$ | $0.411 \pm 0.100$ | $0.470 \pm 0.013$ | 0.406 |

| Strategy | Cardiomegaly | Lung Lesion | Lung Opacity | Pneumonia | Pneumothorax | Mean |
|---|---|---|---|---|---|---|
| *Upper Bounds (for reference)* | | | | | | |
| Oracle | $2.494 \pm 0.181$ | $2.376 \pm 0.472$ | $4.061 \pm 0.416$ | $4.964 \pm 0.520$ | $11.303 \pm 0.974$ | 4.063 |
| True KL-Div. | $0.823 \pm 0.022$ | $0.809 \pm 0.139$ | $0.873 \pm 0.081$ | $1.002 \pm 0.049$ | $1.105 \pm 0.058$ | 0.939 |
| True Rank | $1.012 \pm 0.039$ | $0.573 \pm 0.099$ | $0.952 \pm 0.093$ | $1.027 \pm 0.058$ | $0.591 \pm 0.077$ | 0.822 |
| True Uncert. | $0.394 \pm 0.037$ | $0.201 \pm 0.029$ | $0.184 \pm 0.050$ | $0.624 \pm 0.044$ | $0.054 \pm 0.058$ | 0.493 |
| *Imputation-based (proposed)* | | | | | | |
| KL-Div | $0.346 \pm 0.023$ | $0.560 \pm 0.075$ | $0.524 \pm 0.074$ | $0.441 \pm 0.037$ | $0.551 \pm 0.075$ | 0.585 |
| Probability | $0.448 \pm 0.029$ | $0.569 \pm 0.185$ | $0.404 \pm 0.073$ | $0.481 \pm 0.039$ | $\mathbf{0.959 \pm 0.048}$ | **0.610** |
| Rank | $0.390 \pm 0.023$ | $0.523 \pm 0.091$ | $0.359 \pm 0.088$ | $0.449 \pm 0.034$ | $0.492 \pm 0.045$ | 0.491 |
| Uncertainty | $0.346 \pm 0.019$ | $0.323 \pm 0.061$ | $0.279 \pm 0.041$ | $0.466 \pm 0.034$ | $0.475 \pm 0.062$ | 0.500 |
| *Baselines (no imputation)* | | | | | | |
| Uncertainty | $0.334 \pm 0.018$ | $0.257 \pm 0.049$ | $0.262 \pm 0.035$ | $0.491 \pm 0.029$ | $0.715 \pm 0.033$ | 0.467 |
| Probability | $\mathbf{0.555 \pm 0.015}$ | $0.448 \pm 0.096$ | $\mathbf{0.607 \pm 0.062}$ | $\mathbf{0.528 \pm 0.032}$ | $0.715 \pm 0.033$ | 0.515 |
| Random | $0.415 \pm 0.015$ | $\mathbf{0.575 \pm 0.228}$ | $0.368 \pm 0.069$ | $0.432 \pm 0.057$ | $0.510 \pm 0.090$ | 0.406 |

Table 27: Acquisition performance on MIMIC Symile for AUROC (Lab imputed by Image), showing $G_{\text{full}}$. Strategies are grouped by category. Best strategy among proposed and baseline methods in bold for each column.

| Strategy | Fracture | Enl. Card. | Consolidation | Atelectasis | Edema | Mean |
|---|---|---|---|---|---|---|
| *Upper Bounds (for reference)* | | | | | | |
| Oracle | $2.404 \pm 0.448$ | $5.403 \pm 1.547$ | $2.446 \pm 0.350$ | $4.261$ | $4.715 \pm 0.988$ | $6.306$ |
| True KL-Div. | $0.901 \pm 0.104$ | $0.727 \pm 0.113$ | $0.874 \pm 0.045$ | $0.807$ | $0.670 \pm 0.093$ | $0.634$ |
| True Rank | $0.490 \pm 0.208$ | $0.872 \pm 0.019$ | $0.873 \pm 0.081$ | $1.691$ | $0.627 \pm 0.148$ | $0.604$ |
| True Uncert. | $-0.204 \pm 0.122$ | $0.690 \pm 0.193$ | $0.319 \pm 0.084$ | $-0.511$ | $0.507 \pm 0.100$ | $0.154$ |
| *Imputation-based (proposed)* | | | | | | |
| KL-Div | $0.443 \pm 0.028$ | $0.503 \pm 0.316$ | $0.523 \pm 0.078$ | $\mathbf{0.383}$ | $0.193 \pm 0.090$ | $\mathbf{0.684}$ |
| Probability | $0.087 \pm 0.217$ | $0.428 \pm 0.155$ | $\mathbf{0.574 \pm 0.117}$ | $0.221$ | $0.457 \pm 0.056$ | $0.380$ |
| Rank | $\mathbf{0.848 \pm 0.131}$ | $-0.103 \pm 0.100$ | $0.309 \pm 0.073$ | $-0.390$ | $0.463 \pm 0.125$ | $-0.139$ |
| Uncertainty | $0.074 \pm 0.187$ | $0.593 \pm 0.168$ | $0.415 \pm 0.107$ | $0.278$ | $0.624 \pm 0.029$ | $0.311$ |
| *Baselines (no imputation)* | | | | | | |
| Uncertainty | $0.515 \pm 0.062$ | $\mathbf{0.781 \pm 0.101}$ | $0.394 \pm 0.106$ | $0.310$ | $\mathbf{0.636 \pm 0.041}$ | $0.487$ |
| Probability | $0.105 \pm 0.251$ | $0.444 \pm 0.144$ | $0.567 \pm 0.112$ | $0.267$ | $0.449 \pm 0.055$ | $0.390$ |
| Random | $0.366 \pm 0.519$ | $0.662 \pm 0.085$ | $0.504 \pm 0.111$ | $0.087$ | $0.229 \pm 0.142$ | $0.419$ |

| Strategy | Cardiomegaly | Lung Lesion | Lung Opacity | Pneumonia | Pneumothorax | Mean |
|---|---|---|---|---|---|---|
| *Upper Bounds (for reference)* | | | | | | |
| Oracle | $11.461$ | $9.175 \pm 4.619$ | $10.440 \pm 4.341$ | $4.058 \pm 0.233$ | $8.693 \pm 2.295$ | $6.306$ |
| True KL-Div. | $0.359$ | $0.043 \pm 1.152$ | $0.608 \pm 0.141$ | $0.793 \pm 0.054$ | $0.561 \pm 0.157$ | $0.634$ |
| True Rank | $0.396$ | $0.640 \pm 1.426$ | $-0.637 \pm 0.091$ | $0.797 \pm 0.031$ | $0.296 \pm 0.208$ | $0.604$ |
| True Uncert. | $0.507$ | $1.296 \pm 0.137$ | $-1.510 \pm 0.311$ | $0.320 \pm 0.119$ | $0.128 \pm 0.257$ | $0.154$ |
| *Imputation-based (proposed)* | | | | | | |
| KL-Div | $0.477$ | $\mathbf{3.593 \pm 0.801}$ | $0.055 \pm 1.005$ | $\mathbf{0.630 \pm 0.024}$ | $0.037 \pm 0.291$ | $\mathbf{0.684}$ |
| Probability | $0.551$ | $2.347 \pm 0.438$ | $-2.017 \pm 0.592$ | $0.581 \pm 0.030$ | $\mathbf{0.575 \pm 0.116}$ | $0.380$ |
| Rank | $-0.413$ | $-1.673 \pm 4.164$ | $-0.741 \pm 0.142$ | $0.398 \pm 0.150$ | $-0.084 \pm 0.300$ | $-0.139$ |
| Uncertainty | $\mathbf{0.646}$ | $0.200 \pm 0.174$ | $-0.684 \pm 0.697$ | $0.401 \pm 0.073$ | $0.566 \pm 0.142$ | $0.311$ |
| *Baselines (no imputation)* | | | | | | |
| Uncertainty | $0.277$ | $0.172 \pm 0.139$ | $\mathbf{0.706 \pm 0.448}$ | $0.501 \pm 0.107$ | $0.575 \pm 0.126$ | $0.487$ |
| Probability | $0.471$ | $1.854 \pm 1.407$ | $-1.445 \pm 0.962$ | $0.615 \pm 0.016$ | $0.575 \pm 0.126$ | $0.390$ |
| Random | $-0.872$ | $2.030 \pm 1.179$ | $0.329 \pm 0.460$ | $0.490 \pm 0.124$ | $0.365 \pm 0.137$ | $0.419$ |

Table 28: Acquisition performance on MIMIC Symile for AUPRC (Lab imputed by Image), showing $G_{\text{full}}$. Strategies are grouped by category. Best strategy among proposed and baseline methods in bold for each column.

| Strategy | Fracture | Enl. Card. | Consolidation | Atelectasis | Edema | Mean |
|---|---|---|---|---|---|---|
| *Upper Bounds (for reference)* | | | | | | |
| Oracle | $2.142 \pm 0.360$ | $3.772 \pm 0.624$ | $4.042 \pm 1.747$ | – | $5.560 \pm 1.093$ | 4.751 |
| True KL-Div. | $0.916 \pm 0.056$ | $0.648 \pm 0.079$ | $0.997 \pm 0.160$ | – | $0.568 \pm 0.132$ | 0.699 |
| True Rank | $0.400 \pm 0.206$ | $0.686 \pm 0.095$ | $1.106 \pm 0.342$ | – | $0.603 \pm 0.139$ | 0.590 |
| True Uncert. | $-1.106 \pm 0.446$ | $0.506 \pm 0.100$ | $0.258 \pm 0.136$ | – | $0.477 \pm 0.270$ | 0.153 |
| | | | | | | |
| *Imputation-based (proposed)* | | | | | | |
| KL-Div | $0.499 \pm 0.117$ | $0.481 \pm 0.203$ | $0.413 \pm 0.187$ | – | $0.269 \pm 0.189$ | **0.680** |
| Probability | $-0.381 \pm 0.441$ | $0.618 \pm 0.008$ | $0.339 \pm 0.318$ | – | $\mathbf{0.681 \pm 0.114}$ | 0.442 |
| Rank | $\mathbf{0.824 \pm 0.103}$ | $-0.157 \pm 0.006$ | $-0.044 \pm 0.404$ | – | $0.246 \pm 0.115$ | 0.186 |
| Uncertainty | $-0.429 \pm 0.473$ | $0.415 \pm 0.161$ | $\mathbf{0.541 \pm 0.220}$ | – | $0.570 \pm 0.164$ | 0.124 |
| | | | | | | |
| *Baselines (no imputation)* | | | | | | |
| Uncertainty | $0.455 \pm 0.060$ | $0.484 \pm 0.099$ | $0.537 \pm 0.231$ | – | $0.585 \pm 0.160$ | 0.440 |
| Probability | $-0.362 \pm 0.445$ | $0.629 \pm 0.007$ | $0.325 \pm 0.326$ | – | $0.672 \pm 0.112$ | 0.572 |
| Random | $0.283 \pm 0.513$ | $\mathbf{0.635 \pm 0.001}$ | $0.303 \pm 0.231$ | – | $0.253 \pm 0.115$ | 0.356 |

| Strategy | Cardiomegaly | Lung Lesion | Lung Opacity | Pneumonia | Pneumothorax | Mean |
|---|---|---|---|---|---|---|
| *Upper Bounds (for reference)* | | | | | | |
| Oracle | – | $3.615 \pm 0.753$ | 3.868 | $4.819 \pm 0.587$ | $10.192 \pm 4.607$ | 4.751 |
| True KL-Div. | – | $0.503 \pm 0.458$ | 0.513 | $0.865 \pm 0.112$ | $0.585 \pm 0.087$ | 0.699 |
| True Rank | – | $0.974 \pm 0.661$ | $-0.184$ | $0.709 \pm 0.066$ | $0.426 \pm 0.076$ | 0.590 |
| True Uncert. | – | $0.621 \pm 0.157$ | $-0.406$ | $0.202 \pm 0.069$ | $0.674 \pm 0.151$ | 0.153 |
| | | | | | | |
| *Imputation-based (proposed)* | | | | | | |
| KL-Div | – | $\mathbf{1.905 \pm 0.140}$ | $\mathbf{0.987}$ | $\mathbf{0.723 \pm 0.115}$ | $0.162 \pm 0.123$ | **0.680** |
| Probability | – | $1.133 \pm 0.483$ | $-0.486$ | $0.659 \pm 0.050$ | $\mathbf{0.976 \pm 0.085}$ | 0.442 |
| Rank | – | $0.274 \pm 1.456$ | $-0.263$ | $0.304 \pm 0.238$ | $0.301 \pm 0.113$ | 0.186 |
| Uncertainty | – | $-0.397 \pm 0.271$ | $-0.893$ | $0.261 \pm 0.024$ | $0.925 \pm 0.045$ | 0.124 |
| | | | | | | |
| *Baselines (no imputation)* | | | | | | |
| Uncertainty | – | $0.053 \pm 0.070$ | 0.109 | $0.330 \pm 0.055$ | $0.963 \pm 0.071$ | 0.440 |
| Probability | – | $1.363 \pm 0.660$ | 0.282 | $0.708 \pm 0.061$ | $0.963 \pm 0.071$ | 0.572 |
| Random | – | $0.088 \pm 0.319$ | 0.534 | $0.455 \pm 0.187$ | $0.296 \pm 0.199$ | 0.356 |

Table 29: Acquisition performance on MIMIC Symile for AUROC (Lab imputed by ECG), showing $G_{\text{full}}$. Strategies are grouped by category. Best strategy among proposed and baseline methods in bold for each column.

| Strategy | Fracture | Enl. Card. | Consolidation | Atelectasis | Edema | Mean |
|---|---|---|---|---|---|---|
| *Upper Bounds (for reference)* | | | | | | |
| Oracle | $2.830 \pm 1.006$ | $6.507 \pm 2.813$ | $1.993 \pm 0.252$ | $13.751 \pm 3.349$ | $5.706 \pm 1.964$ | 7.637 |
| True KL-Div. | $0.807 \pm 0.072$ | $1.043 \pm 0.176$ | $0.855 \pm 0.056$ | $0.933 \pm 0.345$ | $0.841 \pm 0.028$ | 0.727 |
| True Rank | $0.454 \pm 0.258$ | $0.880 \pm 0.255$ | $0.756 \pm 0.027$ | $0.462 \pm 0.641$ | $0.398 \pm 0.151$ | 0.290 |
| True Uncert. | $0.665 \pm 0.104$ | $0.273 \pm 0.305$ | $0.688 \pm 0.056$ | $-0.218 \pm 1.149$ | $0.568 \pm 0.041$ | 0.437 |
| *Imputation-based (proposed)* | | | | | | |
| KL-Div | $\mathbf{0.735 \pm 0.130}$ | $\mathbf{1.221 \pm 0.208}$ | $0.670 \pm 0.136$ | $\mathbf{4.739 \pm 1.683}$ | $2.234 \pm 0.684$ | $\mathbf{2.096}$ |
| Probability | $0.569 \pm 0.115$ | $0.126 \pm 0.370$ | $\mathbf{0.775 \pm 0.075}$ | $-4.498 \pm 1.913$ | $\mathbf{2.255 \pm 0.646}$ | $-0.333$ |
| Rank | $0.395 \pm 0.117$ | $0.160 \pm 0.439$ | $0.298 \pm 0.039$ | $1.303 \pm 1.170$ | $-1.119 \pm 0.726$ | 0.619 |
| Uncertainty | $0.612 \pm 0.125$ | $0.499 \pm 0.346$ | $0.773 \pm 0.078$ | $-4.102 \pm 1.614$ | $0.071 \pm 0.175$ | $-0.744$ |
| *Baselines (no imputation)* | | | | | | |
| Uncertainty | $0.421 \pm 0.290$ | $0.544 \pm 0.303$ | $0.456 \pm 0.081$ | $0.431 \pm 0.170$ | $0.324 \pm 0.024$ | 0.574 |
| Probability | $0.069 \pm 0.066$ | $0.372 \pm 0.420$ | $0.505 \pm 0.102$ | $0.161 \pm 0.787$ | $0.324 \pm 0.024$ | 0.521 |
| Random | $0.117 \pm 0.418$ | $0.737 \pm 0.338$ | $0.531 \pm 0.041$ | $-0.400 \pm 1.001$ | $0.107 \pm 0.142$ | 0.189 |

| Strategy | Cardiomegaly | Lung Lesion | Lung Opacity | Pneumonia | Pneumothorax | Mean |
|---|---|---|---|---|---|---|
| *Upper Bounds (for reference)* | | | | | | |
| Oracle | – | $13.135 \pm 7.362$ | $9.638 \pm 2.483$ | $3.075 \pm 0.250$ | $12.098 \pm 2.848$ | 7.637 |
| True KL-Div. | – | $-0.560 \pm 1.556$ | $0.982 \pm 0.050$ | $0.761 \pm 0.031$ | $0.884 \pm 0.076$ | 0.727 |
| True Rank | – | $-2.847 \pm 4.746$ | $1.545 \pm 0.340$ | $0.776 \pm 0.020$ | $0.185 \pm 0.511$ | 0.290 |
| True Uncert. | – | $2.300 \pm 1.585$ | $0.157 \pm 0.299$ | $0.563 \pm 0.120$ | $-1.060 \pm 0.540$ | 0.437 |
| *Imputation-based (proposed)* | | | | | | |
| KL-Div | – | $\mathbf{4.580 \pm 1.600}$ | $\mathbf{2.539 \pm 0.511}$ | $\mathbf{0.646 \pm 0.117}$ | $1.494 \pm 0.303$ | $\mathbf{2.096}$ |
| Probability | – | $-2.903 \pm 3.257$ | $-1.391 \pm 0.383$ | $0.455 \pm 0.124$ | $\mathbf{1.617 \pm 0.337}$ | $-0.333$ |
| Rank | – | $2.305 \pm 0.688$ | $1.567 \pm 0.771$ | $0.642 \pm 0.053$ | $0.017 \pm 0.983$ | 0.619 |
| Uncertainty | – | $-2.336 \pm 1.344$ | $-1.290 \pm 0.341$ | $0.494 \pm 0.140$ | $-1.416 \pm 0.454$ | $-0.744$ |
| *Baselines (no imputation)* | | | | | | |
| Uncertainty | – | $1.397 \pm 1.699$ | $0.190 \pm 0.028$ | $0.541 \pm 0.082$ | $0.867 \pm 0.198$ | 0.574 |
| Probability | – | $0.657 \pm 2.145$ | $1.107 \pm 0.654$ | $0.626 \pm 0.050$ | $0.867 \pm 0.198$ | 0.521 |
| Random | – | $-0.793 \pm 3.159$ | $0.931 \pm 0.196$ | $0.555 \pm 0.085$ | $-0.086 \pm 0.456$ | 0.189 |

Table 30: Acquisition performance on MIMIC Symile for AUPRC (Lab imputed by ECG), showing $G_{\text{full}}$. Strategies are grouped by category. Best strategy among proposed and baseline methods in bold for each column.

| Strategy | Fracture | Enl. Card. | Consolidation | Atelectasis | Edema | Mean |
|---|---|---|---|---|---|---|
| *Upper Bounds (for reference)* | | | | | | |
| Oracle | $4.231 \pm 2.318$ | $4.018 \pm 1.224$ | $2.093 \pm 0.267$ | $6.269 \pm 0.383$ | $5.741 \pm 1.374$ | 6.317 |
| True KL-Div. | $0.924 \pm 0.069$ | $0.825 \pm 0.017$ | $0.883 \pm 0.030$ | $0.717 \pm 0.570$ | $0.897 \pm 0.019$ | 0.972 |
| True Rank | $0.030 \pm 0.698$ | $0.643 \pm 0.103$ | $0.846 \pm 0.041$ | $0.619 \pm 1.260$ | $0.351 \pm 0.208$ | 0.900 |
| True Uncert. | $0.730 \pm 0.109$ | $0.793 \pm 0.209$ | $0.784 \pm 0.034$ | $0.030 \pm 0.615$ | $0.740 \pm 0.024$ | 0.550 |
| *Imputation-based (proposed)* | | | | | | |
| KL-Div | $\mathbf{0.721 \pm 0.166}$ | $\mathbf{1.292 \pm 0.411}$ | $0.780 \pm 0.157$ | $\mathbf{1.533 \pm 0.215}$ | $2.267 \pm 0.487$ | **1.779** |
| Probability | $0.532 \pm 0.179$ | $1.116 \pm 0.598$ | $0.920 \pm 0.086$ | $-0.332 \pm 0.041$ | $\mathbf{2.300 \pm 0.464}$ | 0.751 |
| Rank | $0.092 \pm 0.485$ | $0.353 \pm 0.268$ | $0.372 \pm 0.058$ | $-0.032 \pm 0.027$ | $-0.641 \pm 0.441$ | 0.458 |
| Uncertainty | $0.580 \pm 0.189$ | $1.080 \pm 0.651$ | $\mathbf{0.930 \pm 0.088}$ | $-0.549 \pm 0.088$ | $0.351 \pm 0.131$ | 0.145 |
| *Baselines (no imputation)* | | | | | | |
| Uncertainty | $-1.088 \pm 1.658$ | $0.451 \pm 0.118$ | $0.605 \pm 0.085$ | $0.152 \pm 0.036$ | $0.645 \pm 0.093$ | 0.260 |
| Probability | $-0.282 \pm 0.309$ | $0.672 \pm 0.126$ | $0.594 \pm 0.102$ | $0.769 \pm 0.144$ | $0.645 \pm 0.093$ | 1.112 |
| Random | $-0.458 \pm 0.636$ | $0.631 \pm 0.184$ | $0.659 \pm 0.051$ | $0.925 \pm 0.146$ | $0.264 \pm 0.154$ | 0.556 |

| Strategy | Cardiomegaly | Lung Lesion | Lung Opacity | Pneumonia | Pneumothorax | Mean |
|---|---|---|---|---|---|---|
| *Upper Bounds (for reference)* | | | | | | |
| Oracle | – | $8.610 \pm 6.961$ | $5.951 \pm 0.522$ | $7.534 \pm 2.031$ | $12.402 \pm 2.525$ | 6.317 |
| True KL-Div. | – | $2.009 \pm 1.288$ | $0.811 \pm 0.344$ | $0.772 \pm 0.111$ | $0.910 \pm 0.021$ | 0.972 |
| True Rank | – | $3.099 \pm 2.419$ | $1.137 \pm 0.603$ | $0.812 \pm 0.056$ | $0.561 \pm 0.207$ | 0.900 |
| True Uncert. | – | $1.431 \pm 1.304$ | $-0.150 \pm 0.288$ | $0.555 \pm 0.159$ | $0.032 \pm 0.114$ | 0.550 |
| *Imputation-based (proposed)* | | | | | | |
| KL-Div | – | $\mathbf{5.709 \pm 4.716}$ | $\mathbf{1.662 \pm 0.095}$ | $\mathbf{0.667 \pm 0.138}$ | $1.378 \pm 0.274$ | **1.779** |
| Probability | – | $0.927 \pm 0.497$ | $-0.770 \pm 0.192$ | $0.647 \pm 0.140$ | $\mathbf{1.420 \pm 0.260}$ | 0.751 |
| Rank | – | $1.988 \pm 1.092$ | $0.643 \pm 0.448$ | $0.564 \pm 0.136$ | $0.788 \pm 0.367$ | 0.458 |
| Uncertainty | – | $-0.954 \pm 0.761$ | $-0.714 \pm 0.206$ | $0.591 \pm 0.192$ | $-0.013 \pm 0.125$ | 0.145 |
| *Baselines (no imputation)* | | | | | | |
| Uncertainty | – | $-0.023 \pm 0.028$ | $0.078 \pm 0.073$ | $0.516 \pm 0.137$ | $1.005 \pm 0.151$ | 0.260 |
| Probability | – | $5.083 \pm 4.111$ | $0.941 \pm 0.112$ | $0.581 \pm 0.069$ | $1.005 \pm 0.151$ | 1.112 |
| Random | – | $1.804 \pm 0.871$ | $0.053 \pm 0.419$ | $0.510 \pm 0.130$ | $0.616 \pm 0.097$ | 0.556 |

Table 31: Acquisition performance on MIMIC Symile for AUROC (Lab imputed by Image and ECG), showing $G_{\text{full}}$. Strategies are grouped by category. Best strategy among proposed and baseline methods in bold for each column.

| Strategy | Fracture | Enl. Card. | Consolidation | Atelectasis | Edema | Mean |
|---|---|---|---|---|---|---|
| *Upper Bounds (for reference)* | | | | | | |
| Oracle | $2.416 \pm 0.727$ | $2.670 \pm 0.218$ | $3.090 \pm 0.355$ | $3.712 \pm 0.446$ | $2.174 \pm 0.050$ | 4.233 |
| True KL-Div. | $1.165 \pm 0.267$ | $0.883 \pm 0.024$ | $1.050 \pm 0.058$ | $1.092 \pm 0.080$ | $0.925 \pm 0.006$ | 0.981 |
| True Rank | $1.373 \pm 0.427$ | $0.811 \pm 0.024$ | $1.076 \pm 0.050$ | $0.904 \pm 0.066$ | $0.888 \pm 0.008$ | 0.883 |
| True Uncert. | $1.008 \pm 0.213$ | $0.720 \pm 0.056$ | $0.781 \pm 0.030$ | $-0.092 \pm 0.044$ | $0.675 \pm 0.027$ | 0.494 |
| *Imputation-based (proposed)* | | | | | | |
| KL-Div | $\mathbf{1.088 \pm 0.224}$ | $\mathbf{0.875 \pm 0.022}$ | $0.667 \pm 0.039$ | $\mathbf{1.193 \pm 0.091}$ | $\mathbf{0.885 \pm 0.008}$ | **1.035** |
| Probability | $1.084 \pm 0.241$ | $0.647 \pm 0.048$ | $0.657 \pm 0.039$ | $-0.270 \pm 0.085$ | $0.767 \pm 0.008$ | 0.390 |
| Rank | $-0.190 \pm 0.547$ | $0.273 \pm 0.051$ | $0.415 \pm 0.044$ | $0.448 \pm 0.088$ | $0.192 \pm 0.017$ | 0.350 |
| Uncertainty | $0.981 \pm 0.208$ | $0.718 \pm 0.054$ | $\mathbf{0.736 \pm 0.028}$ | $-0.143 \pm 0.044$ | $0.636 \pm 0.027$ | 0.448 |
| *Baselines (no imputation)* | | | | | | |
| Uncertainty | $0.772 \pm 0.126$ | $0.492 \pm 0.034$ | $0.627 \pm 0.033$ | $0.183 \pm 0.030$ | $0.506 \pm 0.019$ | 0.469 |
| Probability | $0.088 \pm 0.147$ | $0.529 \pm 0.030$ | $0.376 \pm 0.025$ | $0.804 \pm 0.069$ | $0.324 \pm 0.007$ | 0.521 |
| Random | $0.212 \pm 0.278$ | $0.404 \pm 0.030$ | $0.503 \pm 0.030$ | $0.247 \pm 0.113$ | $0.439 \pm 0.014$ | 0.268 |

| Strategy | Cardiomegaly | Lung Lesion | Lung Opacity | Pneumonia | Pneumothorax | Mean |
|---|---|---|---|---|---|---|
| *Upper Bounds (for reference)* | | | | | | |
| Oracle | $2.438 \pm 0.113$ | $6.864 \pm 1.482$ | $6.398 \pm 0.783$ | $4.889 \pm 0.635$ | $7.680 \pm 0.820$ | 4.233 |
| True KL-Div. | $0.887 \pm 0.007$ | $1.245 \pm 0.526$ | $0.863 \pm 0.089$ | $0.809 \pm 0.032$ | $0.892 \pm 0.057$ | 0.981 |
| True Rank | $0.777 \pm 0.012$ | $0.383 \pm 0.609$ | $0.822 \pm 0.095$ | $0.940 \pm 0.026$ | $0.855 \pm 0.073$ | 0.883 |
| True Uncert. | $0.878 \pm 0.008$ | $0.148 \pm 0.198$ | $0.045 \pm 0.084$ | $0.831 \pm 0.047$ | $-0.059 \pm 0.078$ | 0.494 |
| *Imputation-based (proposed)* | | | | | | |
| KL-Div | $\mathbf{0.887 \pm 0.006}$ | $\mathbf{2.529 \pm 0.817}$ | $\mathbf{0.914 \pm 0.102}$ | $0.661 \pm 0.036$ | $0.651 \pm 0.058$ | **1.035** |
| Probability | $0.718 \pm 0.008$ | $-0.467 \pm 0.520$ | $-0.425 \pm 0.153$ | $0.445 \pm 0.063$ | $\mathbf{0.743 \pm 0.056}$ | 0.390 |
| Rank | $0.326 \pm 0.026$ | $0.909 \pm 0.407$ | $0.492 \pm 0.087$ | $0.561 \pm 0.035$ | $0.079 \pm 0.133$ | 0.350 |
| Uncertainty | $0.882 \pm 0.007$ | $-0.351 \pm 0.366$ | $0.040 \pm 0.080$ | $\mathbf{0.838 \pm 0.047}$ | $0.141 \pm 0.068$ | 0.448 |
| *Baselines (no imputation)* | | | | | | |
| Uncertainty | $0.693 \pm 0.009$ | $-0.022 \pm 0.318$ | $0.281 \pm 0.070$ | $0.730 \pm 0.044$ | $0.425 \pm 0.039$ | 0.469 |
| Probability | $0.260 \pm 0.006$ | $1.476 \pm 0.565$ | $0.538 \pm 0.131$ | $0.387 \pm 0.044$ | $0.425 \pm 0.039$ | 0.521 |
| Random | $0.334 \pm 0.016$ | $-0.383 \pm 0.692$ | $0.212 \pm 0.120$ | $0.492 \pm 0.033$ | $0.216 \pm 0.078$ | 0.268 |

Table 32: Acquisition performance on MIMIC Symile for AUPRC (Lab imputed by Image and ECG), showing $G_{\text{full}}$. Strategies are grouped by category. Best strategy among proposed and baseline methods in bold for each column.

| Strategy | Fracture | Enl. Card. | Consolidation | Atelectasis | Edema | Mean |
|---|---|---|---|---|---|---|
| *Upper Bounds (for reference)* | | | | | | |
| Oracle | $1.965 \pm 0.497$ | $2.573 \pm 0.236$ | $3.714 \pm 0.473$ | $2.444 \pm 0.174$ | $2.962 \pm 0.166$ | 3.748 |
| True KL-Div. | $0.961 \pm 0.228$ | $0.918 \pm 0.033$ | $1.042 \pm 0.082$ | $0.985 \pm 0.053$ | $0.954 \pm 0.013$ | 0.878 |
| True Rank | $0.899 \pm 0.233$ | $0.896 \pm 0.037$ | $1.138 \pm 0.112$ | $0.886 \pm 0.051$ | $0.922 \pm 0.019$ | 0.820 |
| True Uncert. | $0.610 \pm 0.076$ | $0.741 \pm 0.066$ | $0.736 \pm 0.075$ | $-0.096 \pm 0.042$ | $0.718 \pm 0.030$ | 0.482 |
| *Imputation-based (proposed)* | | | | | | |
| KL-Div | $\mathbf{0.914 \pm 0.236}$ | $\mathbf{0.921 \pm 0.039}$ | $0.587 \pm 0.056$ | $\mathbf{1.038 \pm 0.046}$ | $\mathbf{0.925 \pm 0.011}$ | **0.835** |
| Probability | $0.811 \pm 0.100$ | $0.619 \pm 0.048$ | $0.677 \pm 0.026$ | $-0.376 \pm 0.097$ | $0.848 \pm 0.014$ | 0.469 |
| Rank | $0.516 \pm 0.085$ | $0.343 \pm 0.047$ | $0.412 \pm 0.046$ | $0.355 \pm 0.050$ | $0.202 \pm 0.019$ | 0.370 |
| Uncertainty | $0.662 \pm 0.086$ | $0.759 \pm 0.070$ | $\mathbf{0.723 \pm 0.084}$ | $-0.131 \pm 0.030$ | $0.691 \pm 0.028$ | 0.482 |
| *Baselines (no imputation)* | | | | | | |
| Uncertainty | $0.573 \pm 0.099$ | $0.564 \pm 0.041$ | $0.601 \pm 0.074$ | $0.152 \pm 0.016$ | $0.527 \pm 0.023$ | 0.485 |
| Probability | $0.265 \pm 0.173$ | $0.475 \pm 0.038$ | $0.424 \pm 0.060$ | $0.712 \pm 0.038$ | $0.344 \pm 0.012$ | 0.518 |
| Random | $0.504 \pm 0.107$ | $0.449 \pm 0.031$ | $0.612 \pm 0.063$ | $0.242 \pm 0.096$ | $0.471 \pm 0.024$ | 0.431 |

| Strategy | Cardiomegaly | Lung Lesion | Lung Opacity | Pneumonia | Pneumothorax | Mean |
|---|---|---|---|---|---|---|
| *Upper Bounds (for reference)* | | | | | | |
| Oracle | $3.070 \pm 0.202$ | $2.036 \pm 0.167$ | $3.622 \pm 0.323$ | $5.602 \pm 0.520$ | $9.490 \pm 1.511$ | 3.748 |
| True KL-Div. | $0.890 \pm 0.014$ | $0.761 \pm 0.095$ | $0.815 \pm 0.055$ | $0.824 \pm 0.051$ | $0.629 \pm 0.127$ | 0.878 |
| True Rank | $0.807 \pm 0.020$ | $0.452 \pm 0.119$ | $0.721 \pm 0.057$ | $0.920 \pm 0.047$ | $0.560 \pm 0.254$ | 0.820 |
| True Uncert. | $0.880 \pm 0.018$ | $0.171 \pm 0.115$ | $0.034 \pm 0.040$ | $0.616 \pm 0.032$ | $0.411 \pm 0.168$ | 0.482 |
| *Imputation-based (proposed)* | | | | | | |
| KL-Div | $\mathbf{0.938 \pm 0.017}$ | $\mathbf{0.964 \pm 0.166}$ | $\mathbf{0.923 \pm 0.067}$ | $\mathbf{0.675 \pm 0.045}$ | $0.466 \pm 0.230$ | **0.835** |
| Probability | $0.756 \pm 0.008$ | $0.182 \pm 0.257$ | $-0.400 \pm 0.112$ | $0.669 \pm 0.034$ | $\mathbf{0.906 \pm 0.035}$ | 0.469 |
| Rank | $0.368 \pm 0.036$ | $0.486 \pm 0.142$ | $0.363 \pm 0.070$ | $0.418 \pm 0.039$ | $0.234 \pm 0.154$ | 0.370 |
| Uncertainty | $0.932 \pm 0.018$ | $0.020 \pm 0.100$ | $-0.017 \pm 0.041$ | $0.606 \pm 0.037$ | $0.573 \pm 0.244$ | 0.482 |
| *Baselines (no imputation)* | | | | | | |
| Uncertainty | $0.720 \pm 0.018$ | $0.158 \pm 0.050$ | $0.219 \pm 0.024$ | $0.560 \pm 0.038$ | $0.781 \pm 0.148$ | 0.485 |
| Probability | $0.266 \pm 0.012$ | $0.860 \pm 0.124$ | $0.576 \pm 0.066$ | $0.479 \pm 0.048$ | $0.781 \pm 0.148$ | 0.518 |
| Random | $0.375 \pm 0.027$ | $0.362 \pm 0.172$ | $0.296 \pm 0.055$ | $0.464 \pm 0.057$ | $0.539 \pm 0.133$ | 0.431 |

Table 33: Acquisition performance on MIMIC Symile for AUROC (ECG imputed by Image), showing $G_{\text{full}}$. Strategies are grouped by category. Best strategy among proposed and baseline methods in bold for each column.

| Strategy | Fracture | Enl. Card. | Consolidation | Atelectasis | Edema | Mean |
|---|---|---|---|---|---|---|
| *Upper Bounds (for reference)* | | | | | | |
| Oracle | $3.489 \pm 1.656$ | $21.015 \pm 5.247$ | – | – | 37.701 | 28.964 |
| True KL-Div. | $0.546 \pm 0.073$ | $0.036 \pm 0.994$ | – | – | 0.028 | 0.032 |
| True Rank | $0.550 \pm 0.379$ | $-2.342 \pm 1.928$ | – | – | $-1.276$ | 0.122 |
| True Uncert. | $0.648 \pm 0.497$ | $-0.099 \pm 0.724$ | – | – | 0.379 | 0.135 |
| *Imputation-based (proposed)* | | | | | | |
| KL-Div | $0.084 \pm 0.465$ | $-1.318 \pm 0.868$ | – | – | **6.736** | **3.317** |
| Probability | $0.542 \pm 0.247$ | $0.096 \pm 0.156$ | – | – | $-0.629$ | 0.507 |
| Rank | $-0.355 \pm 0.092$ | $-1.669 \pm 0.386$ | – | – | 1.017 | 0.062 |
| Uncertainty | $\mathbf{0.672 \pm 0.578}$ | $0.131 \pm 0.351$ | – | – | $-1.407$ | $-0.817$ |
| *Baselines (no imputation)* | | | | | | |
| Uncertainty | $0.637 \pm 0.113$ | $-0.592 \pm 0.877$ | – | – | 0.868 | 0.515 |
| Probability | $0.507 \pm 0.072$ | $0.236 \pm 0.361$ | – | – | $-0.431$ | $-0.192$ |
| Random | $0.440 \pm 0.455$ | $\mathbf{0.389 \pm 1.478}$ | – | – | $-1.424$ | 0.484 |

| Strategy | Cardiomegaly | Lung Lesion | Lung Opacity | Pneumonia | Pneumothorax | Mean |
|---|---|---|---|---|---|---|
| *Upper Bounds (for reference)* | | | | | | |
| Oracle | $10.077 \pm 4.361$ | $2.979 \pm 1.139$ | 7.291 | – | 120.197 | 28.964 |
| True KL-Div. | $0.834 \pm 0.029$ | $0.528 \pm 0.682$ | $-0.013$ | – | $-1.737$ | 0.032 |
| True Rank | $0.645 \pm 0.055$ | $1.070 \pm 0.717$ | 0.008 | – | 2.199 | 0.122 |
| True Uncert. | $0.804 \pm 0.006$ | $-0.572 \pm 0.283$ | $-1.485$ | – | 1.267 | 0.135 |
| *Imputation-based (proposed)* | | | | | | |
| KL-Div | $0.564 \pm 0.077$ | $-0.078 \pm 0.298$ | **0.510** | – | **16.719** | **3.317** |
| Probability | $0.450 \pm 0.024$ | $-0.870 \pm 0.334$ | $-2.693$ | – | 6.656 | 0.507 |
| Rank | $-0.519 \pm 0.529$ | $\mathbf{0.730 \pm 1.632}$ | $-0.053$ | – | 1.286 | 0.062 |
| Uncertainty | $\mathbf{0.762 \pm 0.061}$ | $-0.322 \pm 0.369$ | $-1.467$ | – | $-4.089$ | $-0.817$ |
| *Baselines (no imputation)* | | | | | | |
| Uncertainty | $0.530 \pm 0.132$ | $0.704 \pm 0.031$ | 0.307 | – | 1.149 | 0.515 |
| Probability | $0.371 \pm 0.024$ | $-1.503 \pm 0.841$ | $-1.676$ | – | 1.149 | $-0.192$ |
| Random | $0.379 \pm 0.012$ | $-1.139 \pm 0.014$ | $-0.550$ | – | 5.293 | 0.484 |

Table 34: Acquisition performance on MIMIC Symile for AUPRC (ECG imputed by Image), showing $G_{\text{full}}$. Strategies are grouped by category. Best strategy among proposed and baseline methods in bold for each column.

| Strategy | Fracture | Enl. Card. | Consolidation | Atelectasis | Edema | Mean |
|---|---|---|---|---|---|---|
| *Upper Bounds (for reference)* | | | | | | |
| Oracle | 1.631 | – | $100.646 \pm 71.555$ | – | – | 33.428 |
| True KL-Div. | 0.635 | – | $2.975 \pm 3.762$ | – | – | 1.163 |
| True Rank | 0.943 | – | $-7.003 \pm 1.999$ | – | – | $-1.214$ |
| True Uncert. | $-0.027$ | – | $0.199 \pm 0.327$ | – | – | 0.542 |
| *Imputation-based (proposed)* | | | | | | |
| KL-Div | **0.600** | – | $\mathbf{15.994 \pm 7.400}$ | – | – | **4.721** |
| Probability | 0.075 | – | $-10.442 \pm 1.040$ | – | – | $-2.263$ |
| Rank | $-1.052$ | – | $-0.692 \pm 4.909$ | – | – | $-0.467$ |
| Uncertainty | $-0.225$ | – | $-5.787 \pm 1.743$ | – | – | $-0.999$ |
| *Baselines (no imputation)* | | | | | | |
| Uncertainty | 0.456 | – | $0.652 \pm 0.902$ | – | – | 0.692 |
| Probability | 0.408 | – | $0.503 \pm 0.861$ | – | – | 0.567 |
| Random | $-0.336$ | – | $-7.531 \pm 5.766$ | – | – | $-1.297$ |

| Strategy | Cardiomegaly | Lung Lesion | Lung Opacity | Pneumonia | Pneumothorax | Mean |
|---|---|---|---|---|---|---|
| *Upper Bounds (for reference)* | | | | | | |
| Oracle | 5.642 | – | – | – | $25.793 \pm 2.491$ | 33.428 |
| True KL-Div. | 0.855 | – | – | – | $0.187 \pm 0.187$ | 1.163 |
| True Rank | 0.784 | – | – | – | $0.421 \pm 0.150$ | $-1.214$ |
| True Uncert. | 0.919 | – | – | – | $1.080 \pm 0.443$ | 0.542 |
| *Imputation-based (proposed)* | | | | | | |
| KL-Div | 0.419 | – | – | – | $1.872 \pm 0.563$ | **4.721** |
| Probability | 0.491 | – | – | – | $0.824 \pm 0.321$ | $-2.263$ |
| Rank | 0.340 | – | – | – | $-0.463 \pm 1.077$ | $-0.467$ |
| Uncertainty | **0.880** | – | – | – | $1.136 \pm 0.162$ | $-0.999$ |
| *Baselines (no imputation)* | | | | | | |
| Uncertainty | 0.685 | – | – | – | $0.975 \pm 0.304$ | 0.692 |
| Probability | 0.383 | – | – | – | $0.975 \pm 0.304$ | 0.567 |
| Random | 0.360 | – | – | – | $\mathbf{2.318 \pm 0.166}$ | $-1.297$ |

Table 35: Acquisition performance on MIMIC Symile for AUROC (ECG imputed by Lab), showing $G_{\text{full}}$. Strategies are grouped by category. Best strategy among proposed and baseline methods in bold for each column.

| Strategy | Fracture | Enl. Card. | Consolidation | Atelectasis | Edema | Mean |
|---|---|---|---|---|---|---|
| *Upper Bounds (for reference)* | | | | | | |
| Oracle | $9.961 \pm 1.805$ | $7.528 \pm 1.743$ | $7.425$ | $9.429 \pm 3.404$ | $42.327$ | $14.888$ |
| True KL-Div. | $-0.137 \pm 1.838$ | $0.978 \pm 0.072$ | $0.178$ | $0.356 \pm 0.377$ | $0.288$ | $0.405$ |
| True Rank | $-0.183 \pm 0.795$ | $0.976 \pm 0.600$ | $0.012$ | $0.962 \pm 0.193$ | $-1.285$ | $0.308$ |
| True Uncert. | $0.271 \pm 0.133$ | $0.892 \pm 0.394$ | $1.249$ | $0.377 \pm 0.354$ | $-0.143$ | $0.399$ |
| *Imputation-based (proposed)* | | | | | | |
| KL-Div | $-0.756 \pm 0.967$ | $0.491 \pm 1.880$ | $1.805$ | $\mathbf{1.782 \pm 0.998}$ | $\mathbf{12.001}$ | $\mathbf{2.649}$ |
| Probability | $0.203 \pm 0.341$ | $\mathbf{2.386 \pm 0.784}$ | $1.834$ | $1.222 \pm 0.693$ | $-15.313$ | $-1.990$ |
| Rank | $-2.489 \pm 1.558$ | $0.182 \pm 0.173$ | $-0.177$ | $0.630 \pm 0.303$ | $0.104$ | $-0.312$ |
| Uncertainty | $0.078 \pm 0.296$ | $1.509 \pm 0.221$ | $\mathbf{1.945}$ | $1.538 \pm 1.059$ | $1.277$ | $0.335$ |
| *Baselines (no imputation)* | | | | | | |
| Uncertainty | $0.201 \pm 0.098$ | $0.537 \pm 0.099$ | $0.628$ | $0.023 \pm 0.305$ | $-0.118$ | $0.384$ |
| Probability | $0.201 \pm 0.098$ | $0.474 \pm 0.027$ | $0.760$ | $0.650 \pm 0.152$ | $0.394$ | $0.074$ |
| Random | $\mathbf{1.112 \pm 1.486}$ | $0.492 \pm 0.457$ | $-0.650$ | $0.656 \pm 0.102$ | $-1.051$ | $0.387$ |

| Strategy | Cardiomegaly | Lung Lesion | Lung Opacity | Pneumonia | Pneumothorax | Mean |
|---|---|---|---|---|---|---|
| *Upper Bounds (for reference)* | | | | | | |
| Oracle | $6.604 \pm 1.266$ | $2.091 \pm 0.714$ | $16.140 \pm 11.342$ | $32.694 \pm 13.014$ | $14.683$ | $14.888$ |
| True KL-Div. | $0.800 \pm 0.103$ | $1.316 \pm 0.077$ | $1.403 \pm 0.913$ | $-1.552 \pm 1.821$ | $0.425$ | $0.405$ |
| True Rank | $0.202 \pm 0.192$ | $0.974 \pm 0.041$ | $2.298 \pm 1.714$ | $-0.790 \pm 0.663$ | $-0.083$ | $0.308$ |
| True Uncert. | $0.121 \pm 0.119$ | $-0.012 \pm 0.256$ | $1.245 \pm 0.700$ | $0.285 \pm 0.133$ | $-0.292$ | $0.399$ |
| *Imputation-based (proposed)* | | | | | | |
| KL-Div | $\mathbf{2.764 \pm 0.736}$ | $0.746 \pm 0.429$ | $0.277 \pm 0.403$ | $\mathbf{4.973 \pm 2.248}$ | $\mathbf{2.409}$ | $\mathbf{2.649}$ |
| Probability | $-2.784 \pm 0.997$ | $0.475 \pm 0.982$ | $0.657 \pm 0.299$ | $-10.662 \pm 5.205$ | $2.079$ | $-1.990$ |
| Rank | $0.337 \pm 0.091$ | $0.246 \pm 0.433$ | $\mathbf{1.594 \pm 1.021}$ | $-3.591 \pm 1.582$ | $0.040$ | $-0.312$ |
| Uncertainty | $-1.728 \pm 0.662$ | $0.190 \pm 0.470$ | $1.224 \pm 0.536$ | $-0.974 \pm 2.384$ | $-1.706$ | $0.335$ |
| *Baselines (no imputation)* | | | | | | |
| Uncertainty | $0.397 \pm 0.031$ | $0.312 \pm 0.090$ | $1.060 \pm 0.408$ | $0.287 \pm 0.093$ | $0.512$ | $0.384$ |
| Probability | $-0.015 \pm 0.186$ | $\mathbf{0.989 \pm 0.490}$ | $0.043 \pm 0.189$ | $-3.264 \pm 2.220$ | $0.513$ | $0.074$ |
| Random | $-0.048 \pm 0.167$ | $0.671 \pm 0.542$ | $1.329 \pm 0.512$ | $2.189 \pm 1.764$ | $-0.832$ | $0.387$ |

Table 36: Acquisition performance on MIMIC Symile for AUPRC (ECG imputed by Lab), showing $G_{\text{full}}$. Strategies are grouped by category. Best strategy among proposed and baseline methods in bold for each column.

| Strategy | Fracture | Enl. Card. | Consolidation | Atelectasis | Edema | Mean |
|---|---|---|---|---|---|---|
| *Upper Bounds (for reference)* | | | | | | |
| Oracle | $7.894 \pm 1.207$ | $7.935 \pm 0.606$ | $7.801 \pm 1.620$ | $3.681$ | $13.663 \pm 0.667$ | $9.827$ |
| True KL-Div. | $-0.038 \pm 1.091$ | $0.843 \pm 0.368$ | $-0.530 \pm 0.679$ | $0.175$ | $0.815 \pm 0.157$ | $0.501$ |
| True Rank | $0.172 \pm 0.009$ | $0.532 \pm 0.379$ | $1.309 \pm 0.247$ | $0.635$ | $-0.719 \pm 0.197$ | $0.354$ |
| True Uncert. | $0.480 \pm 0.166$ | $0.645 \pm 0.089$ | $0.762 \pm 0.322$ | $0.205$ | $0.039 \pm 0.013$ | $0.240$ |
| *Imputation-based (proposed)* | | | | | | |
| KL-Div | $-1.014 \pm 0.482$ | $1.979 \pm 1.044$ | $0.596 \pm 0.626$ | $0.331$ | $\mathbf{3.890 \pm 0.009}$ | $\mathbf{1.408}$ |
| Probability | $-0.291 \pm 0.867$ | $\mathbf{2.492 \pm 0.270}$ | $1.173 \pm 0.256$ | $1.229$ | $-3.963 \pm 0.175$ | $-0.348$ |
| Rank | $-3.557 \pm 1.533$ | $0.231 \pm 0.154$ | $-0.653 \pm 0.806$ | $0.206$ | $0.003 \pm 0.267$ | $-0.331$ |
| Uncertainty | $0.079 \pm 0.427$ | $2.413 \pm 0.318$ | $\mathbf{1.197 \pm 0.340}$ | $0.817$ | $-0.565 \pm 0.214$ | $0.324$ |
| *Baselines (no imputation)* | | | | | | |
| Uncertainty | $0.398 \pm 0.120$ | $0.641 \pm 0.487$ | $0.114 \pm 0.434$ | $-0.013$ | $0.097 \pm 0.015$ | $0.265$ |
| Probability | $0.398 \pm 0.120$ | $0.796 \pm 0.109$ | $0.747 \pm 0.126$ | $0.861$ | $0.424 \pm 0.202$ | $0.581$ |
| Random | $\mathbf{0.667 \pm 1.260}$ | $0.750 \pm 0.531$ | $0.520 \pm 0.275$ | $0.946$ | $0.091 \pm 0.283$ | $0.585$ |

| Strategy | Cardiomegaly | Lung Lesion | Lung Opacity | Pneumonia | Pneumothorax | Mean |
|---|---|---|---|---|---|---|
| *Upper Bounds (for reference)* | | | | | | |
| Oracle | $5.680 \pm 1.134$ | $3.483 \pm 1.480$ | $7.752$ | $29.069 \pm 4.831$ | $11.309$ | $9.827$ |
| True KL-Div. | $0.840 \pm 0.151$ | $1.316 \pm 0.130$ | $0.423$ | $0.558 \pm 0.955$ | $0.607$ | $0.501$ |
| True Rank | $0.341 \pm 0.150$ | $1.125 \pm 0.278$ | $0.399$ | $-0.106 \pm 0.270$ | $-0.148$ | $0.354$ |
| True Uncert. | $0.037 \pm 0.123$ | $-0.790 \pm 0.744$ | $0.221$ | $0.157 \pm 0.074$ | $0.648$ | $0.240$ |
| *Imputation-based (proposed)* | | | | | | |
| KL-Div | $\mathbf{2.195 \pm 0.603}$ | $0.565 \pm 0.242$ | $0.762$ | $\mathbf{4.430 \pm 1.624}$ | $0.343$ | $\mathbf{1.408}$ |
| Probability | $-2.197 \pm 0.763$ | $0.887 \pm 0.999$ | $0.932$ | $-4.773 \pm 1.949$ | $\mathbf{1.029}$ | $-0.348$ |
| Rank | $0.218 \pm 0.086$ | $1.061 \pm 1.227$ | $0.155$ | $-1.583 \pm 0.600$ | $0.613$ | $-0.331$ |
| Uncertainty | $-1.266 \pm 0.476$ | $0.223 \pm 0.360$ | $0.606$ | $-0.563 \pm 1.118$ | $0.304$ | $0.324$ |
| *Baselines (no imputation)* | | | | | | |
| Uncertainty | $0.267 \pm 0.041$ | $0.055 \pm 0.237$ | $-0.024$ | $0.178 \pm 0.051$ | $0.936$ | $0.265$ |
| Probability | $0.038 \pm 0.225$ | $\mathbf{1.274 \pm 0.637}$ | $\mathbf{0.945}$ | $-0.603 \pm 1.027$ | $0.936$ | $0.581$ |
| Random | $-0.098 \pm 0.129$ | $1.243 \pm 0.713$ | $0.239$ | $1.417 \pm 0.819$ | $0.076$ | $0.585$ |

Table 37: Acquisition performance on MIMIC Symile for AUROC (ECG imputed by Image and Lab), showing $G_{\text{full}}$. Strategies are grouped by category. Best strategy among proposed and baseline methods in bold for each column.

| Strategy | Fracture | Enl. Card. | Consolidation | Atelectasis | Edema | Mean |
|---|---|---|---|---|---|---|
| *Upper Bounds (for reference)* | | | | | | |
| Oracle | $2.634 \pm 0.819$ | $4.500 \pm 1.001$ | $3.062 \pm 0.288$ | $4.806 \pm 0.645$ | $2.415 \pm 0.090$ | 4.486 |
| True KL-Div. | $1.145 \pm 0.450$ | $1.172 \pm 0.141$ | $0.774 \pm 0.037$ | $0.821 \pm 0.061$ | $0.897 \pm 0.006$ | 0.839 |
| True Rank | $1.120 \pm 0.318$ | $1.107 \pm 0.131$ | $0.858 \pm 0.021$ | $0.776 \pm 0.057$ | $0.916 \pm 0.007$ | 0.836 |
| True Uncert. | $1.047 \pm 0.381$ | $0.787 \pm 0.136$ | $0.632 \pm 0.069$ | $0.338 \pm 0.071$ | $0.675 \pm 0.013$ | 0.537 |
| *Imputation-based (proposed)* | | | | | | |
| KL-Div | $\mathbf{1.120 \pm 0.452}$ | $\mathbf{1.200 \pm 0.150}$ | $\mathbf{0.771 \pm 0.037}$ | $\mathbf{0.812 \pm 0.065}$ | $\mathbf{0.894 \pm 0.006}$ | **0.838** |
| Probability | $0.933 \pm 0.302$ | $0.676 \pm 0.106$ | $0.508 \pm 0.038$ | $0.111 \pm 0.120$ | $0.349 \pm 0.014$ | 0.438 |
| Rank | $0.126 \pm 0.141$ | $0.517 \pm 0.086$ | $0.489 \pm 0.027$ | $0.507 \pm 0.059$ | $0.504 \pm 0.009$ | 0.473 |
| Uncertainty | $1.063 \pm 0.392$ | $0.820 \pm 0.140$ | $0.638 \pm 0.072$ | $0.358 \pm 0.070$ | $0.673 \pm 0.013$ | 0.551 |
| *Baselines (no imputation)* | | | | | | |
| Uncertainty | $0.785 \pm 0.249$ | $0.509 \pm 0.088$ | $0.499 \pm 0.058$ | $0.468 \pm 0.062$ | $0.604 \pm 0.009$ | 0.544 |
| Probability | $0.097 \pm 0.243$ | $0.594 \pm 0.105$ | $0.528 \pm 0.047$ | $0.435 \pm 0.077$ | $0.556 \pm 0.012$ | 0.466 |
| Random | $0.487 \pm 0.212$ | $0.363 \pm 0.126$ | $0.463 \pm 0.043$ | $0.118 \pm 0.177$ | $0.458 \pm 0.013$ | 0.427 |

| Strategy | Cardiomegaly | Lung Lesion | Lung Opacity | Pneumonia | Pneumothorax | Mean |
|---|---|---|---|---|---|---|
| *Upper Bounds (for reference)* | | | | | | |
| Oracle | $2.675 \pm 0.236$ | $3.710 \pm 0.511$ | $7.949 \pm 2.090$ | $4.600 \pm 0.301$ | $8.512 \pm 1.512$ | 4.486 |
| True KL-Div. | $0.897 \pm 0.010$ | $0.549 \pm 0.239$ | $0.504 \pm 0.303$ | $0.902 \pm 0.021$ | $0.724 \pm 0.092$ | 0.839 |
| True Rank | $0.951 \pm 0.011$ | $0.578 \pm 0.154$ | $0.661 \pm 0.122$ | $0.912 \pm 0.025$ | $0.485 \pm 0.086$ | 0.836 |
| True Uncert. | $0.214 \pm 0.031$ | $0.460 \pm 0.273$ | $0.414 \pm 0.121$ | $0.696 \pm 0.035$ | $0.107 \pm 0.073$ | 0.537 |
| *Imputation-based (proposed)* | | | | | | |
| KL-Div | $\mathbf{0.865 \pm 0.013}$ | $0.617 \pm 0.239$ | $0.474 \pm 0.345$ | $\mathbf{0.927 \pm 0.022}$ | $0.704 \pm 0.090$ | **0.838** |
| Probability | $0.188 \pm 0.025$ | $0.518 \pm 0.303$ | $0.125 \pm 0.121$ | $0.178 \pm 0.041$ | $\mathbf{0.789 \pm 0.051}$ | 0.438 |
| Rank | $0.469 \pm 0.018$ | $0.458 \pm 0.105$ | $\mathbf{0.886 \pm 0.236}$ | $0.483 \pm 0.030$ | $0.287 \pm 0.206$ | 0.473 |
| Uncertainty | $0.205 \pm 0.029$ | $0.496 \pm 0.283$ | $0.436 \pm 0.127$ | $0.701 \pm 0.034$ | $0.121 \pm 0.074$ | 0.551 |
| *Baselines (no imputation)* | | | | | | |
| Uncertainty | $0.324 \pm 0.015$ | $0.517 \pm 0.183$ | $0.569 \pm 0.102$ | $0.575 \pm 0.036$ | $0.592 \pm 0.127$ | 0.544 |
| Probability | $0.654 \pm 0.014$ | $0.348 \pm 0.250$ | $0.256 \pm 0.120$ | $0.604 \pm 0.017$ | $0.591 \pm 0.127$ | 0.466 |
| Random | $0.486 \pm 0.023$ | $\mathbf{0.736 \pm 0.238}$ | $0.351 \pm 0.077$ | $0.420 \pm 0.038$ | $0.392 \pm 0.071$ | 0.427 |

Table 38: Acquisition performance on MIMIC Symile for AUPRC (ECG imputed by Image and Lab), showing $G_{\text{full}}$. Strategies are grouped by category. Best strategy among proposed and baseline methods in bold for each column.

| Strategy | Fracture | Enl. Card. | Consolidation | Atelectasis | Edema | Mean |
|---|---|---|---|---|---|---|
| *Upper Bounds (for reference)* | | | | | | |
| Oracle | $1.578 \pm 0.119$ | $3.454 \pm 0.465$ | $3.604 \pm 0.479$ | $3.295 \pm 0.414$ | $2.989 \pm 0.181$ | 3.703 |
| True KL-Div. | $0.634 \pm 0.094$ | $1.070 \pm 0.078$ | $0.807 \pm 0.038$ | $0.702 \pm 0.052$ | $0.904 \pm 0.017$ | 0.814 |
| True Rank | $0.782 \pm 0.060$ | $1.014 \pm 0.057$ | $0.873 \pm 0.054$ | $0.707 \pm 0.076$ | $0.923 \pm 0.019$ | 0.800 |
| True Uncert. | $0.565 \pm 0.081$ | $0.826 \pm 0.085$ | $0.531 \pm 0.060$ | $0.175 \pm 0.056$ | $0.601 \pm 0.018$ | 0.431 |
| *Imputation-based (proposed)* | | | | | | |
| KL-Div | $\mathbf{0.621 \pm 0.092}$ | $\mathbf{1.087 \pm 0.082}$ | $\mathbf{0.815 \pm 0.041}$ | $\mathbf{0.702 \pm 0.061}$ | $\mathbf{0.903 \pm 0.017}$ | $\mathbf{0.812}$ |
| Probability | $0.577 \pm 0.055$ | $0.687 \pm 0.059$ | $0.484 \pm 0.052$ | $-0.037 \pm 0.152$ | $0.382 \pm 0.026$ | 0.336 |
| Rank | $0.182 \pm 0.091$ | $0.481 \pm 0.066$ | $0.422 \pm 0.074$ | $0.514 \pm 0.052$ | $0.469 \pm 0.015$ | 0.430 |
| Uncertainty | $0.572 \pm 0.081$ | $0.851 \pm 0.087$ | $0.533 \pm 0.061$ | $0.188 \pm 0.056$ | $0.602 \pm 0.018$ | 0.441 |
| *Baselines (no imputation)* | | | | | | |
| Uncertainty | $0.456 \pm 0.053$ | $0.596 \pm 0.072$ | $0.377 \pm 0.054$ | $0.287 \pm 0.026$ | $0.521 \pm 0.017$ | 0.444 |
| Probability | $0.435 \pm 0.052$ | $0.502 \pm 0.063$ | $0.684 \pm 0.050$ | $0.499 \pm 0.081$ | $0.660 \pm 0.015$ | 0.585 |
| Random | $0.374 \pm 0.078$ | $0.367 \pm 0.062$ | $0.432 \pm 0.083$ | $-0.072 \pm 0.281$ | $0.430 \pm 0.021$ | 0.347 |

| Strategy | Cardiomegaly | Lung Lesion | Lung Opacity | Pneumonia | Pneumothorax | Mean |
|---|---|---|---|---|---|---|
| *Upper Bounds (for reference)* | | | | | | |
| Oracle | $2.445 \pm 0.148$ | $2.303 \pm 0.239$ | $4.150 \pm 0.421$ | $5.900 \pm 0.609$ | $7.316 \pm 0.974$ | 3.703 |
| True KL-Div. | $0.856 \pm 0.016$ | $0.749 \pm 0.133$ | $0.746 \pm 0.062$ | $0.952 \pm 0.058$ | $0.714 \pm 0.062$ | 0.814 |
| True Rank | $0.938 \pm 0.014$ | $0.690 \pm 0.094$ | $0.702 \pm 0.072$ | $0.914 \pm 0.049$ | $0.461 \pm 0.074$ | 0.800 |
| True Uncert. | $0.161 \pm 0.031$ | $0.098 \pm 0.098$ | $0.321 \pm 0.066$ | $0.598 \pm 0.057$ | $0.438 \pm 0.061$ | 0.431 |
| *Imputation-based (proposed)* | | | | | | |
| KL-Div | $\mathbf{0.832 \pm 0.018}$ | $\mathbf{0.772 \pm 0.124}$ | $\mathbf{0.707 \pm 0.056}$ | $\mathbf{0.978 \pm 0.061}$ | $0.707 \pm 0.067$ | $\mathbf{0.812}$ |
| Probability | $0.137 \pm 0.036$ | $-0.030 \pm 0.151$ | $0.176 \pm 0.123$ | $0.112 \pm 0.108$ | $\mathbf{0.877 \pm 0.029}$ | 0.336 |
| Rank | $0.426 \pm 0.015$ | $0.559 \pm 0.123$ | $0.463 \pm 0.081$ | $0.378 \pm 0.055$ | $0.401 \pm 0.059$ | 0.430 |
| Uncertainty | $0.154 \pm 0.029$ | $0.123 \pm 0.086$ | $0.334 \pm 0.062$ | $0.603 \pm 0.059$ | $0.445 \pm 0.061$ | 0.441 |
| *Baselines (no imputation)* | | | | | | |
| Uncertainty | $0.279 \pm 0.017$ | $0.272 \pm 0.065$ | $0.400 \pm 0.050$ | $0.436 \pm 0.036$ | $0.820 \pm 0.050$ | 0.444 |
| Probability | $0.674 \pm 0.014$ | $0.517 \pm 0.139$ | $0.403 \pm 0.072$ | $0.655 \pm 0.032$ | $0.821 \pm 0.050$ | 0.585 |
| Random | $0.435 \pm 0.016$ | $0.368 \pm 0.178$ | $0.362 \pm 0.119$ | $0.325 \pm 0.048$ | $0.447 \pm 0.078$ | 0.347 |

# I  RESULTS FOR MIMIC HAIM

Table 39: Acquisition performance on MIMIC HAIM for AUROC, showing $G_{\text{full}}$. Strategies are grouped by category. Best strategy among proposed ones and baselines in bold for each column.

| Strategy | Fracture | Enl. Card. | Consolidation | Atelectasis | Edema | Mean |
|---|---|---|---|---|---|---|
| *Upper Bounds (for reference)* | | | | | | |
| Oracle | $5.121 \pm 1.873$ | $6.802 \pm 2.953$ | $4.228 \pm 1.374$ | $5.862 \pm 0.933$ | $3.435 \pm 0.792$ | 4.602 |
| True KL-Div. | $0.842 \pm 0.154$ | $0.558 \pm 0.187$ | $0.673 \pm 0.054$ | $0.518 \pm 0.093$ | $0.708 \pm 0.025$ | 0.608 |
| True Rank | $0.853 \pm 0.107$ | $0.684 \pm 0.056$ | $0.706 \pm 0.034$ | $0.676 \pm 0.087$ | $0.714 \pm 0.041$ | 0.723 |
| True Uncert. | $0.429 \pm 0.264$ | $0.647 \pm 0.107$ | $0.729 \pm 0.099$ | $0.332 \pm 0.116$ | $0.555 \pm 0.057$ | 0.538 |
| *Imputation-based (proposed)* | | | | | | |
| KL-Div | $0.827 \pm 0.213$ | $0.561 \pm 0.142$ | $0.488 \pm 0.053$ | $0.505 \pm 0.109$ | $0.459 \pm 0.074$ | 0.465 |
| Probability | $0.318 \pm 0.280$ | $\mathbf{0.657 \pm 0.105}$ | $0.761 \pm 0.133$ | $0.620 \pm 0.082$ | $0.578 \pm 0.051$ | 0.494 |
| Rank | $0.530 \pm 0.685$ | $-0.048 \pm 0.305$ | $0.311 \pm 0.079$ | $0.429 \pm 0.150$ | $0.485 \pm 0.032$ | 0.391 |
| Uncertainty | $0.022 \pm 0.463$ | $0.647 \pm 0.108$ | $\mathbf{0.880 \pm 0.241}$ | $0.503 \pm 0.123$ | $0.493 \pm 0.056$ | **0.554** |
| *Baselines (no imputation)* | | | | | | |
| Uncertainty | $0.492 \pm 0.209$ | $0.598 \pm 0.038$ | $0.486 \pm 0.061$ | $0.244 \pm 0.163$ | $0.525 \pm 0.032$ | 0.452 |
| Probability | $0.244 \pm 0.302$ | $0.615 \pm 0.116$ | $0.550 \pm 0.045$ | $\mathbf{0.641 \pm 0.065}$ | $\mathbf{0.580 \pm 0.042}$ | 0.457 |
| Random | $\mathbf{1.005 \pm 0.367}$ | $0.143 \pm 0.289$ | $0.470 \pm 0.045$ | $0.546 \pm 0.096$ | $0.510 \pm 0.025$ | 0.526 |

| Strategy | Cardiomegaly | Lung Lesion | Lung Opacity | Pneumonia | Pneumothorax | Mean |
|---|---|---|---|---|---|---|
| *Upper Bounds (for reference)* | | | | | | |
| Oracle | $4.246 \pm 0.416$ | $1.912 \pm 0.205$ | $5.429 \pm 1.539$ | $2.864 \pm 0.716$ | $6.119 \pm 2.055$ | 4.602 |
| True KL-Div. | $0.609 \pm 0.059$ | $0.674 \pm 0.148$ | $0.801 \pm 0.086$ | $0.775 \pm 0.019$ | $-0.083 \pm 0.370$ | 0.608 |
| True Rank | $0.756 \pm 0.027$ | $0.751 \pm 0.166$ | $0.898 \pm 0.051$ | $0.738 \pm 0.022$ | $0.455 \pm 0.155$ | 0.723 |
| True Uncert. | $0.554 \pm 0.047$ | $0.638 \pm 0.131$ | $0.330 \pm 0.127$ | $0.720 \pm 0.019$ | $0.446 \pm 0.110$ | 0.538 |
| *Imputation-based (proposed)* | | | | | | |
| KL-Div | $0.385 \pm 0.058$ | $0.362 \pm 0.161$ | $0.418 \pm 0.109$ | $0.499 \pm 0.046$ | $0.140 \pm 0.290$ | 0.465 |
| Probability | $0.618 \pm 0.041$ | $0.601 \pm 0.048$ | $0.636 \pm 0.066$ | $\mathbf{0.591 \pm 0.023}$ | $-0.435 \pm 0.322$ | 0.494 |
| Rank | $0.305 \pm 0.061$ | $0.422 \pm 0.157$ | $0.306 \pm 0.194$ | $0.519 \pm 0.044$ | $\mathbf{0.653 \pm 0.050}$ | 0.391 |
| Uncertainty | $0.561 \pm 0.042$ | $\mathbf{0.744 \pm 0.146}$ | $0.541 \pm 0.050$ | $0.587 \pm 0.013$ | $0.558 \pm 0.102$ | **0.554** |
| *Baselines (no imputation)* | | | | | | |
| Uncertainty | $0.460 \pm 0.038$ | $0.511 \pm 0.063$ | $0.396 \pm 0.095$ | $0.539 \pm 0.023$ | $0.272 \pm 0.185$ | 0.452 |
| Probability | $\mathbf{0.619 \pm 0.043}$ | $0.508 \pm 0.086$ | $\mathbf{0.754 \pm 0.050}$ | $0.575 \pm 0.025$ | $-0.513 \pm 0.421$ | 0.457 |
| Random | $0.482 \pm 0.022$ | $0.703 \pm 0.129$ | $0.534 \pm 0.038$ | $0.553 \pm 0.009$ | $0.319 \pm 0.136$ | 0.526 |

Table 40: Acquisition performance on MIMIC HAIM for AUPRC, showing $G_{\text{full}}$. Strategies are grouped by category. Best strategy among proposed ones and baselines in bold for each column.

| Strategy | Fracture | Enl. Card. | Consolidation | Atelectasis | Edema | Mean |
|---|---|---|---|---|---|---|
| *Upper Bounds (for reference)* | | | | | | |
| Oracle | $1.490 \pm 0.249$ | $3.181 \pm 0.283$ | $3.084 \pm 0.331$ | $2.738 \pm 0.363$ | $2.782 \pm 0.349$ | 3.087 |
| True KL-Div. | $0.967 \pm 0.098$ | $0.652 \pm 0.076$ | $0.435 \pm 0.167$ | $0.847 \pm 0.065$ | $0.717 \pm 0.030$ | 0.662 |
| True Rank | $1.014 \pm 0.085$ | $0.721 \pm 0.068$ | $0.541 \pm 0.112$ | $0.958 \pm 0.097$ | $0.769 \pm 0.031$ | 0.754 |
| True Uncert. | $0.842 \pm 0.068$ | $0.592 \pm 0.081$ | $0.605 \pm 0.047$ | $0.425 \pm 0.084$ | $0.611 \pm 0.057$ | 0.587 |
| *Imputation-based (proposed)* | | | | | | |
| KL-Div | $0.889 \pm 0.097$ | $0.578 \pm 0.054$ | $0.301 \pm 0.163$ | $0.655 \pm 0.110$ | $0.518 \pm 0.030$ | 0.516 |
| Probability | $0.584 \pm 0.111$ | $0.672 \pm 0.041$ | $\mathbf{0.807 \pm 0.075}$ | $0.480 \pm 0.018$ | $0.754 \pm 0.017$ | 0.686 |
| Rank | $0.890 \pm 0.120$ | $0.337 \pm 0.081$ | $0.247 \pm 0.115$ | $\mathbf{0.706 \pm 0.052}$ | $\mathbf{0.474 \pm 0.023}$ | 0.456 |
| Uncertainty | $0.799 \pm 0.056$ | $0.540 \pm 0.082$ | $0.680 \pm 0.071$ | $0.353 \pm 0.047$ | $0.584 \pm 0.030$ | 0.570 |
| *Baselines (no imputation)* | | | | | | |
| Uncertainty | $0.920 \pm 0.131$ | $0.556 \pm 0.082$ | $0.409 \pm 0.085$ | $0.365 \pm 0.011$ | $0.487 \pm 0.053$ | 0.479 |
| Probability | $0.480 \pm 0.112$ | $\mathbf{0.713 \pm 0.046}$ | $0.755 \pm 0.054$ | $0.589 \pm 0.053$ | $0.752 \pm 0.017$ | **0.701** |
| Random | $\mathbf{0.941 \pm 0.107}$ | $0.404 \pm 0.060$ | $0.455 \pm 0.057$ | $0.452 \pm 0.180$ | $0.546 \pm 0.024$ | 0.561 |

| Strategy | Cardiomegaly | Lung Lesion | Lung Opacity | Pneumonia | Pneumothorax | Mean |
|---|---|---|---|---|---|---|
| *Upper Bounds (for reference)* | | | | | | |
| Oracle | $4.751 \pm 0.276$ | $1.878 \pm 0.237$ | $2.730 \pm 0.219$ | $3.910 \pm 0.892$ | $4.329 \pm 0.483$ | 3.087 |
| True KL-Div. | $0.374 \pm 0.137$ | $0.688 \pm 0.130$ | $0.660 \pm 0.196$ | $0.699 \pm 0.035$ | $0.575 \pm 0.039$ | 0.662 |
| True Rank | $0.636 \pm 0.071$ | $0.725 \pm 0.153$ | $0.861 \pm 0.144$ | $0.683 \pm 0.035$ | $0.630 \pm 0.042$ | 0.754 |
| True Uncert. | $0.599 \pm 0.059$ | $0.430 \pm 0.127$ | $0.505 \pm 0.173$ | $0.732 \pm 0.031$ | $0.530 \pm 0.043$ | 0.587 |
| *Imputation-based (proposed)* | | | | | | |
| KL-Div | $0.248 \pm 0.124$ | $0.573 \pm 0.133$ | $0.413 \pm 0.218$ | $0.399 \pm 0.089$ | $\mathbf{0.588 \pm 0.043}$ | 0.516 |
| Probability | $\mathbf{0.808 \pm 0.057}$ | $0.764 \pm 0.115$ | $0.737 \pm 0.030$ | $\mathbf{0.736 \pm 0.025}$ | $0.518 \pm 0.040$ | 0.686 |
| Rank | $0.167 \pm 0.117$ | $0.216 \pm 0.119$ | $0.494 \pm 0.147$ | $0.490 \pm 0.079$ | $0.538 \pm 0.024$ | 0.456 |
| Uncertainty | $0.641 \pm 0.052$ | $0.450 \pm 0.120$ | $0.561 \pm 0.092$ | $0.621 \pm 0.030$ | $0.468 \pm 0.030$ | 0.570 |
| *Baselines (no imputation)* | | | | | | |
| Uncertainty | $0.372 \pm 0.063$ | $0.274 \pm 0.111$ | $0.359 \pm 0.220$ | $0.496 \pm 0.042$ | $0.550 \pm 0.036$ | 0.479 |
| Probability | $0.795 \pm 0.076$ | $\mathbf{0.806 \pm 0.109}$ | $\mathbf{0.855 \pm 0.043}$ | $0.725 \pm 0.021$ | $0.536 \pm 0.033$ | **0.701** |
| Random | $0.504 \pm 0.042$ | $0.606 \pm 0.151$ | $0.572 \pm 0.086$ | $0.569 \pm 0.022$ | $0.556 \pm 0.022$ | 0.561 |

Table 41: Acquisition performance on MIMIC HAIM for AUROC (Image imputed by Lab), showing $G_{\text{full}}$. Strategies are grouped by category. Best strategy among proposed and baseline methods in bold for each column.

| Strategy | Fracture | Enl. Card. | Consolidation | Atelectasis | Edema | Mean |
|---|---|---|---|---|---|---|
| *Upper Bounds (for reference)* | | | | | | |
| Oracle | $4.447 \pm 3.050$ | $3.381 \pm 1.055$ | $2.055 \pm 0.185$ | $6.068 \pm 1.280$ | $1.710 \pm 0.036$ | 3.810 |
| True KL-Div. | $1.002 \pm 0.142$ | $0.425 \pm 0.184$ | $0.623 \pm 0.061$ | $0.299 \pm 0.113$ | $0.687 \pm 0.022$ | 0.476 |
| True Rank | $0.902 \pm 0.059$ | $0.704 \pm 0.044$ | $0.667 \pm 0.035$ | $0.575 \pm 0.139$ | $0.698 \pm 0.010$ | 0.671 |
| True Uncert. | $0.908 \pm 0.185$ | $0.768 \pm 0.035$ | $0.641 \pm 0.022$ | $0.301 \pm 0.207$ | $0.709 \pm 0.008$ | 0.610 |
| *Imputation-based (proposed)* | | | | | | |
| KL-Div | $0.686 \pm 0.060$ | $0.579 \pm 0.031$ | $0.463 \pm 0.054$ | $0.412 \pm 0.085$ | $0.539 \pm 0.022$ | 0.416 |
| Probability | $0.514 \pm 0.051$ | $\mathbf{0.773 \pm 0.153}$ | $\mathbf{0.560 \pm 0.026}$ | $\mathbf{0.768 \pm 0.113}$ | $0.590 \pm 0.010$ | 0.445 |
| Rank | $\mathbf{1.530 \pm 0.782}$ | $0.045 \pm 0.280$ | $0.418 \pm 0.037$ | $0.203 \pm 0.164$ | $0.450 \pm 0.021$ | 0.497 |
| Uncertainty | $0.710 \pm 0.087$ | $0.730 \pm 0.091$ | $0.548 \pm 0.029$ | $0.597 \pm 0.215$ | $0.587 \pm 0.012$ | $\mathbf{0.625}$ |
| *Baselines (no imputation)* | | | | | | |
| Uncertainty | $0.512 \pm 0.247$ | $0.635 \pm 0.040$ | $0.491 \pm 0.032$ | $0.086 \pm 0.287$ | $0.587 \pm 0.015$ | 0.438 |
| Probability | $0.109 \pm 0.371$ | $0.759 \pm 0.146$ | $0.533 \pm 0.026$ | $0.739 \pm 0.095$ | $\mathbf{0.592 \pm 0.010}$ | 0.392 |
| Random | $1.112 \pm 0.371$ | $0.483 \pm 0.041$ | $0.509 \pm 0.015$ | $0.556 \pm 0.138$ | $0.542 \pm 0.008$ | 0.582 |

| Strategy | Cardiomegaly | Lung Lesion | Lung Opacity | Pneumonia | Pneumothorax | Mean |
|---|---|---|---|---|---|---|
| *Upper Bounds (for reference)* | | | | | | |
| Oracle | $3.257 \pm 0.286$ | $2.139 \pm 0.317$ | $3.951 \pm 0.558$ | $1.968 \pm 0.068$ | $9.124 \pm 3.961$ | 3.810 |
| True KL-Div. | $0.440 \pm 0.063$ | $0.714 \pm 0.312$ | $0.734 \pm 0.096$ | $0.775 \pm 0.021$ | $-0.936 \pm 0.646$ | 0.476 |
| True Rank | $0.671 \pm 0.017$ | $0.808 \pm 0.318$ | $0.898 \pm 0.066$ | $0.711 \pm 0.015$ | $0.081 \pm 0.265$ | 0.671 |
| True Uncert. | $0.558 \pm 0.042$ | $0.625 \pm 0.057$ | $0.456 \pm 0.097$ | $0.734 \pm 0.020$ | $0.400 \pm 0.221$ | 0.610 |
| *Imputation-based (proposed)* | | | | | | |
| KL-Div | $0.321 \pm 0.083$ | $0.458 \pm 0.273$ | $0.342 \pm 0.129$ | $0.552 \pm 0.027$ | $-0.187 \pm 0.576$ | 0.416 |
| Probability | $\mathbf{0.698 \pm 0.059}$ | $0.544 \pm 0.099$ | $0.693 \pm 0.063$ | $\mathbf{0.600 \pm 0.020}$ | $-1.295 \pm 0.522$ | 0.445 |
| Rank | $0.233 \pm 0.093$ | $0.339 \pm 0.349$ | $0.468 \pm 0.090$ | $0.498 \pm 0.017$ | $\mathbf{0.783 \pm 0.075}$ | 0.497 |
| Uncertainty | $0.620 \pm 0.047$ | $0.668 \pm 0.165$ | $0.581 \pm 0.057$ | $0.591 \pm 0.015$ | $0.613 \pm 0.206$ | $\mathbf{0.625}$ |
| *Baselines (no imputation)* | | | | | | |
| Uncertainty | $0.434 \pm 0.052$ | $0.568 \pm 0.074$ | $0.394 \pm 0.109$ | $0.557 \pm 0.023$ | $0.114 \pm 0.373$ | 0.438 |
| Probability | $0.693 \pm 0.057$ | $0.541 \pm 0.162$ | $\mathbf{0.790 \pm 0.054}$ | $0.589 \pm 0.018$ | $-1.427 \pm 0.750$ | 0.392 |
| Random | $0.482 \pm 0.030$ | $\mathbf{0.941 \pm 0.176}$ | $0.518 \pm 0.049$ | $0.558 \pm 0.010$ | $0.119 \pm 0.262$ | 0.582 |

Table 42: Acquisition performance on MIMIC HAIM for AUPRC (Image imputed by Lab), showing $G_{\text{full}}$. Strategies are grouped by category. Best strategy among proposed and baseline methods in bold for each column.

| Strategy | Fracture | Enl. Card. | Consolidation | Atelectasis | Edema | Mean |
|---|---|---|---|---|---|---|
| *Upper Bounds (for reference)* | | | | | | |
| Oracle | $1.490 \pm 0.249$ | $3.177 \pm 0.420$ | $3.038 \pm 0.399$ | $2.586$ | $1.812 \pm 0.037$ | $2.790$ |
| True KL-Div. | $0.967 \pm 0.098$ | $0.523 \pm 0.091$ | $0.372 \pm 0.195$ | $0.717$ | $0.664 \pm 0.033$ | $0.579$ |
| True Rank | $1.014 \pm 0.085$ | $0.632 \pm 0.084$ | $0.498 \pm 0.128$ | $0.773$ | $0.685 \pm 0.019$ | $0.706$ |
| True Uncert. | $0.842 \pm 0.068$ | $0.790 \pm 0.040$ | $0.624 \pm 0.054$ | $0.575$ | $0.776 \pm 0.014$ | $0.660$ |
| *Imputation-based (proposed)* | | | | | | |
| KL-Div | $0.889 \pm 0.097$ | $0.540 \pm 0.066$ | $0.260 \pm 0.194$ | $0.456$ | $0.586 \pm 0.031$ | $0.478$ |
| Probability | $0.584 \pm 0.111$ | $0.674 \pm 0.058$ | $\mathbf{0.796 \pm 0.087}$ | $0.455$ | $0.710 \pm 0.011$ | $0.662$ |
| Rank | $0.890 \pm 0.120$ | $0.288 \pm 0.118$ | $0.229 \pm 0.139$ | $\mathbf{0.650}$ | $0.459 \pm 0.026$ | $0.416$ |
| Uncertainty | $0.799 \pm 0.056$ | $0.683 \pm 0.085$ | $0.700 \pm 0.081$ | $0.263$ | $0.665 \pm 0.022$ | $0.615$ |
| *Baselines (no imputation)* | | | | | | |
| Uncertainty | $0.920 \pm 0.131$ | $0.700 \pm 0.096$ | $0.448 \pm 0.096$ | $0.387$ | $0.651 \pm 0.025$ | $0.553$ |
| Probability | $0.480 \pm 0.112$ | $\mathbf{0.714 \pm 0.071}$ | $0.749 \pm 0.063$ | $0.483$ | $\mathbf{0.715 \pm 0.011}$ | $\mathbf{0.686}$ |
| Random | $\mathbf{0.941 \pm 0.107}$ | $0.429 \pm 0.088$ | $0.464 \pm 0.067$ | $0.590$ | $0.595 \pm 0.012$ | $0.615$ |

| Strategy | Cardiomegaly | Lung Lesion | Lung Opacity | Pneumonia | Pneumothorax | Mean |
|---|---|---|---|---|---|---|
| *Upper Bounds (for reference)* | | | | | | |
| Oracle | $4.700 \pm 0.289$ | $1.725 \pm 0.306$ | $2.830 \pm 0.264$ | $2.510 \pm 0.170$ | $4.037 \pm 0.525$ | $2.790$ |
| True KL-Div. | $0.054 \pm 0.147$ | $0.651 \pm 0.392$ | $0.621 \pm 0.266$ | $0.731 \pm 0.039$ | $0.493 \pm 0.045$ | $0.579$ |
| True Rank | $0.463 \pm 0.068$ | $0.834 \pm 0.425$ | $0.903 \pm 0.193$ | $0.661 \pm 0.030$ | $0.601 \pm 0.047$ | $0.706$ |
| True Uncert. | $0.490 \pm 0.057$ | $0.666 \pm 0.066$ | $0.495 \pm 0.235$ | $0.726 \pm 0.038$ | $0.615 \pm 0.050$ | $0.660$ |
| *Imputation-based (proposed)* | | | | | | |
| KL-Div | $0.064 \pm 0.176$ | $0.384 \pm 0.277$ | $0.385 \pm 0.295$ | $0.540 \pm 0.044$ | $\mathbf{0.680 \pm 0.038}$ | $0.478$ |
| Probability | $0.920 \pm 0.072$ | $0.556 \pm 0.160$ | $0.718 \pm 0.033$ | $0.725 \pm 0.030$ | $0.485 \pm 0.043$ | $0.662$ |
| Rank | $-0.025 \pm 0.160$ | $0.216 \pm 0.282$ | $0.456 \pm 0.198$ | $0.478 \pm 0.034$ | $0.518 \pm 0.025$ | $0.416$ |
| Uncertainty | $0.655 \pm 0.077$ | $0.656 \pm 0.200$ | $0.574 \pm 0.124$ | $0.638 \pm 0.037$ | $0.519 \pm 0.034$ | $0.615$ |
| *Baselines (no imputation)* | | | | | | |
| Uncertainty | $0.293 \pm 0.089$ | $0.529 \pm 0.084$ | $0.416 \pm 0.296$ | $0.544 \pm 0.042$ | $0.643 \pm 0.024$ | $0.553$ |
| Probability | $\mathbf{0.944 \pm 0.092}$ | $0.671 \pm 0.071$ | $\mathbf{0.850 \pm 0.058}$ | $\mathbf{0.732 \pm 0.022}$ | $0.523 \pm 0.031$ | $\mathbf{0.686}$ |
| Random | $0.462 \pm 0.059$ | $\mathbf{0.958 \pm 0.269}$ | $0.576 \pm 0.117$ | $0.580 \pm 0.019$ | $0.553 \pm 0.031$ | $0.615$ |

Table 43: Acquisition performance on MIMIC HAIM for AUROC (Lab imputed by Image), showing $G_{\text{full}}$. Strategies are grouped by category. Best strategy among proposed and baseline methods in bold for each column.

| Strategy | Fracture | Enl. Card. | Consolidation | Atelectasis | Edema | Mean |
|---|---|---|---|---|---|---|
| *Upper Bounds (for reference)* | | | | | | |
| Oracle | $6.244 \pm 0.851$ | $12.503 \pm 7.484$ | $11.469 \pm 3.810$ | $5.604 \pm 1.451$ | $5.899 \pm 1.530$ | 6.851 |
| True KL-Div. | $0.575 \pm 0.311$ | $0.780 \pm 0.402$ | $0.842 \pm 0.049$ | $0.792 \pm 0.088$ | $0.737 \pm 0.054$ | 0.770 |
| True Rank | $0.773 \pm 0.299$ | $0.649 \pm 0.136$ | $0.836 \pm 0.030$ | $0.802 \pm 0.076$ | $0.736 \pm 0.102$ | 0.796 |
| True Uncert. | $-0.370 \pm 0.158$ | $0.445 \pm 0.274$ | $1.020 \pm 0.440$ | $0.371 \pm 0.065$ | $0.334 \pm 0.084$ | 0.416 |
| *Imputation-based (proposed)* | | | | | | |
| KL-Div | $\mathbf{1.063 \pm 0.606}$ | $0.531 \pm 0.397$ | $0.572 \pm 0.156$ | $0.622 \pm 0.223$ | $0.345 \pm 0.175$ | 0.519 |
| Probability | $-0.010 \pm 0.798$ | $0.463 \pm 0.079$ | $1.433 \pm 0.393$ | $0.435 \pm 0.086$ | $0.560 \pm 0.129$ | **0.551** |
| Rank | $-1.136 \pm 0.350$ | $-0.202 \pm 0.707$ | $-0.044 \pm 0.241$ | $\mathbf{0.712 \pm 0.244}$ | $0.535 \pm 0.072$ | 0.180 |
| Uncertainty | $-1.124 \pm 0.961$ | $0.508 \pm 0.248$ | $\mathbf{1.986 \pm 0.840}$ | $0.385 \pm 0.063$ | $0.359 \pm 0.123$ | 0.492 |
| *Baselines (no imputation)* | | | | | | |
| Uncertainty | $0.460 \pm 0.449$ | $\mathbf{0.535 \pm 0.076}$ | $0.467 \pm 0.283$ | $0.441 \pm 0.053$ | $0.438 \pm 0.063$ | 0.458 |
| Probability | $0.470 \pm 0.594$ | $0.375 \pm 0.158$ | $0.608 \pm 0.200$ | $0.519 \pm 0.069$ | $\mathbf{0.563 \pm 0.105}$ | 0.511 |
| Random | $0.826 \pm 0.879$ | $-0.425 \pm 0.748$ | $0.337 \pm 0.193$ | $0.534 \pm 0.143$ | $0.464 \pm 0.056$ | 0.435 |

| Strategy | Cardiomegaly | Lung Lesion | Lung Opacity | Pneumonia | Pneumothorax | Mean |
|---|---|---|---|---|---|---|
| *Upper Bounds (for reference)* | | | | | | |
| Oracle | $5.483 \pm 0.647$ | $1.730 \pm 0.268$ | $9.123 \pm 5.217$ | $7.345 \pm 3.084$ | $3.113 \pm 0.375$ | 6.851 |
| True KL-Div. | $0.821 \pm 0.038$ | $0.641 \pm 0.142$ | $0.969 \pm 0.173$ | $0.774 \pm 0.071$ | $0.771 \pm 0.016$ | 0.770 |
| True Rank | $0.861 \pm 0.023$ | $0.706 \pm 0.194$ | $0.896 \pm 0.088$ | $0.873 \pm 0.035$ | $0.830 \pm 0.017$ | 0.796 |
| True Uncert. | $0.550 \pm 0.096$ | $0.648 \pm 0.244$ | $0.017 \pm 0.360$ | $0.652 \pm 0.013$ | $0.492 \pm 0.038$ | 0.416 |
| *Imputation-based (proposed)* | | | | | | |
| KL-Div | $0.465 \pm 0.074$ | $0.286 \pm 0.213$ | $0.608 \pm 0.194$ | $0.233 \pm 0.147$ | $0.467 \pm 0.028$ | 0.519 |
| Probability | $0.518 \pm 0.032$ | $0.646 \pm 0.037$ | $0.492 \pm 0.164$ | $0.543 \pm 0.113$ | $0.425 \pm 0.017$ | **0.551** |
| Rank | $0.395 \pm 0.065$ | $0.489 \pm 0.114$ | $-0.098 \pm 0.659$ | $\mathbf{0.626 \pm 0.318}$ | $\mathbf{0.522 \pm 0.033}$ | 0.180 |
| Uncertainty | $0.487 \pm 0.070$ | $\mathbf{0.805 \pm 0.241}$ | $0.441 \pm 0.090$ | $0.570 \pm 0.037$ | $0.503 \pm 0.028$ | 0.492 |
| *Baselines (no imputation)* | | | | | | |
| Uncertainty | $0.492 \pm 0.057$ | $0.466 \pm 0.100$ | $0.401 \pm 0.220$ | $0.454 \pm 0.058$ | $0.430 \pm 0.020$ | 0.458 |
| Probability | $\mathbf{0.526 \pm 0.049}$ | $0.482 \pm 0.103$ | $\mathbf{0.664 \pm 0.114}$ | $0.508 \pm 0.151$ | $0.400 \pm 0.018$ | 0.511 |
| Random | $0.482 \pm 0.033$ | $0.512 \pm 0.145$ | $0.575 \pm 0.056$ | $0.530 \pm 0.018$ | $0.519 \pm 0.024$ | 0.435 |

Table 44: Acquisition performance on MIMIC HAIM for AUPRC (Lab imputed by Image), showing $G_{\text{full}}$. Strategies are grouped by category. Best strategy among proposed and baseline methods in bold for each column.

| Strategy | Fracture | Enl. Card. | Consolidation | Atelectasis | Edema | Mean |
|---|---|---|---|---|---|---|
| *Upper Bounds (for reference)* | | | | | | |
| Oracle | – | $3.188 \pm 0.307$ | $3.312 \pm 0.007$ | $2.814 \pm 0.615$ | $4.169 \pm 0.490$ | 4.011 |
| True KL-Div. | – | $0.884 \pm 0.047$ | $0.749 \pm 0.053$ | $0.912 \pm 0.012$ | $0.792 \pm 0.043$ | 0.776 |
| True Rank | – | $0.881 \pm 0.081$ | $0.758 \pm 0.195$ | $1.051 \pm 0.050$ | $0.889 \pm 0.035$ | 0.811 |
| True Uncert. | – | $0.237 \pm 0.075$ | $0.508 \pm 0.005$ | $0.350 \pm 0.067$ | $0.376 \pm 0.071$ | 0.467 |
| *Imputation-based (proposed)* | | | | | | |
| KL-Div | – | $0.647 \pm 0.098$ | $0.508 \pm 0.078$ | $\mathbf{0.755 \pm 0.079}$ | $0.419 \pm 0.032$ | 0.489 |
| Probability | – | $0.668 \pm 0.054$ | $\mathbf{0.858 \pm 0.163}$ | $0.492 \pm 0.022$ | $\mathbf{0.816 \pm 0.022}$ | 0.723 |
| Rank | – | $0.424 \pm 0.082$ | $0.335 \pm 0.044$ | $0.734 \pm 0.076$ | $0.495 \pm 0.042$ | 0.485 |
| Uncertainty | – | $0.282 \pm 0.094$ | $0.580 \pm 0.160$ | $0.398 \pm 0.022$ | $0.469 \pm 0.034$ | 0.461 |
| *Baselines (no imputation)* | | | | | | |
| Uncertainty | – | $0.298 \pm 0.039$ | $0.214 \pm 0.113$ | $0.354 \pm 0.001$ | $0.254 \pm 0.039$ | 0.294 |
| Probability | – | $\mathbf{0.712 \pm 0.031}$ | $0.782 \pm 0.104$ | $0.642 \pm 0.002$ | $0.806 \pm 0.028$ | **0.726** |
| Random | – | $0.361 \pm 0.069$ | $0.406 \pm 0.089$ | $0.383 \pm 0.287$ | $0.478 \pm 0.045$ | 0.472 |

| Strategy | Cardiomegaly | Lung Lesion | Lung Opacity | Pneumonia | Pneumothorax | Mean |
|---|---|---|---|---|---|---|
| *Upper Bounds (for reference)* | | | | | | |
| Oracle | $4.828 \pm 0.574$ | $1.969 \pm 0.352$ | $2.430 \pm 0.426$ | $8.578 \pm 2.467$ | $4.816 \pm 0.981$ | 4.011 |
| True KL-Div. | $0.854 \pm 0.056$ | $0.711 \pm 0.032$ | $0.776 \pm 0.034$ | $0.595 \pm 0.055$ | $0.712 \pm 0.014$ | 0.776 |
| True Rank | $0.895 \pm 0.034$ | $0.660 \pm 0.093$ | $0.735 \pm 0.039$ | $0.756 \pm 0.125$ | $0.678 \pm 0.081$ | 0.811 |
| True Uncert. | $0.762 \pm 0.089$ | $0.289 \pm 0.175$ | $0.537 \pm 0.099$ | $0.752 \pm 0.052$ | $0.389 \pm 0.036$ | 0.467 |
| *Imputation-based (proposed)* | | | | | | |
| KL-Div | $0.524 \pm 0.084$ | $0.687 \pm 0.135$ | $0.495 \pm 0.131$ | $-0.069 \pm 0.185$ | $0.434 \pm 0.051$ | 0.489 |
| Probability | $\mathbf{0.640 \pm 0.029}$ | $\mathbf{0.889 \pm 0.138}$ | $0.794 \pm 0.073$ | $\mathbf{0.773 \pm 0.051}$ | $\mathbf{0.574 \pm 0.078}$ | 0.723 |
| Rank | $0.455 \pm 0.084$ | $0.216 \pm 0.127$ | $0.607 \pm 0.038$ | $0.533 \pm 0.382$ | $0.570 \pm 0.049$ | 0.485 |
| Uncertainty | $0.621 \pm 0.069$ | $0.326 \pm 0.134$ | $0.523 \pm 0.042$ | $0.566 \pm 0.018$ | $0.381 \pm 0.036$ | 0.461 |
| *Baselines (no imputation)* | | | | | | |
| Uncertainty | $0.489 \pm 0.062$ | $0.121 \pm 0.130$ | $0.187 \pm 0.017$ | $0.333 \pm 0.042$ | $0.396 \pm 0.034$ | 0.294 |
| Probability | $0.571 \pm 0.058$ | $0.887 \pm 0.166$ | $\mathbf{0.870 \pm 0.029}$ | $0.703 \pm 0.060$ | $0.559 \pm 0.076$ | **0.726** |
| Random | $0.568 \pm 0.050$ | $0.395 \pm 0.110$ | $0.561 \pm 0.044$ | $0.535 \pm 0.077$ | $0.560 \pm 0.029$ | 0.472 |

