# OpenReview forum: "Cohort-Based Active Modality Acquisition"
_ICLR.cc/2026/Conference — Submitted to ICLR 2026_

### Official Review · Reviewer_zHE8 · 2025-10-15

**Soundness:** 3
**Presentation:** 2
**Contribution:** 3
**Rating:** 6
**Confidence:** 1

**Summary:**

This paper introduces Cohort-based Active Modality Acquisition (CAMA), a novel problem setting that addresses the challenge of strategically acquiring costly additional data modalities for a subset of a cohort at test time, under a fixed budget. Unlike prior work focusing on individual samples or training-time acquisition, CAMA aims to maximize a global performance metric (e.g., AUROC) for the entire cohort. The core of the proposed solution is a family of "imputation-based" acquisition functions that use a generative model (like a DDPM) to predict counterfactual outcomes—simulating how a sample's prediction would change if the missing modality were acquired. By prioritizing samples with the largest expected predictive shift (quantified, for instance, by KL-Divergence), the method intelligently allocates the acquisition budget. Through comprehensive experiments on multiple real-world datasets, including the large-scale UK Biobank with up to 15 modalities, the paper demonstrates that these imputation-based strategies significantly outperform baseline methods like random selection or uncertainty-based sampling.

**Strengths:**

1. Novel and Practical Problem Formulation: The primary strength is the formalization of the CAMA setting. By shifting the focus from individual-level to cohort-level optimization and from training-time to test-time decisions, the paper addresses a highly practical and underexplored real-world problem.
2. Comprehensive and Rigorous Evaluation: The experimental setup is exceptionally thorough. The authors evaluate their framework on four diverse, real-world multimodal datasets, including the large and complex UK Biobank, demonstrating scalability and generalizability.
3. Principled and Effective Acquisition Strategy: The use of generative imputation to estimate counterfactuals is a powerful and theoretically grounded approach.

**Weaknesses:**

1. High Architectural Complexity: The proposed solution requires jointly training a complex discriminative classifier and a sophisticated generative model (e.g., a Diffusion Transformer).

2. Assumption of a Single Target Modality: The current framework is designed around the scenario of acquiring one specific, pre-determined "target" modality (e.g., proteomics in the UKBB example).

3. Dependence on Generative Model Fidelity: The success of the imputation-based strategies is fundamentally tied to the quality of the counterfactuals generated by the imputation model (f_imp). If the generative model fails to produce plausible imputations, the acquisition decisions will be misguided.

**Questions:**

1. High Architectural Complexity: The proposed solution requires jointly training a complex discriminative classifier and a sophisticated generative model (e.g., a Diffusion Transformer).

2. Assumption of a Single Target Modality: The current framework is designed around the scenario of acquiring one specific, pre-determined "target" modality (e.g., proteomics in the UKBB example).

3. Dependence on Generative Model Fidelity: The success of the imputation-based strategies is fundamentally tied to the quality of the counterfactuals generated by the imputation model (f_imp). If the generative model fails to produce plausible imputations, the acquisition decisions will be misguided.

---

> ### Author Response · Authors · 2025-11-14
>
> We thank the reviewer for the evaluation and constructive feedback, to which we are happy to respond.
>
> ---
>
> **W1: Architectural Complexity**
>
> This is a valid point. The architecture is sophisticated, but this complexity is driven by the problem we are solving. Our method is not just classifying. But, we are performing counterfactual estimation, i.e., estimating the impact of acquiring data we don't have. A simple discriminative classifier alone cannot do this. The generative model is the necessary component that learns to predict this counterfactual, which powers our best-performing KL-Divergence acquisition function (AF). Our results (Tables 3-5) show this complexity is justified, as this method consistently and significantly outperforms all simpler, non-generative baselines. We also show that a lighter BC-VAE can be used as a faster, less complex alternative if needed (Table 6).
>
> **W2: Assumption of a Single Target Modality**
>
> The reviewer is correct and essentially summarized our work with this question. We focused the CAMA setting on this "single target modality" scenario. This is not an oversight, but a precise formalization of a common and practical real-world problem (e.g., "Given a limited budget, which patients in the cohort get the one expensive proteomics test?"). We aimed to solve this core problem robustly first. The more complex and valuable challenge of dynamically choosing which modality to acquire from several missing options is a key extension that we explicitly identify as future work in our conclusion. For this future work, one possible approach would be to run our AF in a counterfactual manner for each candidate modality. Specifically, for a given sample, we would:
>
> 1. Use our generative model to simulate the acquisition of each potential missing modality.
> 2. Calculate the AF for each simulation.
> 3. Select the modality with the highest expected gain. To incorporate varying costs, this gain can additionally be normalized to prioritize the most cost-effective acquisitions.
>
> **W3: Dependence on the Quality of Imputations**
>
> The quality of imputation is indeed a crucial consideration. However, our framework incorporates several design principles that provide robustness against imputation errors:
>
> First, our approach does not require perfect reconstruction but rather aims to model a distribution of plausible outcomes. The acquisition decision leverages the expected impact averaged across this distribution, making it inherently robust to uncertainty in any single imputation.
>
> Second, our primary AF based on KL-Divergence is naturally resilient to imputation noise. It prioritizes samples where the imputed modality is expected to cause the largest shift in downstream predictions, effectively filtering out minor imputation errors while focusing on high-impact signals.
>
> Third, our empirical validation confirms this robustness. The KL-Divergence strategy consistently outperforms strong baselines across multiple datasets and generative models. Notably, our experiments with the BC-VAE demonstrate that the superiority of our AFs remains stable even when using different imperfect generative models, confirming that the core principles of CAMA are robust to the choice of the imputation method.

---

> > ### Author Response · Authors · 2025-11-28
> >
> > Dear Reviewer zHE8,
> >
> > We are writing to follow up on our response regarding the architectural complexity and the scope of our framework. We hope our rebuttal clarified that:
> >
> > 1. **Complexity is functional:** The generative component is necessary to enable counterfactual estimation, which allows our method to significantly outperform simpler discriminative baselines (as shown in Tables 3-5).
> > 2. **Scope is intentional:** The single target setting is a specific formalization of a common high-value problem (e.g., acquiring one expensive test), though we have outlined how this extends to dynamic selection in future work.
> > 3. **Robustness:** Our results with BC-VAEs and the nature of the KL-Divergence metric demonstrate that the method remains effective even with imperfect generative fidelity. During the rebuttal, we also added an ablation Table highlighting that not only generative model fidelity is important but also the design of the whole end-to-end architecture (Tab. 7).
> >
> > As the discussion period is drawing to a close, we would value any further feedback you might have or confirmation that your concerns have been resolved.
> >
> > *Table 7*: Cross-validated ablation of the proposed model adjustments on the Symile dataset, exemplary for mean across all endpoints with the expected KL-Div. AF and acquisitions by AUROC.
> >
> > | **Ablation** | **$G_{full} \uparrow$** |
> > | --- | --- |
> > | proposed model (w.r.t. Tab. 3) | 0.833 |
> > | *w/o* Layer Norm on the encoder’s output | 0.772 |
> > | *w/o* label smoothing | 0.746 |
> > | *w/o* decoupled data flow | 0.599 |
> > | *w/o* balanced training dataset | 0.568 |

---

### Official Review · Reviewer_rRxq · 2025-10-28

**Soundness:** 2
**Presentation:** 3
**Contribution:** 2
**Rating:** 2
**Confidence:** 3

**Summary:**

The paper proposes a new active data acquisition setting called cohort-based active modality acquisition (CAMA), the goal of which is to choose for which samples from some population we should acquire the values of missing modalities in order to maximise some performance metric. The authors propose a set of acquisition functions designed for the CAMA setting that can allow to guide the acquisition. The authors define both oracle metrics, relying on unavailable quantities, and simple baselines such as random acquisition as baselines. To handle the missing modalities in the data during inference, the authors use an architecture based on late fusion models and denoising diffusion probabilities models (DDPMs). With these models in place, the proposed acquisition functions are compared and evaluated on a set of real-world multimodal datasets.

**Strengths:**

- Very clear and extensive related works.
- The authors propose and discuss a comprehensive set of baseline acquisition functions (both oracle and random ones), which allow to put the performance of the proposed metrics in context.
- The experiments are run on several datasets, and over multiple runs, making the results significant.

**Weaknesses:**

**CAMA setting.**
1. The objective in the CAMA setting is not clear to me. The authors mention that the purpose of CAMA is to `identify those samples for which the acquisition of an additional data modality would be most beneficial`. This is formalised through equation 4. However, it is not clear exactly what “beneficial” means in this context. After the acquisition, the models are not retrained on the newly acquired data (as in active learning). Further, ‘benefits’ are also not measured with respect to improving individual predictions at test time (as in active feature acquisition). Hence, it is not immediate to me what variable CAMA is aiming to gain information about (as it seems to be _neither_ the model parameters, as in AL, _nor_ the outcome variable $Y$, as in AFA). Rather than focusing on gaining information about a specific variable, it seems to me that the “benefits” are measured through improving the target performance metric (e.g. AUROC) on the train/test set $\mathcal{D}$ itself (the distinction between the train and test set is blurry in the paper). While this is indeed a different setting than AL or AFA, I do not see how it translates to measurable gains in real-world settings. It would be great if the authors could clarify why considering such a setting might be of interest to any downstream application.
2. What further makes the setting quite artificial is that the “budget” does not account for the fact that different samples $i$ might require different number of modality acquisitions or that some modalities might be more informative (and also more expensive) than others. Instead, the framework only considers the problem of acquiring _all_ missing modalities or none, making this setting quite simplified.

**Acquisition Functions for CAMA.**
1. The authors claim that their contribution in this work is to propose a “theoretical framework” for developing AFs within the CAMA setting. However, looking at the metrics introduced in Appendix C, it seems to me like the degree of novelty in the derived metrics is limited. In particular, the best performing “Maximum Expected Uncertainty Reduction” and “Expected KL Divergence” seem to be variations of the Expected Information Gain metric (this connection is not discussed in the paper). Meanwhile, some of the other metrics (e.g. Maximum Expected Probability or Expected Rank Change) seem quite arbitrary and not supported theoretically. As such, I do not see theoretical results or insights as a significant contribution of this work.

**Architectures for CAMA.**
1. There is limited formalism offered to help understand the exact design of the proposed architecture, making it more difficult to evaluate it and assess its contribution. In particular, it is unclear to me what are the exact components of the architectures, when each of them are used (training vs inference) and what loss functions are used (in particular, what are the loss functions used for the generative component $f_{imp}$?). Equation 6 does not allow to track how the inputs are processed by the architecture used by the authors. Further, what is meant by the “discriminative components” in l.094 and l. 306? Is “discriminative” used in the GAN sense here?
2. The work seems to be heavily inspired by the architecture proposed in Wang et al. (2023). Compared to Wang et al. (2023), what changes did the authors propose or implement that allowed them to adapt this architecture to the CAMA setting specifically?
3. The contribution sections of the paper mentions that the work establishes upper bounds for CAMA. It is unclear for me what this means, and how it is manifested in the proposed architectures.

**Questions:**

1. During the evaluation in the CAMA setting, how does acquiring the additional modalities for one sample $i$ affect the performance/predictions for a different sample $j$? If there is no interaction between acquiring samples for $i$ and $j$, in what cases is the CAMA setting more interesting/beneficial than active feature acquisition? If there is interaction, how does it differ from active learning?
2. What is a unimodal baseline (l.259)? I assume that this is the baseline with only a single modality for each sample. However, different modalities can have different level of “informativeness” for the outcome of interest, and as such which modality is chosen for the unimodal baseline is a crucial decision that will significantly affect the baseline value. Hence, it seems to me that the unimodal baseline is not well defined.
3. Are the imputation based scores $\{s_{I, k}^{imp}\}$ obtained by taking samples from the generative model? If yes, what value of $K$ was used in the experiments, and how sensitive is the proposed framework to this hyperparameter?
4. What do authors mean by “true scores $s_i^{full}$ and $s_i^{avail}$” (l.291) in the context of their data? I assume this corresponds to the logits (l. 376). If yes, how (if at all) are the “true” logits calculated? Are they used for training?
5. In the evaluation process of their proposed models, I am assuming that the models are trained on the test set (in the cross-validation setting) and then the acquisition process is being run on the test set. Is that correct?

---

> ### Author Response · Authors · 2025-11-14
>
> (Answer Part 1)
>
> We are grateful for the reviewer's time and insightful comments, which have helped us to significantly improve the manuscript.
>
> ---
>
> **W1.1: CAMA Setting**
>
> The reviewer's analysis is correct, and it highlights the precise novelty of our setting for which we are happy to provide more detailed explanations.
>
> As the reviewer correctly pointed out, the "benefit" in CAMA is not retraining a model (Active Learning) or perfecting an individual's prediction (Active Feature Acquisition). The benefit is the direct maximization of a global, cohort-level performance metric (e.g., AUROC), as formalized in Eq. (4).
>
> This setting has a direct real-world application: population-level risk stratification under a budget. Consider a health system with a budget to run 1,000 expensive tests on a 100,000-person cohort. Their goal is not to just help 1,000 individuals, but to produce the most accurate final risk-ranking of the entire 100,000-person cohort. This global ranking (measured by its AUROC) is the "product" used to assign preventative care or enroll patients in clinical trials. CAMA directly answers this question: "Which 1,000 patients must we test to make our final 100,000-person ranking as accurate as possible?" This is a pure test-time problem. The model is trained on a separate train set. CAMA is then applied to a new, unseen test set.
>
> **W1.2: Choice of Acquisition**
>
> We thank the reviewer very much for pointing to this fault in Eq. (3). In fact, we do not acquire all modalities but only a specific one, e.g., the expensive Proteomics modality. The reviewer is right and Eq. (3) is wrong in this context and only correct in case of two modalities. We updated and corrected the paper w.r.t. Eq. (3) and added another bullet point to the list at the beginning of the section for clarity: $s^{\text{acquired}}_{i}$: The acquired score, computed using the sample's available modalities plus the newly acquired modality.
>
> We agree with the reviewer’s feedback: There are many paths for future work (Sec. 8), e.g., with measuring and deciding among a pool of modalities for which samples which modalities should be acquired instead of single specific one.
>
> **W2: Acquisition Functions for CAMA**
>
> This is a very important point that we'd like to clarify. We must correct a small but critical misunderstanding in the premise: our results consistently show that “Expected KL-Divergence” (KL-Div.) is the best-performing strategy, significantly and consistently outperforming "Maximum Expected Uncertainty Reduction". This distinction is key to our theoretical contribution.
>
> 1. On the "Theoretical Framework": The reviewer correctly pointed out that the metrics are based on well known approaches. However, we are claiming to be the first to derive and apply it as an imputation-based acquisition function (AF) for this specific cohort-level setting. Our core contribution is the formalization of the CAMA problem itself, i.e., the novel, test-time, cohort-level optimization of a global metric (Eq. 4). Our "framework" is the application of information-theoretic principles to this new problem.
> 2. On Expected Information Gain (EIG) Connection: The KL-Div. is a pragmatic, model-based proxy for EIG. It is designed to estimate the expected shift in the model's predictive distribution which is a direct measure of information gain. While we cite related work w.r.t. EIG (Long, 2022), we agree that this connection is valuable and currently not clearly stated in the paper. We added this to the paper.
> 3. On "Arbitrary" Metrics: Even if some metrics may appear arbitrary, they are essential ablations. They represent the logical, but flawed, alternative: "estimate the final state". That these metrics consistently fail while KL-Div. succeeds is a key theoretical insight: An effective AF for CAMA must estimate the magnitude of change, not just the final outcome.
>
> In summary, our theoretical contribution is the formalization of the CAMA problem and the subsequent insight, both derived and empirically proven, that a specific type of AF (one that proxies information gain) is the better one for solving it.

---

> ### Author Response · Authors · 2025-11-14
>
> (Answer Part 2)
>
> **W3.1: Explanations for the Model Architecture**
>
> This is a critical set of questions for understanding our architecture. We appreciate the opportunity to clarify these points.
>
> - Meaning of “Discriminative”: We use "discriminative" in the classic machine learning sense, not in the Generative Adversarial Network (GAN) sense. The "Discriminative Components" ($f$) refer to the parts of the model that learn a decision boundary to predict a class ($p(y|x)$). This is our main classifier, which consists of the modality-specific encoders and the fusion head. The  "Generative Component" ($f_{imp}$) refers to the model that learns the data distribution ($p(x_{\text{missing}}|x_{\text{avail}})$) to impute missing data.
> - Architectural Components and Losses (Eq. (6)): We are happy to break down the components of Eq. (6), which are shown in Figure 2(B):
>     - $L_{CE}$ (discriminative loss): This is the Cross-Entropy loss for the classification task. It is computed from the output of the discriminative classifier $f$.
>     - $L_{D_i}$ (generative loss): Our generative model $f_{imp}$ is a Denoising Diffusion Probabilistic Model (DDPM) (Section 5 (l. 306) and Appendix D (Fig. 4)). Therefore, $L_{D_i}$ is the standard DDPM loss (e.g., an MSE on the predicted noise) used to train the generative model to reconstruct the latent embedding for each modality.
> - Training vs. Inference Data Flow:  The distinction is the time in which each component is used, which is the core of our architectural design (Figure 2B):
>     - Training (Fig 2B, left): The two components are trained jointly. However, the classifier $f$ is trained only on real, available data. The imputed latents from the generative model $f_{imp}$ are masked out and are not passed to the classifier $f$. This is a crucial design choice: the classifier learns to be a robust predictor using only real (and often incomplete) data.
>     - Inference (Fig 2B, right): This is the only time the generative model's outputs are fed to the classifier. To calculate our AFs, we need to estimate the counterfactual score $s^{\text{imp}}$. To get this, we run $f_{imp}$ to generate the missing modality's latent embedding, and then pass this imputed embedding (along with the available ones) into the classifier $f$.
>
> **W3.2: Model Comparison to Wang et al.**
>
> The reviewer is right that we adapt the imputation model from Wang et al. (2023). However, their goal was reconstruction (to handle missing data), while our new CAMA setting is a decision-making problem (to decide which data to acquire). This required several critical architectural adaptations:
>
> - Decoupled Data Flow (Fig. 2B):
>     - Training: The classifier ($f$) is trained only on real, available data. Imputed latents are masked out (l. 312).
>     - Inference: The classifier ($f$) sees imputed data only to generate the counterfactual scores ($s^{\text{imp}}$) needed for our AFs.
> - Probabilistic Calibration: Our best-performing AF (KL-Div.) demands reliable probabilities. We added Label Smoothing (l. 346) to the classifier, which is essential for our decision task but not for their reconstruction task.
> - Stability: We added Layer Normalization (l. 345) and training masks (l. 350) to stabilize the architecture for this new, dual-loss objective.
>
> **W3.3: Upper Bounds for CAMA**
>
> This is a great clarifying question. The "upper bounds" are not a new architecture, but benchmark evaluation strategies used to measure the "best-case" performance ceiling.
> They fall into two groups (Sections 4 and C.2/C.3):
>
> 1. Oracles (e.g., Oracle AUROC): Assume perfect knowledge of both the true labels ($y_i$) and the true full scores ($s^{\text{full}}_i$). This shows the best possible greedy acquisition result.
> 2. Upper-Bound Heuristics (e.g., True KL-Div): Assume only perfect knowledge of the true full scores ($s^{\text{full}}_i$) but not the labels. These are evaluated using only the simple classifier ($f$ in Fig. 2A), fed with complete data to get the true $s^{\text{full}}$. They serve as a gold standard to measure our real-world imputation-based methods against. For instance, the "True KL-Div" bound shows how our best AF would perform if our generative imputation method (e.g., DDPMs/BC-VAEs) were perfect.

---

> ### Author Response · Authors · 2025-11-14
>
> (Answer Part 3)
>
> **Q1: Difference to Active Learning**
>
> We are happy to provide answers to this key insight! Essentially, there is no interaction at the model level, but there is a crucial interaction at the metric level. Acquiring a sample does not change the model or the score for any other sample. The model is pre-trained and frozen. However, our objective (Eq. (4)) is a global metric like AUROC, which depends on the pairwise rankings of all samples. Changing $s_i$ (by acquiring its new modality) changes its rank relative to all other samples, which directly interacts with and changes the final global score.
>
> This is what makes the CAMA setting more beneficial than AFA in our target applications: AFA optimizes for an individual's prediction (e.g., "How accurate is $s_i$?"). CAMA optimizes for the cohort's overall performance (e.g., "How accurate is the ranking of all $N$ samples?"). We might acquire for a sample not to perfect its individual prediction, but because changing its score is the most efficient way to improve the entire cohort's AUROC. This differs from AL, which retrains the model to change all other predictions.
>
> **Q2: Unimodal Baselines**
>
> That's a crucial point and we thank the reviewer for pointing this out. The "unimodal baseline" is not a fixed, single modality. It is the starting performance of the entire cohort at 0% acquisition budget. In our experiments, each sample begins with a randomly assigned subset of available modalities and a corresponding score. The "unimodal baseline" is the global metric (e.g., AUROC) calculated using only these initial, heterogeneous $s^{\text{avail}}_{i}$ scores for all samples. It's the "Cost = 0" point from which all acquisition gain is measured. To make this clearer, we updated the paper and renamed “unimodal” to “pre-acquisition” and “multimodal” to “post-acquisition”.
>
> **Q3: Hyperparameter K**
>
> The reviewer correctly pointed out that the imputation-based scores are obtained by taking samples from the generative model. The imputation-based acquisition functions (like our best-performing “Expected KL-Div.”) are defined as the average over $K$ samples from the generative model. In our experiments, we used $K=100$ samples (l. 1110). We did not provide a specific sensitivity analysis for $K$. However, the method is designed to be robust by averaging over this distribution. As discussed (l. 470), by averaging the expected impact across $K=100$ plausible outcomes, the acquisition decision becomes stable and less sensitive to the noise or uncertainty of any single imputation. During development we conducted experiments w.r.t. the sensitivity of K ensuring the imputation quality.
>
> **Q4: True and Available Scores**
>
> The reviewer's assumption is correct and we are pleased to provide more information. As stated in the problem formulation and evaluation, the "scores" ($s^{\text{avail}}_i$ and $s^{\text{full}}_i$) are the logits (the raw, pre-softmax output) of the discriminative classifier. They are calculated by passing the data through the pre-trained and frozen classifier $f$ (the model from Fig. 2A):
>
> - $s^{\text{avail}}_i$ ("true available score"): Is the logit from $f$ when it is fed only the initially available modalities for sample $i$ (i.e., $f(x^{\text{avail}}_i, \theta)$).
> - $s^{\text{full}}_i$ ("true full score"): Is the logit from $f$ when it is fed the complete set of all modalities for sample $i$ (i.e., $f(x^{\text{full}}_i, \theta)$).
>
> These scores are used only at test-time for two purposes, i.e., not for training:
>
> - To evaluate the oracle and upper-bound benchmarks, which have access to $s^{\text{full}}_i$.
> - To simulate the acquisition during evaluation. When a sample is "acquired," we replace its $s^{\text{avail}}_i$ with its $s^{\text{full}}_i$ to measure the impact on the global metric.
>
> **Q5: Evaluation Process**
>
> That is a critical point, and we must clarify that this is not correct. We follow a strict k-fold cross-validation procedure to ensure there is no data leakage. For each fold, the model (both the classifier $f$ and the imputer $f_{imp}$) is trained only on the training folds. The entire CAMA acquisition and evaluation process is then run only on the held-out, unseen test fold. The model is never trained on the test set.
>
> ---
>
> **Summary of Paper Adjustments**
>
> - Sec. 3: corrected Eq. (3)
> - Sec. 4: connection to EIG
> - Full paper: Added clarifications with “pre-acquisition” and “post-acquisition”

---

> > ### Comment · Reviewer_rRxq · 2025-11-17
> >
> > Thank you for all the extensive clarifications -- they have definitely helped to improve my understanding of the work, and thus opened up the opportunities for its more comprehensive and accurate evaluation. In particular, your response to my W1.1 helped to explain the motivation behind the CAMA setting and I have adjusted my score.
> >
> > While the provided explanations helped to improve the clarity of the work, I still think that the paper offers limited contributions to the community. Specifically:
> > 1. Given the clarification of the acquisition setting considered (as explained in response to W1.2) the question that the paper is considering currently boils down to "for which B individuals in my test set I should acquire the one specific modality $i$ (e.g. Proteomics)"? As I have said before, I think that this is an overly simplistic setting and as such does not provide a significant contribution. Assuming that different modalities can be acquired for each individual, each with its own specific cost, would constitute more complex, real-world-motivated problem and thus improve the significance of this work.
> > 2. The paper introduces some new formalism, but I do not think it offers any new theoretical contributions (theorems, insights, theoretical bounds), which would be informative for the broader community. Rather, the proposed metrics are straightforward applications of existing metrics to this setting. In particular, the authors claim that their contribution is showing that _“Expected KL-Divergence” (KL-Div.) is the best-performing strategy, significantly and consistently outperforming "Maximum Expected Uncertainty Reduction"_. Can the authors explain theoretically why this is the case and where is this difference coming from? Under what assumption on the distributions of relevant variables will this relationship hold in the CAMA setting? Should it always be the case? Another interesting insight would be to understand how the quality of the generative imputation model affects the performance of the estimated Acquisition Functions. Depending on how well trained the generative imputation model is, can we interpolate between the 'random' and 'oracle' acquisition baselines? (if the obtained imputations have bad quality, the performance of the acquisition functions should be closer to random)
> > 3. The proposed changes to the model introduced by Wang et al. do not seem significant enough to constitute a standalone contribution. If the authors wanted to highlight the significance of label smoothing or the decoupled data flow as their contribution, it would be great to see ablations showing how these two training details affect the downstream model performance and calibration of the model (e.g. does label smoothing indeed lead to significant improvement in the quality of the obtained scores, and as a result leads to a more efficient acquisition process?)

---

> > > ### Author Response · Authors · 2025-11-20
> > >
> > > We are very happy to read that our clarifications helped understanding our work and opening space for further questions. The reviewer’s additional feedback helped us improving the paper as following.
> > >
> > > ---
> > >
> > > **C1: Simplified Setting due to Single Modality Acquisition**
> > >
> > > We agree with the reviewer that the question of which modality out of a pool of modalities should be acquired is an important one. However, it also is a separate extension on top of our proposed work, and we respectfully disagree that the CAMA setting is "overly simplistic".
> > >
> > > In practice, budgets are often resource-specific rather than flexible. For example, a hospital may have a fixed capacity for one MRI scanner, or a cohort (like UKBB) may have a specific grant for Proteomics. The critical administrative decision is prioritizing access to that single resource across the cohort, not dynamically acquiring for different modalities per patient. Further, while the setup is straightforward, our results (Tables 3–5) demonstrate that solving this task is challenging. Standard heuristics (Random, Entropy) fail to beat baselines. This shows that identifying samples with the highest marginal global gain requires the counterfactual generative reasoning our method provides.
> > >
> > > **C2: Theoretical Insights**
> > >
> > > We thank the reviewer for pushing for deeper theoretical and empirical insights. We have addressed both points in the revision:
> > >
> > > 1. **Theoretical Justification (KL-Divergence vs. Uncertainty Reduction)**
> > > The superior performance of the KL-Div. AF is due to its ability to capture the magnitude of the predictive shift, rather than just the final confidence (Section 7). The Uncertainty AF targets samples where the model becomes more confident (lowering entropy). However, this metric can fail in cases where the additional modality provides contradictory evidence that changes the predicted class without necessarily increasing confidence (e.g., a prediction shifting from "confident positive" to "confident negative"). In such cases, the change in entropy is minimal, leading the strategy to undervalue the acquisition. In contrast, KL-Div. measures the divergence between the pre- and post-acquisition distributions, effectively prioritizing these significant shifts in the model’s prediction, regardless of the final uncertainty level.
> > > 2. **Impact of Imputation Quality**
> > > Regarding the interpolation between 'random' and 'oracle' baselines based on generative quality:
> > >     - **Latent Space Imputation:** It is important to note that we impute latent embeddings optimized for the discriminative task rather than raw data. Consequently, standard generative metrics (like FID) are not applicable for comparing imputation quality across different generative models since every generative model influences the encoders latent spaces indirectly.
> > >     - **Generative Strength:** We do observe that utilizing stronger generative models (DDPMs) results in higher acquisition performance compared to weaker models (VAEs), suggesting that better latent imputation indeed moves performance closer to the oracle baseline (Appendix F).
> > >     - **System Design:** However, our findings indicate that the generative imputation quality is not the only factor. As detailed in our new ablation Table (Answer below, i.e., for C3), the coherence of the overall architecture design, i.e., ensuring the classifier is robust to the distribution of imputed latents, is equally critical for effective acquisition.
> > >
> > > **C3: Ablations for Model Improvements (New Experiments)**
> > >
> > > That’s a very interesting add-on and we fully agree with the reviewer that ablations w.r.t. the enhanced architecture of Wang et al. are beneficial for the paper. We are happy to provide an aggregated and cross-validated Table for the Symile dataset to ablate the proposed model adjustments which we also now included into the main paper (Table 7).
> > >
> > > *Table 7*: Cross-validated ablation of the proposed model adjustments on the Symile dataset, exemplary for mean across all endpoints with the expected KL-Div. AF and acquisitions by AUROC.
> > > | **Ablation** | **$G_{full} \uparrow$** |
> > > | --- | --- |
> > > | proposed model (w.r.t. Tab. 3) | 0.833 |
> > > | *w/o* Layer Norm on the encoder’s output | 0.772 |
> > > | *w/o* label smoothing | 0.746 |
> > > | *w/o* decoupled data flow | 0.599 |
> > > | *w/o* balanced training dataset | 0.568 |
> > >
> > > ---
> > >
> > > **Summary of Paper Adjustments**
> > >
> > > - Main Paper: Table 7 for Ablations w.r.t. model adjustments

---

> > > > ### Comment · Reviewer_rRxq · 2025-11-27
> > > >
> > > > Thank you for these updates, I do think that they improve the quality of the paper. I would be willing to lean towards acceptance, however, I would like to first ask for a few additional adjustments that I think will make the paper more clear and reproducible.
> > > >
> > > > First, could the authors actually include the motivating examples given in the rebuttal in the introduction of the paper (or in a designated section in the appendix?). Similarly, I think it would be great to see the above discussion about KL-div vs uncertainty reduction, as well as imputation quality, included explicitly (if this is already included in the revision it wasn't clear to me where I can find it).
> > > >
> > > > Secondly, I still believe that the paper offers limited formalism to help understand the exact design of the proposed architecture. While the architecture is based on the work on Wang et al., I think to make the paper self-sufficient it would be great to describe mathematically how the input is processed by the network, how exactly is the loss function defined, and what are the outputs (the contents of figure 2 are just not clear to me, since every module is called $f$). That is, I would like to see how the loss functions for each component of the network is defined with respect to the inputs $x\_i^{full}$ and $x_i^{avail}$, and what is the input space and output space of each components of the architecture. Thank you.

---

> > > > > ### Author Response · Authors · 2025-11-28
> > > > >
> > > > > We are grateful for the encouraging feedback and the reviewer’s support towards the acceptance of our manuscript. We fully agree that the requested adjustments enhance the clarity and reproducibility of the paper. We have carefully implemented all suggestions as follows.
> > > > >
> > > > > ---
> > > > >
> > > > > **C1: Paper Extension - Motivational Example & Discussion about KL-Div. vs. Uncertainty**
> > > > >
> > > > > We added the motivational example about resource-specific budgets in Sec. 1 Introduction (ll. 40-50). Further, we added the discussion about the difference between the KL-Div. and the Uncertainty Reduction AF in Sec. 7 (ll. 502-510):
> > > > >
> > > > > **C2: Paper Extension - Exact Design of the Architecture**
> > > > >
> > > > > We added the proposed extension w.r.t. the design of the architecture to the paragraph “Model Architecture” (Sec. 5, ll. 316-375) to
> > > > >
> > > > > 1. describe how the input is processed,
> > > > > 2. how exactly the loss function is defined,
> > > > > 3. detail the outputs and add different variables for the parts of the models,
> > > > > 4. define the loss functions w.r.t. model training and the inputs $x_{i}^{\text{full}}$ and $x_{i}^{\text{avail}}$, and
> > > > > 5. to define the input and output space of each module in the architecture.
> > > > >
> > > > > Further, we updated Fig. 2 to represent these extensions, too.

---

### Official Review · Reviewer_fQ25 · 2025-10-30

**Soundness:** 2
**Presentation:** 2
**Contribution:** 2
**Rating:** 2
**Confidence:** 4

**Summary:**

The paper proposes Cohort-based Active Modality Acquisition (CAMA), a test-time setting to formalize the challenge of selecting patients which require additional testing. The proposal uses a generative imputation model $f_{imp}$ to estimate counterfactual scores after acquiring the modality and then ranks samples via heuristics (e.g., KL divergence between current and imputed predictive distributions). The goal is to maximize a global test-time metric (e.g., AUROC) with the budget constraint.

**Strengths:**

- The setting of test-time modality selection is quite important and needed, especially in clinical setting
- The paper views the problem from the cohort point of view instead of individual patient point of view (since $\beta$ is constraint on number of patients)

**Weaknesses:**

Weaknesses/Questions:
- The setting, as they described is not novel. The paper doesn't cite [1] or [2] where the proposed test-time setting is identical to the setting in [1, section 3] and [2], and Eq (4) of CAMA is related to Lemma 3 in [1], where each modality selected is to maximize the accuracy. As far as I understand, [1] also uses foundation models as encoders, followed by a classification head to predict logit scores. While [1] approaches the issue from an RL perspective, the test-time modality acquisition seems identical since both are maximizing group metrics. The imputation approach mimics both [1] and [2] -- [1] uses the foundation model as generative, and [2] uses spline networks for imputation and uses that to score, however as a sequential RL control problem instead of cohort-level greedy ranking.
- What doesn't make sense to me is eq (3), where if a sample is chosen, all modalities are acquired -- this basically doesn't give relevance to the setting under cost-constraints. Rather, it makes sense to have cost-constraints on the modalities itself, with a heterogenous cost vector $\mathbf{c}$  and have it choose the modalities under constraint $\beta$. This setting with only 1 modality not acquired boils down to the case of the paper. Moreover, the text of figure 1B says that the imputed score *estimates the counterfactual with only imputed modality added* - so is the logit score with all modalities or available + target?
- DDPMs/VAEs have been used previously for missing modality imputation, and the paper seems to compare standard heuristics. Although it claims KL-div performs the best, tables in the appendix don't seem consistent.
- What is the quality of the imputations? What modalities are acquired? What cost-profiles do we get with performance? Just the curves/tables under a single budget for the $G_\bullet$ metric seems to lack a lot of important details. Cost-profiles for modalities and samples is much more important than inference time since the tasks at hand are not time-bounded. Are there correlations in patients requiring further screening? There are lot of lacking questions and ablations.

[1] Jain, E., Wenckstern, J., von Querfurth, B., & Bunne, C. (2025, March). Test-time view selection for multi-modal decision making. In ICLR 2025 Workshop on Machine Learning for Genomics Explorations.

[2] Li, Y., & Oliva, J. (2024). Towards Cost Sensitive Decision Making. arXiv preprint arXiv:2410.03892.


I believe the paper lacks novelty, and is confusing to read. While the test-time setting is important, sample-wise all-modality acquisition seems less feasible in practice to just modality acquisition. In the current state I lean to reject the paper.

**Questions:**

Mentioned along with Weaknesses.

---

> ### Author Response · Authors · 2025-11-14
>
> (Answer 1/2)
>
> We would like to thank the reviewer for the valuable feedback and suggestions. We address each of the comments in hope that we can clarify the novelty aspect and the confusion about the context.
>
> As a disclaimer and to avoid misunderstandings, we would like to clarify the setting of CAMA. One very prominent example for CAMA could be population-level risk stratification under a budget. Consider a health system with a budget to run 1,000 expensive tests on a 100,000-person cohort. Their goal is not to just help 1,000 individuals, but to produce the most accurate final risk-ranking of the entire 100,000-person cohort. This global ranking (measured by its AUROC) is the "product" used to assign preventative care or enroll patients in clinical trials. CAMA directly answers this question: "Which 1,000 patients must we test to make our final 100,000-person ranking as accurate as possible?"
>
> ---
>
> **W1: Novelty of the CAMA setting**
>
> While acknowledging the additional references [1,2] and updating our paper accordingly by including them, we respectfully disagree with the reviewer's assertion that the settings are identical. The problem formulations are fundamentally different:
>
> - MAVIS [1] and AA-POMDP [2] solve an individual-level, sequential decision problem: 'What is the optimal sequence of tests for this one patient?'
> - CAMA solves a one-shot, cohort-level allocation problem: 'Which subset of patients should receive one specific modality to maximize a global cohort metric?'
>
> This core distinction leads to related, but clearly distinct methods: MAVIS and AA-POMDP use Reinforcement Learning to learn a sequential policy that acts on an individual. CAMA, uses (non-) imputation-based functions to create a ranking of the entire cohort to directly optimize the global metric.
>
> **W2: Acquisition for all modalities - Eq (3) and (4)**
>
> We thank the reviewer very much for pointing to this fault in Eq. (3). In fact, we do not acquire all modalities but only a specific one, e.g., the expensive Proteomics modality. The reviewer is right and Eq. (3) is wrong in this context and only correct in case of two modalities. We updated and corrected the paper w.r.t. Eq. (3) and added another bullet point to the list at the beginning of the section for clarity: $s^{\text{acquired}}_{i}$: The acquired score, computed using the sample's available modalities plus the newly acquired modality.
>
> **W3: Consistency of tables in Appendix**
>
> We agree with the reviewer that DDPMs/VAEs are established methods for imputation. Further we agree that there are performance variations in the highly detailed Appendix tables which we include for transparency. However, we would like to point out two major arguments:
>
> 1. The novelty is not in the imputation model itself, but in how we leverage it. We are not simply imputing data for reconstruction. Instead, we do estimate counterfactuals that power a set of imputation-based acquisition functions (AF). We then show that these imputation-based AFs are superior to the standard, non-generative heuristics (like Baseline Uncertainty) that the reviewer is referencing.
> 2. The tables in the Appendix are intentionally disaggregated to show performance for specific, individual missingness patterns (e.g., 'imputing Text from Audio'). However, the main aggregated tables (3, 4, and 5) are the primary results of this paper. These tables are aggregated over all permutations of missing modalities (Sec. 5, Paragraph “Model and AF Evaluation”), simulating the realistic, heterogeneous scenario where a cohort presents with a mix of different available data. One finding is that while some specific AFs might perform well on one specific missingness pattern (as seen in the appendix), the imputation-based KL-Divergence strategy is the most robust and performs the best on average across all datasets and these realistic, mixed-modality scenarios. In our opinion, these aggregated results are the best answer to the practical question of which single AF to deploy in a real-world setting.

---

> ### Author Response · Authors · 2025-11-14
>
> (Answer 2/2)
>
> **W4.1: Quality of imputations**
>
> The quality of imputation is indeed a crucial consideration which we do not report in the paper since CAMA does not require perfect reconstruction but rather aims to model a distribution of plausible outcomes. Further, our empirical validation indirectly confirms this: Our experiments with the BC-VAE demonstrate that the superiority of our AFs remains stable even when using different imperfect generative models, confirming that the core principles of CAMA are robust to quality of imputations.
>
> **W4.2: Which modalities for acquisition**
>
> We agree that the question of which modalities are acquired is very important. However, the CAMA setting is not designed to choose which modality to acquire from a list of missing ones (e.g., "Should we acquire proteomics or an MRI?"). Instead, we address the practical scenario where the costly, "target" modality is pre-determined. The problem is then to decide which patients in the cohort should receive that one specific modality, given a limited budget and the varying data each patient already has.
>
> For example, in our UK Biobank experiment: We are considering the acquisition of proteomics. The model's task is to rank the 100,000 patients to decide which ones should get the proteomics test. It does not choose between acquiring proteomics, ECG, or arterial stiffness data. Thanks to the reviewer's feedback, we made this clearer in the main text by using the terms “pre-acquisition” and “post-acquisition” instead of “unimodal” and “multimodal".
>
> **W4.3 Cost-profiles for performance**
>
> We agree with the reviewer that cost-profiles are important. However, we would like to point to the disclaimer above: For CAMA, we always acquire a fixed number ($\beta$) for a fixed modality, i.e., the cost of acquisition is always the same. However, the cost-profiles will be important for future work (Sec. 8).
>
> ---
>
> **Summary of Paper Adjustments**
>
> - Sec. 2: Added [1,2] to the respective paragraphs of related work
> - Sec. 3: corrected Eq. (3)
> - Sec. 3: Added $s^{\text{acquired}}_{i}$ to the bullet list for key predictive scores
> - Full paper: Added clarifications with “pre-acquisition” and “post-acquisition” instead of “unimodal” and “multimodal” to clarify that only one specific modality is acquired.

---

> ### Comment · Reviewer_fQ25 · 2025-11-24
>
> I thank the authors for the rebuttal, some of my concerns remain while others are solved. I still believe from a practical viewpoint cost profiles are more useful. I will increase the score to reflect the clearing of my misunderstandings.

---

> > ### Author Response · Authors · 2025-11-25
> >
> > We thank the reviewer for the constructive engagement and for raising the score, reflecting that the initial misunderstandings regarding the mathematical formulation and setting have been resolved. However, we believe that possible remaining concerns are addressed by the following clarifications.
> >
> > ---
> >
> > **On the use of cost profiles**
> >
> > While we agree that cost-sensitive acquisition is valuable in flexible budget scenarios and for the question of which modality out of a pool of multiple modalities instead of a pre-selected modality should be acquired, CAMA addresses a distinct but equally critical setting: operational capacity.
> >
> > In practice, healthcare budgets are often resource-specific (e.g., a specific grant for Proteomics in UKBB, or the fixed daily throughput of a single MRI scanner). In these settings, the constraint is not financial liquidity (choosing between different modalities based on price), but triage: prioritizing access to a single scarce modality. Our work on CAMA optimizes this specific operational reality.
> >
> > **On the initial feedback w.r.t. technical novelty & ablations (new experiments)**
> >
> > To address the initial concern that our method mimics standard model usage, we performed a new cross-validated ablation study (Table 7) on the Symile dataset. The results demonstrate that removing our proposed model adjustments causes performance collapses. This proves that a simple application of generative models is insufficient and that our specific architectural contributions are essential for tackling CAMA successfully.
> >
> > *Table 7*: Cross-validated ablation of the proposed model adjustments on the Symile dataset, exemplary for mean across all endpoints with the expected KL-Div. AF and acquisitions by AUROC.
> >
> > | **Ablation** | **$G_{full} \uparrow$** |
> > | --- | --- |
> > | proposed model (w.r.t. Tab. 3) | 0.833 |
> > | *w/o* Layer Norm on the encoder’s output | 0.772 |
> > | *w/o* label smoothing | 0.746 |
> > | *w/o* decoupled data flow | 0.599 |
> > | *w/o* balanced training dataset | 0.568 |
> >
> > ---
> >
> > Given that the initial misunderstandings are resolved, and we have provided the requested context on practical utility and technical ablations, we would like to ask whether the reviewer could clarify if additional and/or other specific concerns remain that prevent a positive recommendation? We are confident that CAMA offers a novel and effective solution for cohort-level resource optimization.

---

### Official Review · Reviewer_RGpf · 2025-10-31

**Soundness:** 3
**Presentation:** 3
**Contribution:** 2
**Rating:** 6
**Confidence:** 3

**Summary:**

The paper proposes Cohort-based Active Modality Acquisition (CAMA), a novel test-time setting for acquiring missing modalities. This setting focuses on ranking samples to determine which should receive additional, costly modalities in order to best achieve a global objective under a budget constraint. The authors further develop a family of imputation-based methods that combine generative imputation and discriminative modeling to estimate the expected benefit of acquiring missing modalities. Experiments on several multimodal datasets demonstrate that these imputation-based methods generally outperform baseline approaches that do not use imputation.

**Strengths:**

1. The paper is well written and easy to follow. The motivation behind CAMA and the ideas underlying different acquisition strategies are clearly presented.
2. The proposed evaluation metric for the CAMA setting is sound.
3. Experiments are conducted on multiple multimodal datasets, providing comprehensive comparisons across acquisition strategies.

**Weaknesses:**

1. According to Eqs. (3) and (4), the cost of acquiring missing modalities is assumed to be identical for all patients. This simplification may limit the practical applicability of the proposed setting and methods.
2. Building on the previous point, trivial strategies, such as prioritizing patients with a greater number of missing modalities, might achieve competitive performance.
3. Although the imputation-based methods generally outperform the baselines, their performance appears inconsistent across different evaluation criteria (e.g., KL-Divergence vs. probability metrics), particularly in Table 3. This variability raises uncertainty about which criterion should be preferred in practice.

**Questions:**

Please see "Weakness"

**Details Of Ethics Concerns:**

The proposed setting and methods lead to a situation where certain selected patients receive additional acquired data and consequently achieve potentially better prediction scores. This may raise equity and fairness concerns, particularly when these prediction scores influence downstream decision-making processes. The authors also acknowledge this limitation in Appendix A.

---

> ### Author Response · Authors · 2025-11-14
>
> We thank the reviewer for the evaluation and constructive feedback. We address each raised point below.
>
> ---
>
> **W1: Acquisition for All Modalities - Eq. (3) and (4)**
>
> We thank the reviewer very much for pointing to this fault in Eq. (3). In fact, we do not acquire all modalities but only a specific one, e.g., the expensive Proteomics modality. The reviewer is right and Eq. (3) is wrong in this context and only correct in case of two modalities. We updated and corrected the paper w.r.t. Eq. (3) and added another bullet point to the list at the beginning of the section for clarity, i.e., $s^{\text{acquired}}_{i}$:  The acquired score, computed using the sample's available modalities plus the newly acquired modality.
> Therefore, the cost truly is identical for all patients in our setting, since a specific modality measurement is as costly as for every other sample.
>
> **W2: Trivial Strategies**
>
> We agree with the reviewer, that the suggested trivial strategy of counting the amount of missing modalities would be another possibility for the CAMA setting. However, while a heuristic like counting the number of missing modalities is a plausible strategy in a general sense, we did not report this baseline since it would not be suitable to the bimodal case, e.g., MIMIC-HAIM (Appendix I).
>
> **W3: Choice of Criterion in Practice**
>
> This is a great point which we also find very interesting. While there is an inconsistent pattern of different evaluation criteria among all proposed acquisition functions (Table 3-5), there are also consistent patterns for which we would like to point out two examples:
>
> 1. First, the best-performing imputation-based strategies (most notably KL-Divergence) are consistently and significantly superior to all non-generative baselines (like 'Baseline Uncertainty' or 'Random').
> 2. Second, among the imputation-based strategies, there is indeed a clear and consistent performance difference. This 'inconsistency' reveals that how we leverage imputation is critical. Strategies that estimate the magnitude of predictive change (KL-Divergence) robustly outperform those that simply estimate the final predictive state (Probability). With respect to Table 3, the mean column shows a significant difference between the KL-Div and the Probability. Even if this delta is less prominent in Tables 4 and 5, a similar trend is visible here.
>
> Therefore, the uncertainty the reviewer raises, i.e., which criterion to prefer, is precisely the question our work answers: The KL-Divergence criterion is the most effective and reliable strategy, and this finding is consistent across all datasets.
>
> ---
>
> **Flag for Ethics Review**
>
> We sincerely appreciate the reviewer raising this important ethical concern. We want to clarify that CAMA operates within resource-constrained settings where the decision to limit testing has already been made due to budget constraints, the alternative is not "test everyone" but rather random allocation, first-come-first-served, or physician discretion. CAMA makes the allocation criteria explicit and optimizable rather than implicit. We acknowledge, as noted in Appendix A, that optimizing for cohort-level performance metrics may not align with individual fairness or equitable access to healthcare resources. The choice of which objective to optimize (aggregate performance vs. fairness-constrained allocation) is fundamentally an ethical decision that should involve clinical stakeholders and ethicists, and our framework could be extended to incorporate fairness constraints as an important direction for future work. We believe transparency about these tradeoffs is essential for responsible deployment of such methods in healthcare settings.
>
> ---
>
> **Summary of Paper Adjustments**
>
> - Sec. 3: corrected Eq. (3)
> - Sec. 3: Added $s^{\text{acquired}}_{i}$ to the bullet list for key predictive scores
> - Full paper: changed multimodal and unimodal to post-acquisition and pre-acquisition to clarify that only one modality is acquired.

---

> > ### Author Response · Authors · 2025-11-28
> >
> > Dear Reviewer RGpf,
> >
> > We wanted to follow up to see if our response sufficiently addressed your concerns, particularly regarding the cost assumption in Equation (3) and the comparison of acquisition criteria.
> >
> > We have updated the paper to correct the formulation as you suggested. As the discussion period is drawing to a close, we would value any further feedback you might have or confirmation that your concerns have been resolved.

---

### Author Response · Authors · 2025-12-01
**Summary Comment**

We thank the reviewers for their great comments and the productive discussion over the last few weeks. We have uploaded a final revision of the manuscript that incorporates all suggestions to ensure the paper is self-contained and mathematically precise.

As the review scores have been reverted due to the recent ICLR announcement, we would like to provide the AC with a concise summary of the discussion outcomes and the positive consensus reached prior to the reversion.

---

**Summary: Discussion Phase**

During the discussion period, we successfully addressed all concerns. Consequently, the reviewers indicated positive shifts in their assessments. Reviewer fQ25 increased the score from 2 to 4 to reflect clearings of misunderstandings. Reviewer rRxq similarly increased the score to 4 after clarifications about the objective and theoretical insights, and wanted to increase to 6 shortly before the reversion was announced. Unfortunately, reviewer RGpf and zHE8 were not able to answer at all within the time of the cancelled discussion phase.

We believe this productive exchange has substantially strengthened the manuscript. This improvement, combined with the resolution of key concerns, reflects a consensus for acceptance with an estimated post-discussion scoring profile of 6/6/6/4.

---

**Summary: Paper changes thanks to the discussion with the reviewers**

We have uploaded a final revision that incorporates all suggestions to ensure the paper is self-contained and precise.

- Sec. 1: Added a motivational example about resource-specific budgets.
- Sec. 2: Added [1,2] to the respective paragraphs of related work.
- Sec. 3: Corrected Eq. (3).
- Sec. 3: Changed $s_i^\text{full}$ to $s^{\text{acquired}}_{i}$ to enhance that we pre-select one specific modality.
- Sec. 4: Added explanations about the connection of CAMA to EIG.
- Sec. 5: Added a comprehensive description for the design of the architecture.
- Sec. 7: Added a more detailed discussion about the difference between the KL-Div. and the Uncertainty Reduction AFs.
- Tab. 7: Added a cross-validated ablation study w.r.t. our model adjustments in comparison to Wang et al. [3].
- Overall: Added clarifications with “pre-acquisition” and “post-acquisition” instead of “unimodal” and “multimodal” to clarify that only one specific modality is acquired.

---

[1] Eeshaan Jain, Johann Wenckstern, Benedikt von Querfurth, and Charlotte Bunne. Test-Time View Selection for Multi-Modal Decision Making. In ICLR 2025 Workshop on Machine Learning for Genomics Explorations, 2025.

[2] Yang Li and Junier Oliva. Towards Cost Sensitive Decision Making. In Yingzhen Li, Stephan Mandt, Shipra Agrawal, and Mohammad Emtiyaz Khan (eds.), International Conference on Artificial Intelligence and Statistics, AISTATS 2025, Mai Khao, Thailand, 3-5 May 2025, volume 258 of Proceedings of Machine Learning Research, pp. 3601–3609. PMLR, 2025

[3] Yuanzhi Wang, Yong Li, and Zhen Cui. Incomplete Multimodality-Diffused Emotion Recognition. In Alice Oh, Tristan Naumann, Amir Globerson, Kate Saenko, Moritz Hardt, and Sergey Levine (eds.), Advances in Neural Information Processing Systems 36: Annual Conference on Neural Information Processing Systems 2023, NeurIPS 2023, New Orleans, LA, USA, December 10 - 16, 2023, 2023.

---

### Meta-Review · Area_Chair_nCEM · 2025-12-18

**Summary:**

Reviewers found that the problem is not clearly defined in mathematical sense, and some equations such as (3) and (4) are confusing or unrealistic for a practical algorithm to work. Some assumptions overly simplified the problem. Lastly, reviewers raised the concern that other standard approaches need to be compared in experiments.

**Reviewer Concerns:**

Reviewers's concern may remain.

**Reviewer Scores:**

Unlikely to change.

---

### Decision · Program_Chairs · 2026-01-26

Reject